# Analyzing the chemical composition, morphology and size of ice-nucleating particles by coupling a scanning electron microscope to an offline diffusion chamber

Lisa Schneider[1], Jann Schrod[2], Daniel Weber[2,a], Heinz Bingemer[2], Konrad Kandler[1], Joachim Curtius[2], Martin Ebert[1]

[1]Institute of Applied Geoscience, Technical University of Darmstadt, Darmstadt, 64287, Germany
[2]Institute for Atmospheric and Environmental Sciences, Goethe University Frankfurt, Frankfurt am Main, 60438, Germany
[a]now at: Federal Waterways Engineering and Research Institute, Karlsruhe, 76187, Germany

*Correspondence to*: Martin Ebert (mebert@geo.tu-darmstadt.de)

**Abstract.** To understand and predict the formation of clouds and precipitation and their influence on our climate, it is crucial to know the characteristics and abundance of ice-nucleating particles (INPs) in the atmosphere. As the ice-nucleating efficiency is a result of individual particle properties, a detailed knowledge on these properties is essential. Here, an offline method for the comprehensive single particle analysis of ambient INPs that benefits from the combination of two instruments already used for ice nucleation measurements is presented, focusing on the methodological description of the coupling, whereby strengths and weaknesses of the method are discussed.

First, the aerosol is sampled on silicon wafers. INPs are then activated at different temperature and humidity conditions in the deposition nucleation and condensation freezing mode using a static diffusion chamber. The positions of grown ice crystals are defined by a coordinate system, which allows for recovery and detailed analysis of the individual INPs by a scanning electron microscope. Based on their physico-chemical properties (elemental composition and morphology) the INPs can be classified into categories. In combination with the size information, a size-resolved distribution of the INP classes can be determined. Such results are useful for evaluating INP type-specific parametrizations, e.g., for use in atmospheric modeling, and in closure studies.

A case study from the high-altitude research station Jungfraujoch, Switzerland shows that the targeted INP analysis as obtained by this method is able to identify the main INP classes in reliable proportions. Most of the deposition nucleation / condensation freezing mode INPs activated at -30 °C, indicated a geogenic mineral origin (mainly aluminosilicates / Al-rich particles, but also carbonates and silica). Other major contributors were carbonaceous particles, consisting of both smaller soot particles and larger biological particles, and mixed particles (mostly Al/C mixed particles). The INPs had projected area diameters ranging from 300 nm to 35 µm, with a distinct maximum at 1 - 2 µm. Mineral particles were present throughout the entire size range, while mixed particles were identified in higher abundances at sizes of 3 µm and above. Minor contributions were seen from sulfates and metal oxides, the latter with an increased proportion in the size range below 500 nm. During a Saharan dust event, a significant increase of mineral particles in the INP composition was detected.

# 1 Introduction

Ice-nucleating particles (INPs) have a significant impact on climate and weather. They influence cloud formation and thus have an effect on cloud structure, extent and lifetime, as well as on radiation and precipitation properties (Kanji et al., 2017, and references therein).

Ice formation in the atmosphere can be initiated via several mechanisms depending on ambient conditions. At temperatures below approximately -38 °C, supercooled solution droplets may freeze spontaneously without a crystallization nucleus (homogeneous freezing). To start ice formation at warmer temperatures (T > -38 °C), the energy barrier for nucleation has to be reduced. This can be accomplished by the presence of an INP. Conventionally, four mechanisms are distinguished for heterogeneous freezing processes: (1) deposition nucleation, (2) condensation freezing, (3) contact freezing and (4) immersion freezing. Detailed information on ice nucleation terminology can be found in Vali et al. (2015) and Kanji et al. (2017).

Only a small fraction of the total aerosol can act as INPs and their concentrations can show variations of several orders of magnitude in space and time (DeMott et al., 2010; Kanji et al., 2017). In addition to the prevailing environmental conditions, i.e., temperature and humidity, the potential for an INP to be activated depends on individual particle properties, e.g., surface imperfections (Kiselev et al., 2017), chemical composition and specific chemical properties, crystal structure, coating (Kanji et al., 2008), etc., as well as its atmospheric processing, including potential agglomeration or pre-activation (Marcolli, 2017). The promoting sites on the surface of an INP are termed active sites. As particle size increases, the number of active sites also tends to increase. So typically, the larger an atmospheric particle is, the more likely it is to act as an INP (Archuleta et al., 2005; Welti et al., 2009).

Ice-forming activity has been verified for many atmospheric particle classes. Mineral dust, which is emitted from arid and semi-arid regions and is globally distributed in the atmosphere (Ansmann et al., 2003; Perry et al., 1997; Schepanski, 2018), is a good INP at temperatures below -15 °C (Hoose and Möhler, 2012). The composition of mineral dust is highly variable depending on the source region (Scheuvens and Kandler, 2014). Furthermore, soil dust (mineral dust which is often mixed internally with organic components) from agricultural regions is regarded to be a source of INPs (O'Sullivan et al., 2014). Metal oxides can be components of mineral dust from natural sources and are also emitted by anthropogenic sources. Their efficiency to activate as INPs depends on the type of metallic cation as well as on the oxidation state (Archuleta et al., 2005; Yakobi-Hancock et al., 2013). At temperatures warmer than -15 °C, mainly biological INPs are ice active (Després et al., 2012, and references therein). These include primary particles such as bacteria, fungal spore, pollen and plant debris, as well as some biological macromolecules (Pummer et al., 2012). Particles from biomass burning and fossil fuel combustion are considered as another particle class relevant for ice formation. This includes soot (mostly a mixture of black carbon with organic carbon) as a product of incomplete combustion as well as fly ash from the non-combustible components. However, the contribution of soot to atmospheric ice formation is still subject of discussion, e.g., Cozic et al. (2008) and Kupiszewski

et al. (2016) found opposing results at the same location. Besides the continental sources, the oceans also serve as a source for atmospheric INPs. In addition to sea salt, sea spray aerosol also contains increased amounts of marine organic material from the sea surface microlayer, which is considered to have significant ice-nucleating properties (Wilson et al., 2015). A detailed overview of all atmospherically relevant INPs is given by Kanji et al. (2017) and Burrows et al. (2022).

Although there is a variety of methods to determine the INP concentration in the laboratory (DeMott et al., 2018; Hiranuma et al., 2015, 2019; Hoose and Möhler, 2012) and in the field (Brasseur et al., 2022; Lacher et al., 2024; Schrod et al., 2020b; Wex et al., 2019), only a few of them are simultaneously able to report on the chemical characteristics of individual ice-nucleating particles.

Most analytical methods for the chemical characterization of ambient ice nuclei are based on the principle of first separating
the ice nuclei or ice particles from the total atmospheric aerosol using different approaches. Ice crystals can be separated directly from clouds by using specialized inlets (Kupiszewski et al., 2015; Mertes et al., 2007; Schenk et al., 2014; Schwarzenboeck et al., 2000). In this case, the particles are heated after separation so that the water evaporates and ice residuals (IRs) remain. In another approach, the particles are activated under defined conditions in an online reaction chamber (e.g., Rogers, 1988) after the collection of the total aerosol. To analyze the activated particles, it is necessary to
separate the ice crystals from droplets and evaporate the ice by one of the specialized inlet systems or a droplet evaporation zone.

In a second step, the separated INPs/IRs are then either analyzed in the air stream or separated and transferred to an offline analysis. To our knowledge, the only online experiment that has been used to determine all particle groups relevant to ice nucleation simultaneously is single particle mass spectrometry (SPMS) (Brands et al., 2011; Kamphus et al., 2010; Thomson
et al., 2000). In general, online methods allow a real-time analysis with the potential of high time resolution but may have problems at low INP concentrations. However, several studies have reliably demonstrated the coupling between a separation technique and SPMS in a field setting (Cozic et al., 2008; Cziczo et al., 2009, 2013; Kamphus et al., 2010; Lacher et al., 2021; Pratt et al., 2009; Schmidt et al., 2017). Electron microscopy (EM), as an offline method, offers an alternative approach to study the chemical composition of INPs/IRs. For single particle analysis by EM, INPs or IRs are collected on
substrates after evaporating the ice phase. Single particle analysis can be performed automated for large data sets or manually with operator control (Eriksen Hammer et al., 2019). Even though the method cannot provide high temporal resolution measurements due to longer sampling times, it can provide detailed information on morphology in addition to chemistry and size of individual INPs and IRs (China et al., 2017; Cziczo et al., 2009, 2013; Ebert et al., 2011; Eriksen Hammer et al., 2018; Knopf et al., 2014; McCluskey et al., 2014; Mertes et al., 2007; Prenni et al., 2009; Wang et al., 2012;
Worringen et al., 2015).

These methods typically analyze large numbers of INPs/IRs. However, the major challenge of all these methods is that due to the extremely low number of INPs within a sampled air volume compared to the much higher number of non-INP particles (ratio ~ $1/10^4 – 1/10^6$), the separation must be carried out with a very high accuracy. Conclusions about the

chemistry of INPs may not be entirely accurate, since there is no way to distinguish particles that have been falsely separated as INPs from real INPs afterwards. There is also the risk that additional artifacts can be introduced into the INP fraction during the multi-step process. This problem is partially illustrated in the comparison of the chemical analysis of the INPs/IRs fraction by three different methods in Worringen et al. (2015).

This paper describes an offline method for measuring atmospheric INP concentration in combination with a subsequent characterization of the activated INPs. The recently established method couples the ice nucleation counter FRankfurt Ice nucleation Deposition freezinG Experiment (FRIDGE) to a scanning electron microscope (SEM). With this method, individual particles can be specifically analyzed, of which it is known that the ice formation has taken place on the substrate exactly at their position. The FRIDGE-SEM-coupling technique has been used for several campaigns in recent years, providing valuable results (He et al., 2023; Schrod et al., 2017, 2020b; Weber, 2019). Details of the FRIDGE method were described by Schrod et al. (2016), but the coupling has not been described in detail in previous studies.

The present publication expands significantly on these studies by detailing the technical procedure to gain reliable information on physico-chemical properties of INPs by SEM, which were previously activated in FRIDGE. The first part of the paper presents a detailed description of the FRIDGE-SEM coupling - highlighting the strengths and discussing the weaknesses - followed by the results of a case study from the high-altitude research station Jungfraujoch (JFJ), Switzerland in 2017.

## 2 Methodology: Coupling a scanning electron microscope to an ice nucleus counter

The here presented offline coupling procedure (Fig. 1) combines the advantage of two devices which are already used for several years in the field of INP/IR research. Particles can be collected from ambient aerosol onto substrates by electrostatic precipitation (Fig. 1A). The sampled aerosol is then analyzed with respect to its ice nucleation ability at various combinations of activation temperature and supersaturation with respect to ice, yielding the INP concentration (Fig. 1B). The activated INPs can subsequently be characterized by SEM to gain information on their elemental composition, morphology and size (Fig. 1C). Based on the definition by Cziczo et al., (2017) we refer to the identified particles as INPs, as they were activated under defined conditions after the collection of the total aerosol and not sampled as ice crystals. Therefore, we are able to investigate truly activated particles in contrast to methods analyzing IRs, which face challenges in order to distinguish between IRs and scavenged particles. However, some of the INPs analyzed with SEM may have undergone changes due to the measurement procedure in FRIDGE, and thus be announced as IRs. But we assume that these changes are of minor importance for the main INP classes that we can analyze with this method (see Sect. 2.6), which is why we have decided to continue referring to them as INPs.

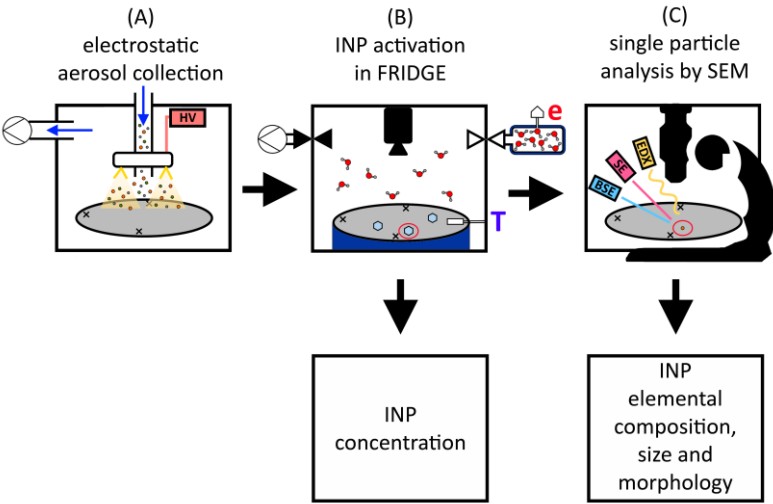

**Figure 1: Schematic for the coupled INP analysis. (A) Electrostatic Aerosol Collector (EAC): The aerosol (colored particles) is pumped through the sampling unit in the direction of the blue arrows. By applying a high voltage (HV), gold filaments (yellow) arranged in a ring generate an electrostatic field (light yellow triangles). The charged particles are deposited on the silicon substrate (gray). (B) FRIDGE: Diffusion chamber setup with silicon substrate (gray) placed on the cold stage (dark blue) equipped with a temperature sensor (T). The water vapor pressure (e) in the water vapor source (top right) is measured with a pressure sensor. A vacuum pump (top left) is used to evacuate the reaction chamber before the measurement starts. Starting a measurement, the valve to the water vapor source is opened, the diffusion chamber gets flooded with water vapor (red/grey molecules) and ice crystals (light blue) start to grow on the substrate. A camera (top center) monitors the ice growth and records images. This can be used to calculate an INP number concentration. (C) SEM: Silicon substrate (grey) with the three engraved crosses defining the coordinate system, placed in the scanning electron microscope (black schematic) equipped with detectors for back scattered electrons (BSE) (blue) and secondary electrons (SE) (red) and an energy dispersive x-ray detector (EDX) (yellow)). Each individual INP which induced ice crystal growth in FRIDGE (particle circled in red) is analyzed (elemental composition, size and morphology).**

## 2.1 Sample substrates

A silicon disk with a diameter of 45 mm serves as the sample substrate, on which the aerosol particles are deposited by electrostatic precipitation (Sect. 2.2). The semi-conductive substrate is made from commercially available silicon wafers, which are widely used as basis for microchips in electronic devices. The pure crystalline silicon surface is highly inefficient for ice-nucleation, which prevents random icing on the wafer, that would induce an artificial background signal and would lead to incorrect INP concentrations. The extremely smooth wafer surface allows for an unambiguous separation of particles from the background in the electron microscope. Each wafer is marked with three laser-engraved crosses near the edge, which span a 90° angle. These markers define a coordinate system, which allows for the precise localization of the ice-nucleation spots (Sect. 2.4 and Sect. 2.5.1).

After analysis, the wafer substrates are cleaned in a simple two-step process and can be reused subsequently. For this, wafers are pre-cleaned with ethanol and laboratory wipes (Kimtech Science, 7557, Kimberly-Clark) to eliminate oil residues from previous measurements and other coarse contamination. Then, in order to remove fine particles from the surface, the substrates are treated with a beam of dry ice crystals (Sno-Gun II, Va-Trans System, Inc.). The cleaning procedure is performed inside a particle free work space (SPECTEC, laminar flow box FBS). To verify the cleaning process, randomly selected cleaned wafers are analyzed in the ice nucleation chamber. However, even after thorough cleaning a small amount of ice formation can regularly be observed at temperatures at or below -30°C, constituting the background concentration and defining the limit of detection. This limit is volume-dependent and typically in the order of 0.1 $L^{-1}$ of atmospheric air for a collection volume of 100 L. However, for the case study, an especially low background value of 0.03 $L^{-1}$ was achieved (based on a collection volume of 100 L), which decreased further to between 0.026 $L^{-1}$ and 0.001 $L^{-1}$ when the actual collection volumes were considered.

## 2.2 Electrostatic Aerosol Collector

Aerosol is precipitated onto the substrates using an electrostatic aerosol collector (EAC) (Klein et al., 2010). Several EACs have been deployed for the use in the laboratory (DeMott et al., 2018), in field campaigns (DeMott et al., 2025), for measurements with unmanned aerial vehicles (Schrod et al., 2017), and for long-term observations at research stations (Schrod et al., 2020b). The most recent version, which was also used in the case study (Sect. 3), PEAC7, is a programmable EAC (Schrod et al., 2016) designed for semi-automated operation for one week of daily sampling.

Inside the collection unit, sample air passes through the corona discharge unit, which charges the particles negatively when a high voltage of about 12kV is applied. Charged aerosol particles follow the electric field to the grounded plate and are deposited on the silicon wafer substrate (Fig. 1A). This sampling process leads to a rather homogeneous particle distribution on the wafer, which is of great importance both for the measurements in the ice nucleation chamber and for the later individual particle analysis by EM. However, not all particles are deposited on the wafer during electrostatic precipitation,

some are deposited elsewhere in the system. Characterization experiments determined a size-independent collection efficiency of 60 % in the 0.5-3 µm size range (Schrod et al., 2016).

## 2.3 FRIDGE

The FRankfurt Ice nucleation Deposition freezinG Experiment (FRIDGE) is an offline isostatic vacuum diffusion chamber in which the activation of atmospheric INPs and the associated growth of ice crystals can be observed and documented under laboratory conditions (Bundke et al., 2008; Klein et al., 2010). The diffusion chamber addresses the deposition nucleation and condensation freezing modes (re-evaluated by Schrod et al. (2016)), but the instrument can also be used in a different setup as a droplet freezing device to address the immersion freezing mode (Boose et al., 2016; Schrod et al., 2020a). As the immersion mode setup is not subject of our study, the following section describes the measurement procedure for the deposition nucleation and condensation freezing modes (schematic shown in Fig. 1B), with a particular emphasis on the desired coupling.

For coupling the INP activation experiment to the single particle analysis by SEM, it is important to keep the three laser-engraved crosses on the wafer surface visible during the FRIDGE measurement (see Fig.1 and Fig. S1 from the supplement). The temperature sensor (PT1000) is therefore attached opposite to the middle cross (see Fig.1B). A small amount of silicon oil is applied on the bottom of the wafer as well as on the temperature sensor to ensure good thermal contact and a homogeneous temperature distribution. The temperature variance is estimated to be below 0.5°C across the entire wafer (DeMott et al., 2025). When the chamber is evacuated and the selected activation conditions are stable, the water vapor diffuses into the cold chamber, activating the INPs on the wafer surface. The growth of ice crystals is observed as a function of time (time step of 10 s) by a CCD camera (2/3" CCD $\geq$ 5 megapixels, 1 pixel $\approx$ 400 µm$^2$) placed above the reaction chamber. Ice crystals are identified by an image analysis software (LabView) comparing the brightness of new objects to a previously recorded reference image. Details can be found in Schrod et al. (2016). For the coupling procedure it is beneficial to stop the growth of ice before individual ice crystals grow to large sizes or coalescence, because the determination of the ice crystal center (Sect. 2.4), which is assumed to be the position of the INP, is more precise with small crystals. Additionally, this also reduces the spatial extent of potential particle shift during the ice crystal growth. By directly evaporating the ice crystals at the end of a measurement cycle with the objective of avoiding the liquid phase, the risk of possible particle shift is also reduced.

As a routine, the wafers are measured in 12 cycles combining three temperatures (T = -20 °C, -25 °C, -30 °C) and four relative humidities (RH) (RH = 95 %, 97 %, 99 %, 101 %). An efficient evacuation between measurement cycles is necessary to ensure the complete water evaporation from the particles to avoid pre-activation effects from residual water/ice in microscopic cavities on the particles surface (Jing et al., 2022). Based on the ice crystal numbers, the collection volume and the PEAC7 sampling efficiency, the INP concentration at different temperature and humidity settings is calculated.

For the subsequent EM with energy dispersive x-ray spectroscopy, it is important to completely remove the oil from the edge and underside of the wafer after the FRIDGE measurements, as otherwise the chemical analysis of the INPs can be

influenced by the evaporating oil. Because of this oil-removing step, the edge and the area of the temperature sensor are excluded from further analysis.

## 2.4 Identification of ice crystal positions

To match the ice crystals formed in FRIDGE to their corresponding ice-nucleating particles in SEM, the origin of each ice crystal must be transferred from the pixel coordinates of the FRIDGE images into the SEM coordinate system used for locating the SEM stage at the coordinates of interest. A scheme is given in Fig. 3. The ice crystal positions are identified by image analysis using the internal particle analyzer of the free image processing software ImageJ (Schneider et al., 2012), the coordinate transfer is carried out by using a wafer internal coordinate system.

In a first step, parameters for an affine transformation from FRIDGE image coordinates into the wafer internal coordinate system must be obtained. The wafer internal coordinate system is defined by the three laser-engraved crosses serving as reference points, which define another Cartesian coordinate system. The FRIDGE image coordinates of these reference points have to be defined manually by tagging the centers of the three crosses in a reference image as their non-uniformity prevents an automated approach. Transformation parameters are then calculated from the image pixel coordinates of the

defined reference points.

In a second step, the software identifies the image with the highest number of ice crystals for each measurement cycle and reports the ice crystal positions in the images as the center of an area exceeding a size threshold of 30 pixels with a minimum brightness of 30 of 256 on a scale stretching from the darkest to the brightest recorded signal (Schrod et al., 2016). Two separate ice crystals are detected as long as they are separated by at least one pixel that falls below the brightness threshold.

If this is not the case, a center point is determined for the entire area. The image coordinates of these center point positions are projected into the wafer internal Cartesian coordinate system (Fig. 3a) defined by the calibration marks. As a result, all ice crystals can now be assigned corresponding coordinates of the form $[X_n/Y_n]$ in the wafer internal normalized coordinate system by retaining the base vector but assigning new coordinates to the three selected calibration points, giving the base vectors a standard length ($l = 100$). It can be safely assumed that these coordinates represent the position of the

corresponding INP with sufficient accuracy, as in general a radially symmetric ice crystal growth can be observed in the range of the selected activation conditions in FRIDGE. Nevertheless, a potentially imperfect radial symmetry of the ice crystal growth, coupled with the restricted resolution of the FRIDGE images ($20 \times 20$ µm), may result in an uncertainty in the identification of the ice crystal origin. As the size of an ice crystal increases, the probability and extent of such a non-symmetrical growth increase, too. The quantification of this uncertainty proved to be difficult, as it depends on the

apparently random symmetry deviation. To reduce this uncertainty due to imperfect symmetry, the ice crystal position identification should be performed on images with smaller ice crystals, i.e., shortly after the initial activation. For this process, a homogeneous distribution of ice crystals on the substrate with an appropriate crystal density range is also desirable.

The wafer internal coordinates can now be projected into the SEM internal coordinates using a second affine transformation (see Sect. 2.5.1).

Figure 2 shows a comparison between the number of ice crystals counted by FRIDGE (parameters from (Schrod et al., 2016)) and the number of ice crystal positions identified for the SEM analysis by the image analysis procedure described above. Usually, almost all ice crystal positions identified by the counting algorithm can be related to real grown ice crystals, defined as clearly visible bright objects, which continue to grow in the ice supersaturated regime as the measuring time progresses. The few deviations are caused by various reasons. Higher numbers of ice crystal positions identified for SEM are mainly caused by misclassifying areas with condensation, which may occur while working close to RH = 100 %. A lower number of ice crystal positions identified for SEM is often caused by ice crystals that have grown together due to prolonged measurement in FRIDGE or by the presence of ice crystals in close proximity to one another, resulting in only one position for two or more crystals (see also Fig. S1). These positions are excluded from further analysis.

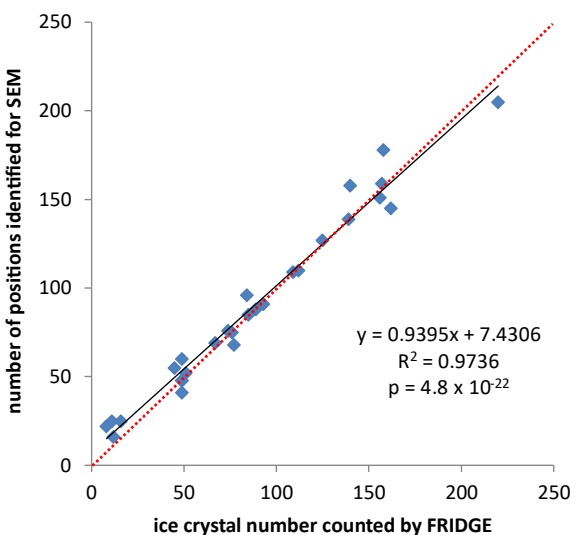

**Figure 2: Comparison of ice crystal numbers counted by FRIDGE with the number of identified positions for SEM analysis by the image analysis software. The linear regression is represented by the black line, the 1:1-line is shown in red. The regression equation relates the number of positions identified for SEM (y) to the ice crystal numbers counted by FRIDGE (x). It is given along with its coefficient of determination ($R^2$) and its p-value.**

## 2.5 Scanning Electron Microscopy

A Quanta 200 FEG environmental scanning electron microscope (ESEM) by FEI (Field Electron and Ion Company; Eindhoven, Netherlands) coupled to an energy dispersive X-ray detector (EDX) (EDAX, AMETEK, Tilburg, Netherlands) was used for analysis. The instrument is equipped with an Everhardt-Thornley detector (ETD) which maps the topology of a particle by secondary electrons (SE) and a solid-state detector (SSD), providing the distribution of elements on the particle

by backscattered electrons (BSE) giving information on homogeneous or heterogeneous distribution of elements and on inclusions. The EDX provides an elemental composition of an individual particle, which can be used to attribute the analyzed particles to different classes of compositions and sources. As all analyses were carried out in high vacuum ($10^{-6}$ mbar), the instrument is referred to as SEM in the following. The acceleration voltage was 12.5 kV or 15 kV, the working distance was 10 mm as standard.

### 2.5.1 Coordinate calibration

To find the identified position for each ice crystal / INP by SEM, its normalized wafer internal coordinates $[X_n/Y_n]$ (Fig. 3a) from the ice crystal identification step (Sect. 2.4) must be converted to instrument specific coordinates $[X_{ESEM}/Y_{ESEM}]$ by another affine transformation. This is necessary because the internal Cartesian SEM coordinate system is determined by the axes of the mechanical movement of the stage, whose position is encoded by high-precision encoders. The SEM coordinate system is centered in the middle of the stage (Fig. 3c). The parameters of this this transformation are obtained in analogy to Sect. 2.4, but now measuring the reference point locations in SEM coordinates, based on the calibration image (Fig. 3b), showing the defined wafer internal coordinate system calibration points from the previous step (Sect. 2.4). Based on the internal SEM coordinates for the calibration points, all INP coordinates $[X_n/Y_n]$ can be converted to internal SEM coordinates $[X_{SEM}/Y_{SEM}]$. It is highly important to locate these calibration points with the highest possible precision, since the position of each ice crystal in the subsequent analysis is calculated based on this calibration. Manual calibration provides the highest precision, as the systems use different physical imaging processes yielding largely different image resolutions and contrast mechanisms. The high magnification of the electron microscope (Fig. 3d) in conjunction with the surface sensitivity of electron emission, in contrast to the limited resolution of the FRIDGE calibration image (Fig. 3b) produced by visible light reflection, impair any precise automated calibration. Due to the limited resolution of the FRIDGE images of about $20 \times 20$ µm, the calibration leads to a positioning uncertainty of the same scale.

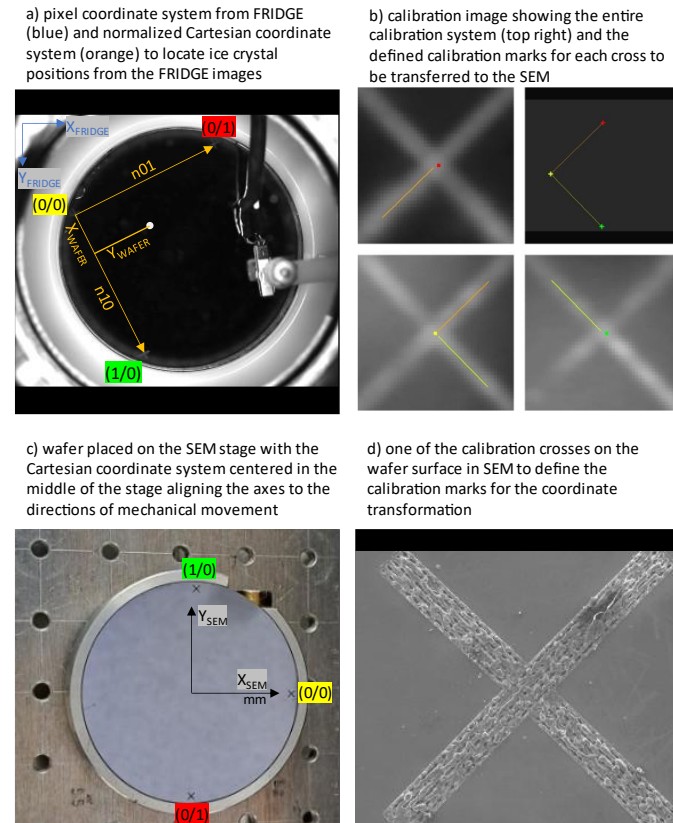

**Figure 3: a) pixel coordinate system from the FRIDGE image (blue) and wafer internal normalized Cartesian coordinate system(orange) to locate ice crystal positions (white circle) from the FRDIGE images: the normalized coordinate system (orange) is defined by two normalized vectors (n01, n10 with a length of 100) identifying a position by its x- and y-coordinate; b) calibration image showing the marked calibration points for the coordinate system from the ice crystal identification step and a picture showing the entire calibration system (top right); c) wafer on the SEM stage showing the SEM internal cartesian coordinate system aligned to the directions of the mechanical movement in millimeters (mm); d) SEM picture of a calibration cross on the wafer surface.**

### 2.5.2 Simulating the INP identification in SEM

To simulate the effects of the position uncertainty, the INP fraction and the number of particles on precision of the INP identification in SEM, a Montecarlo simulation was carried out. Figure 4 shows the conceptual model of the situation around a INP (blue) with non-INPs (yellow) and random locations around. Due to uncertainties from different sources outlined in Sect. 2.4 and Sect. 2.5.1, the location where the search starts (red cross) is in a distance to the original INP location. The circles show different search radii. If the search radius is chosen too small (inner circle), no particle will be found, if it is chosen too large, more than one particle is detected inside (out circle). The optimal search radius is at the minimum between the two probabilities.

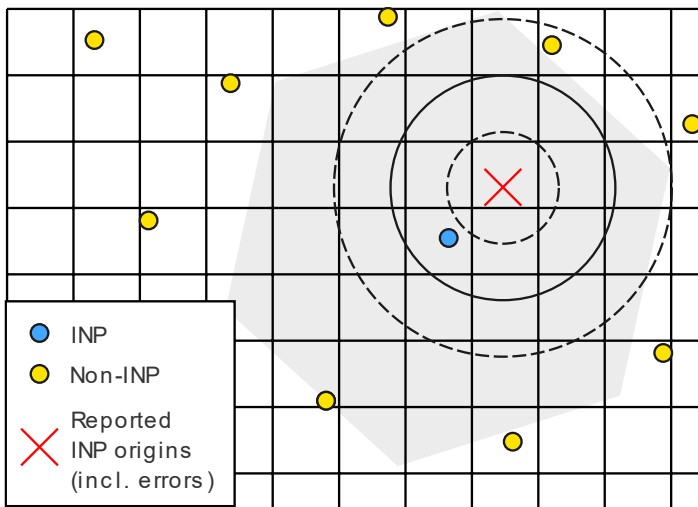

**Figure 4: Conceptual model for the simulation of the INP-recovery from FRIDGE in SEM. The blue circle is the true location of the INP, the yellow circles are non-INPs. The red cross is the search origin position in SEM. It deviates from the true INP position due to the errors by INP shift, unprecise ice crystal center determination and the re-positioning uncertainty due to calibration mismatch and mechanical backlash. The circles show search radii of 25, 50 and 75 µm from the red search origin position. The grid shows the approximate pixel size of the optical camera of FRIDGE.**

For the simulation, INP and non-INP particles in variable fractions are deposited randomly on an area of 42 mm diameter, corresponding to the active wafer surface, until a chosen total number is reached. From the known locations of the INPs a virtual starting point for the SEM search is estimated by shifting the coordinates. The direction of shift is randomly chosen, the distance is randomly sampled from a mirrored normal distribution with a standard deviation of some typical uncertainty

assumptions for the accuracy of relocating a defined point on the wafer in the electron microscope. The uncertainties associated with the particle shift and the potentially asymmetric ice crystal growth are not incorporated into the simulation, as their values are not yet sufficiently established. For each search location, the nearest particle and its type is determined as a function of search radius. This is done for all INPs on a virtual substrate. The model is then repeated until 10,000 search locations in total were processed; after 10,000 search locations the identification efficiency curves remain stable.

Figure 5 shows the results of a selection of parameters corresponding to typical application conditions (total number of 20000 / 50000 / 100000 particles on the wafer with INP fractions of 0.0005 / 0.001 / 0.002 (DeMott et al., 2017; Ren et al., 2023)). In this case the standard deviation of the position uncertainty of 25 µm was assumed. A wider range of calculation parameters is shown in Fig. S2 in the supplement. Along the search radius axis, first the cumulative probability of finding no

particle decreases; with increases search radius then, the probability of finding more than one particle increases. The

probability of falsely detecting a non-INP as INP increases as well but is mainly determined by the number of particles on the substrate and the INP fraction.

From the model simulations it seems that the number of particles on the wafer and the position uncertainty have a significant influence on the chance to find a single particle at the identified position as well as on the potential to identify a non-INP

falsely as an INP, whereas the fraction of INPs seems not to have great impact. With increasing particle numbers on the wafer, the chance to find only one particle in the vicinity of a coordinate decreases, whereby the change to identify falsely a non-INP as INP increases. With increasing position uncertainty, the required search radius increases just like the chance to identify a non-INP as INP.

Depending on the positioning uncertainty, the fraction of INP and the total number of particles the optimum search radius is

335 between 40 and 100 µm with most frequent values of 40 to 50 µm.

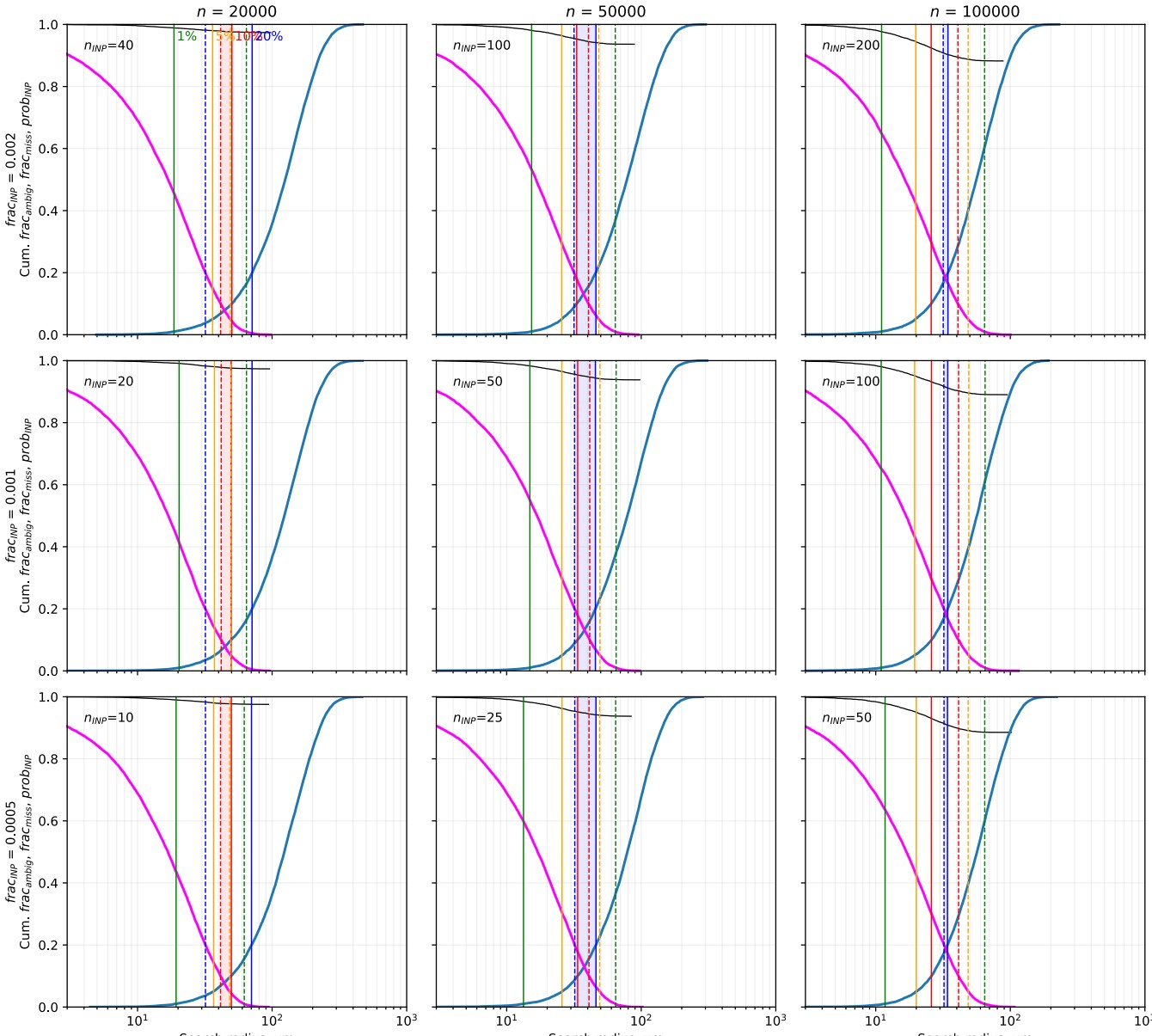

**Figure 5: Model simulation results.** Cumulative fraction of unrecovered particles $frac_{miss}$ (pink line), fraction of ambiguous locations $frac_{ambig}$ (blue), and fraction of correctly identified INPs $prob_{INP}$ (thin black) as function of search radius – originating from the reported INP position including errors - for different INP fraction $frac_{INP}$ and total particle numbers $n$. For this figure, a standard deviation of the positioning uncertainty of 25 µm is used. The total number of INPs $n_{INP}$ is shown for each plot. The search radius where 20% (blue), 10% (red), 5% (orange) and 1% (green) fraction / probability is reached is marked by vertical lines for $frac_{miss}$ (broken line) and for $frac_{ambig}$ (solid line). The lowest possible error probability range is indicated by a shaded area for below 20% (blue), below 10% (red), below 5% (orange) and below 1% (green). If no shaded area is visible, the error probability is greater than 20%.

### 2.5.3 INP identification

Each ice crystal position, based on a real grown ice crystal, is inspected by SEM to identify the presence of particles. Given the uncertainties associated with the ice crystal identification process (Sect. 2.4) and the coordinate calibration (Sect. 2.5.1), it is crucial to consider not only the exact calculated coordinate but also the surrounding area. This area must take into account the aforementioned uncertainties and, at the same time, limit the probability that several particles will be observed in the scanned area. Based on the simulations and empirical values, a radius of 50 µm around the identified coordinate was chosen as a standard search radius for the following INP identification. Since the substrate loading depends strongly on the prevailing total aerosol concentration at the sampling location, the conditions for analysis can be optimized by adjusting the search radius for different wafer loadings.

While searching for particles around the identified ice crystal position, three cases can be distinguished:

(1) If one single particle is found within the specified radius, it is considered to be the corresponding INP. If the BSEimage, in which the contrast depends on chemistry, indicates chemical differences within the particle, multiple EDX spectra of the different areas are recorded (Fig. 6 and Fig. 7). The comprehensive single particle analysis (Sect. 2.6) enables the identification of physico-chemical properties that may be pertinent to ice formation.

(2) If more than one particle is identified within the defined area around the coordinate, it is not possible to make a definite allocation of the INP. These positions have to be excluded from the analysis. Their number typically increases as the total number of particles increases.

(3) In the absence of a particle within the 50 µm radius, these blank positions are disregarded. A blank position may be the consequence of possible particle shift during the processing in FRIDGE (discussed in Sect. 2.3), or the result of an erroneous position of the ice crystal origin (discussed in Sect. 2.4). As it can be seen from the model simulations, an incorrectly selected search radius can also lead to a higher number of blank positions. In cases where the substrate loading is low, it may be beneficial to increase the search radius in the case of a blank position in order to increase the number of identified INPs. Figure 6 illustrates a result of the INP identification step with the SEM based on the corresponding FRIDGE picture with the grown ice crystals.

The total number of INPs that can be unambiguously attributed to an ice crystal origin is significantly influenced by the total wafer loading (INPs and non-INPs), which is determined by the sampling parameters (e.g., flow rate, sampling time, deposition efficiency) in combination with the aerosol concentration and aerosol properties. Even if the aerosol concentration is known, it is difficult to specify a suitable collection volume in advance due to the priori unknown fraction of INPs. Besides the total number of particles on the wafer, which is decisive for the chance to identify one particle in the defined radius and influences significantly on the ratio of falsely identified INPs (see modelling data in the supplement – Fig. S2), the ratio of potential INPs is also important to ensure that enough INPs are deposited on the wafer for suitable counting

statistics. Based on the modeling, we would consider a collection of around 100,000 particles on the wafer as a good number for meaningful measurements under typical free-tropospheric conditions. Of course, it has to be adapted to the actual conditions, e.g., when the fraction of INP is significantly different.

Therefore, we recommend to determine the particle concentration in the atmosphere in parallel to the collection and then perform a quick analysis of the wafers in FRIDGE to calculate the proportion of INPs in the total aerosol. Based on this proportion the sampling parameter can be adjusted to balance a sufficient number of INPs on the wafer but prevent an overload.

The variability of the aerosol concentration and the fraction of INPs lead to highly varying identification rates for individual samples, which is why it would be misleading to give an average identification rate for the method presented. The identification rate represents the proportion of ice crystal coordinates analyzed by SEM to which an unambiguous INP could be assigned. However, as an example, the values from the case study conducted at the high-altitude research station Jungfraujoch (Sect. 3) are discussed in the following. The average INP identification rate for the JFJ samples (based on 14 analyzed samples) was calculated to be 30 %. The large variation from 13 % to 50 % for the individual samples is an expected consequence of the different wafer loadings of the individual samples, as well as of the partly low counting statistics. In addition, the study identified the presence of multiple particles at 45 % of the locations, ranging from 7 % to 81 % for the individual samples. While a substantial number of multiple particle positions does result in a reduction of the absolute number of identified INPs, it exerts no influence on the frequency of individual particle classes or sizes. Although a correlation of multiple particle positions with the total number of particles on the substrate surface is suspected, such a correlation is not obtained directly, as the particle distribution on the wafer and the possible loss of volatile components (see Sect. 2.6) have an impact. The remaining 25 % (ranging in the extremes from 2 % to 66 %) were found to be blank positions, for the reasons discussed above. In case of a blank position due to an uncertain ice crystal position in connection with the search radius restriction, the overall result remains representative. No bias is expected in the relative contribution of individual particle classes. In the case of particle shift being the reason for the blank position, there is a potential for a bias in particle frequencies or size distribution, if the potential for a particle shift would correlate with particle class or size. Similarly, as volatile components and thin films cannot be detected in the electron microscope reliably (Sect. 2.6), a blank position might be detected here, although they could have triggered ice formation in FRIDGE.

Based on the uncertainties and assumptions discussed, many positions and potential INPs are excluded from further analysis. As a result, the number of identified INPs per sample is limited and typically low in contrast to the grown ice crystals in FRIDGE. These INPs, however, are accurately identified as we know that ice formation has taken place on the substrate at their position. This allows for conclusions on the analyzed INPs within the limits of their counting statistics and depending on the significance level, even for a small number of identified INPs. From the statistical calculation in Tab. S1 with a 95% confidence level, a limit of approximately 10 particles per group can be derived to make a reliable quantitative statement, for

groups with less particles the uncertainties become large. Further considerations of the limits of certain statements must depend on the particular statement and should employ common statistical approaches for counting statistics and compositional data.

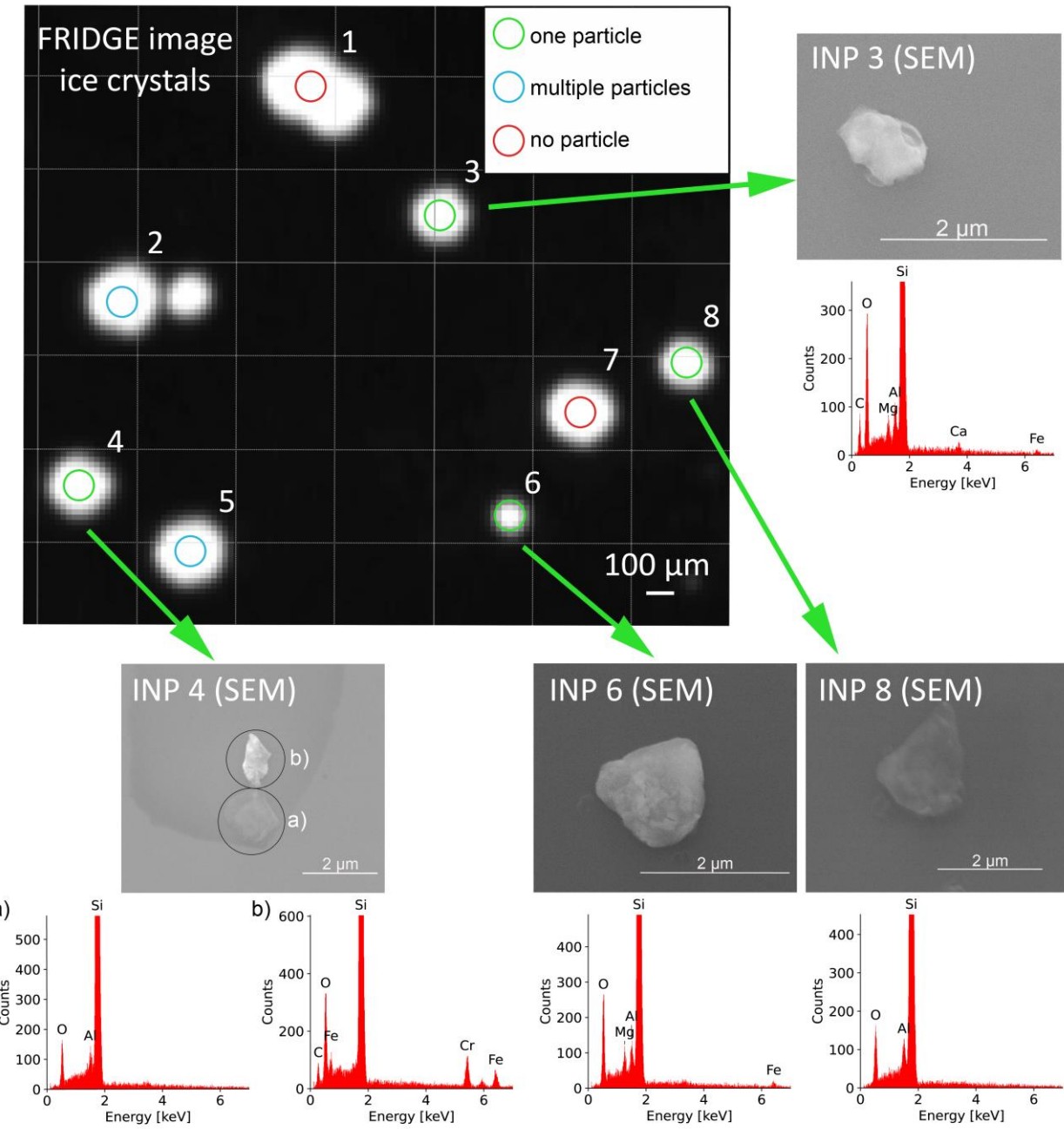

**Figure 6: FRIDGE picture (pixel size 20 × 20 μm) showing grown ice crystals including one coalesced crystal (top center). The corresponding search radius of 50 μm from SEM is shown by the colored circles: Blank positions are shown in red (position 1 & 7), multiple particle positions in blue (positions 2 & 5) and identified INPs in green (positions 3, 4, 6 & 8). However, one ice crystal was not detected by the ice crystal identification software. For the identified INPs, the corresponding SEM pictures and EDX spectra are shown. For one particle (bottom left), the BSE image indicates a different elemental composition, so two spectra (a & b) were recorded.**

## 2.6 Individual particle analysis

The analysis by SEM and EDX is an efficient method for characterizing INPs in detail, as it provides information on elemental composition and distribution as well as on morphology and surface properties. With this detailed information, it is possible, for example, to determine the mixing state of a particle (see Fig. 6 and Fig. 7). The morphological information can be used for source apportionment (e.g., for biological particles, soot, spherical particles from high temperature processes).

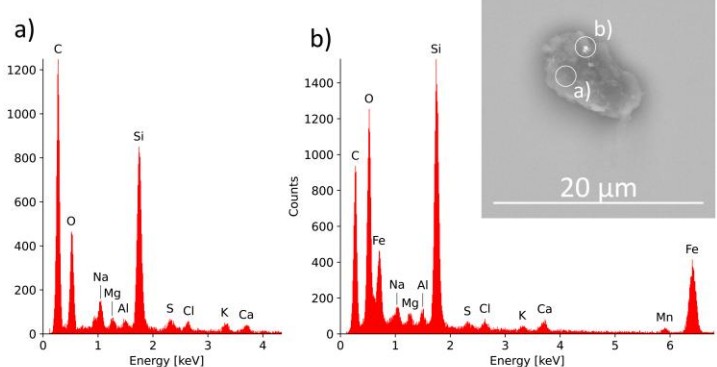

**Figure 7: EDX spectra and BSE image of a C-rich INP (a) with Fe-rich areas (b).**

The size of the INPs is determined in the last step of the coupling method from the SE/BSE pictures. The INPs have been processed in FRIDGE (multiple activation/evacuation cycles) and they were analyzed in a high vacuum under the electron beam. Therefore, the morphology of the particles may have undergone alterations. This may be especially the case for soluble/volatile components within a sample, which may evaporate during the analysis procedure. It can be assumed that these changes are of minor importance for most of the INP classes that can be determined using this method.

In the following section we define a classification scheme, which is mainly based on elemental composition (Fig. 8) and in some cases on the morphology of particles (Fig. 10). The subgroups defined in Fig. 8 can be summarized in three main groups (mineral particles, carbonaceous particles and other particles). It should be noted, however, that it is not possible to quantify the silicon content of a particle with this method. Given that we are working on a silicon substrate, a Si background signal in the resulting spectrum is always present. This may limit the chemical characterization of particles with a very small size, as their element signals with respect to the background signal may be insufficient. Instability with respect to the

electron beam also leads to limited detection of particle properties. In case of a main particle with small inclusions (as shown in Fig. 7 and Fig. 10d, for example) the composition of the main particle determines the classification. If a particle clearly consists of several individual components (see Fig. 6 INP4, for example), it will be classified as a mixture.

Based on the research question or the occurrence of specific particle classes at the sampling site, the classification scheme may be adapted.

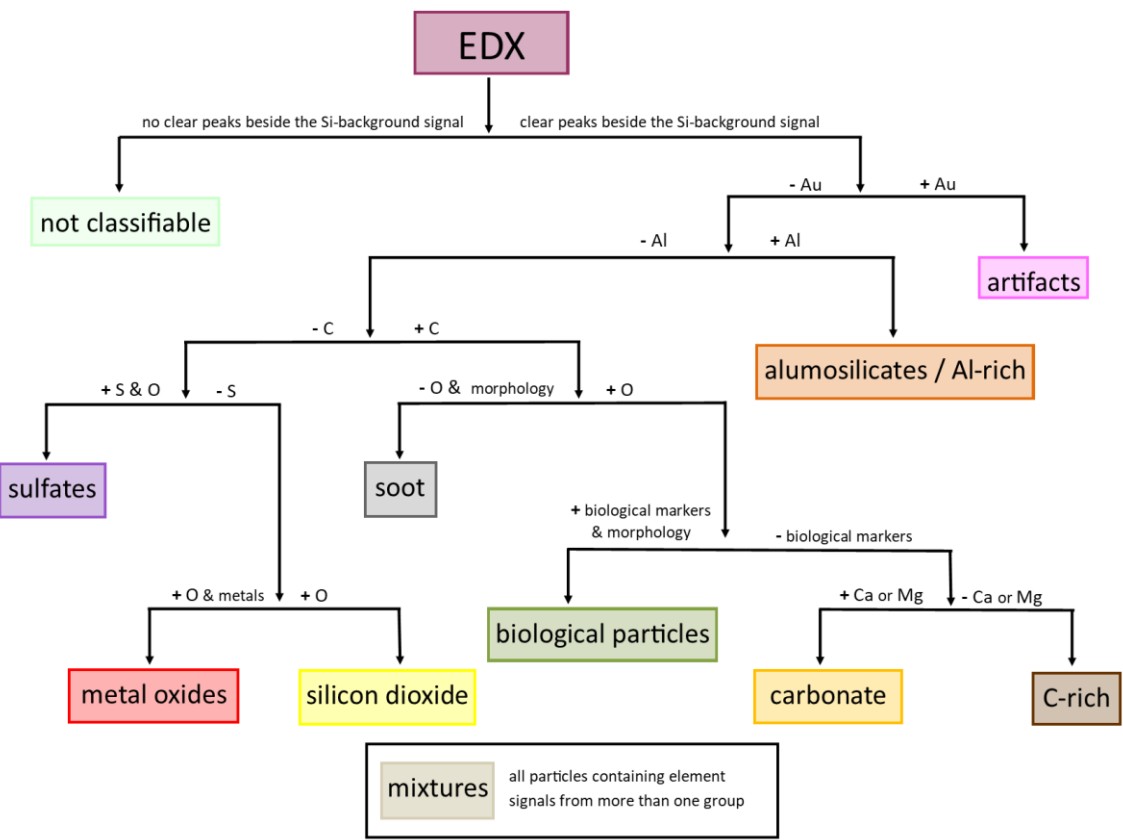

**Figure 8: INP classification scheme.**

*Mineral particles*

Aluminosilicates / Al-rich particles, carbonates and silicon dioxide are summarized as mineral particles. Typically, their irregular structure (Fig.7a-c) indicates a geogenic origin.

The *aluminosilicate / Al-rich* group is identified based on the Al signal and represents a combined group, as it is not possible to quantify the silicon content of a particle. The dominating source for aluminosilicates in the atmosphere is mineral dust from arid and semi-arid regions, while Al-dominated particles are usually rare (Kandler et al., 2007; Okada and Kai, 2004). In case of aluminosilicates, the particles may contain several minor elements (e.g., Na, Mg, K, Ca, Fe) in different ratios

(Fig. 9), depending on the minerals from which they originate. Their internal element distribution may be not homogeneous.

This method can therefore be used to estimate the abundance of individual mineral components in the INP fraction, potentially identifying an enrichment of highly effective ice nucleating particles (e.g., K-feldspar).

*Carbonates* can contain, in addition to carbon and oxygen, different counterions (e.g., calcium and/or magnesium), based on the mineralogical origin (e.g., calcite, dolomite).

*Silicon dioxide* is identified by the presence of only oxygen alongside - the not quantifiable - silicon. A distinction between

470 different particle sources can be made on the basis of their morphology. While most geogenic quartz particles show irregular shapes with typical sharp edges (Fig. 10c) (Whalley and Krinsley, 1974), anthropogenic $SiO_2$ particles from industrial high temperature processes show more spherical shapes. Fragments of the wafers can be clearly identified as artifacts by their sharp edges, glassy fracture and lack of oxygen signal.

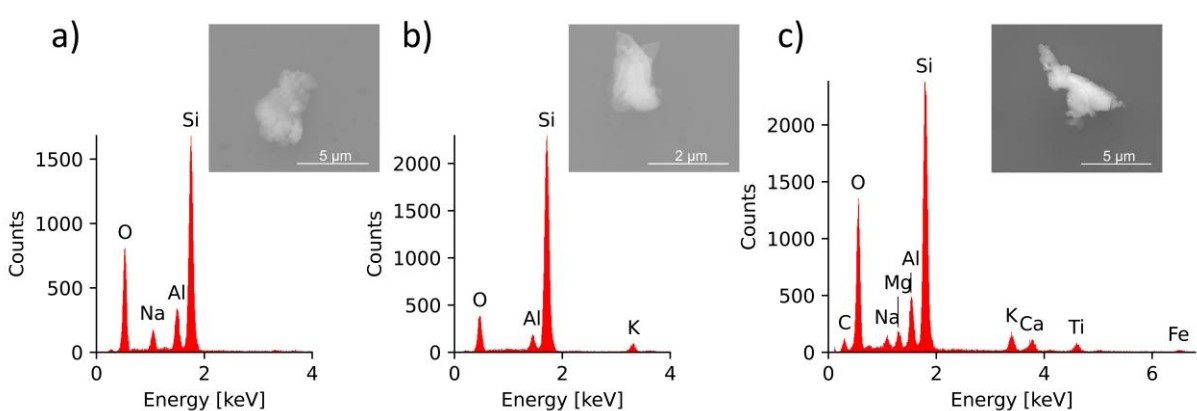

**Figure 9: EDX spectra and SE-images of various aluminosilicates: a) Na-containing aluminosilicate; b) K-containing aluminosilicate; c) complex aluminosilicate**

*Carbonaceous particles*

All particles with carbon as their main element (Fig. 10d-f) are combined as carbonaceous particles.

The group of *biological particles,* which includes plant debris, pollen, bacteria and fungal spores as well as their fragments, can be characterized by the presence of biogenic trace elements such as P, K, Mg, Ca, Na, etc. (Ebert et al., 2000) and by their characteristic morphology. Based on our criteria, particles are only classified as biological if the bulk particle fulfills

these criteria.

In many cases, *soot* particles can be clearly assigned based on their typical morphology (Fig. 10e), which often shows long chains or larger agglomerates of small, spherical primary particles (Sorensen and Feke, 1996). If it is not possible to characterize them by their morphology, they can be identified by their very low oxygen content compared to carbon.

The *C-rich group* contains all particles with high carbon, which cannot be clearly classified as biological particles or soot. It may also contain components from all other organic particles in the atmosphere, which can be analyzed with this method.

*Other particle classes*

All subgroups that cannot be clearly assigned to one of the two previous main groups are summarized as other particle classes.

The *metal oxides* are characterized by the presence of oxygen and a corresponding metal (except Al, which is assigned to the previous aluminosilicate / Al-rich particles group). Metal oxides can originate from geogenic minerals as well as from anthropogenic sources, that is why we refrain from clearly assigning this group to mineral particles. The morphology of these particles can be either irregular (e.g., natural mineral dust, anthropogenic urban dust), or spherical (Fig. 10h), with the latter possibly originating from high-temperature processes (e.g., coal combustion) or aircraft emissions.

*Sulfates* are mainly characterized by the presence of sulfur and oxygen. This method can detect only beam-stable sulfates reliably. Ammonium sulfate, for example, is not beam-stable during the analysis and therefore cannot be detected reliably. The morphology of these particles varies from clean crystallization to agglomerates and irregular shapes, depending on their source and formation processes. Besides the geogenic sources (minerals like gypsum or anhydrite), possible anthropogenic sources are industrial processes (mainly coal combustion), flue gas desulfurization and fertilizers. Due to the diversity of their possible sources and characteristics, we abstain from classifying them as minerals, although some of them may have a mineral origin.

All particles which are containing elements from more than one of the groups presented are assigned to *mixtures* (Fig7. i). Particles with gold deposits are classified as *artifacts* due to the use of gold wires as electrodes during the sampling process.

At this point, it should also be noted that our findings revealed an absence of small volatile compounds on the wafers in the EM, which are typically present in larger numbers in the total aerosol. Presumably, there is a loss of these components during sampling collection or processing. However, as these volatile particles are not known to be efficient INPs in the considered temperature range (Murray and Liu, 2022), it can be assumed that their absence does not significantly affect the results.

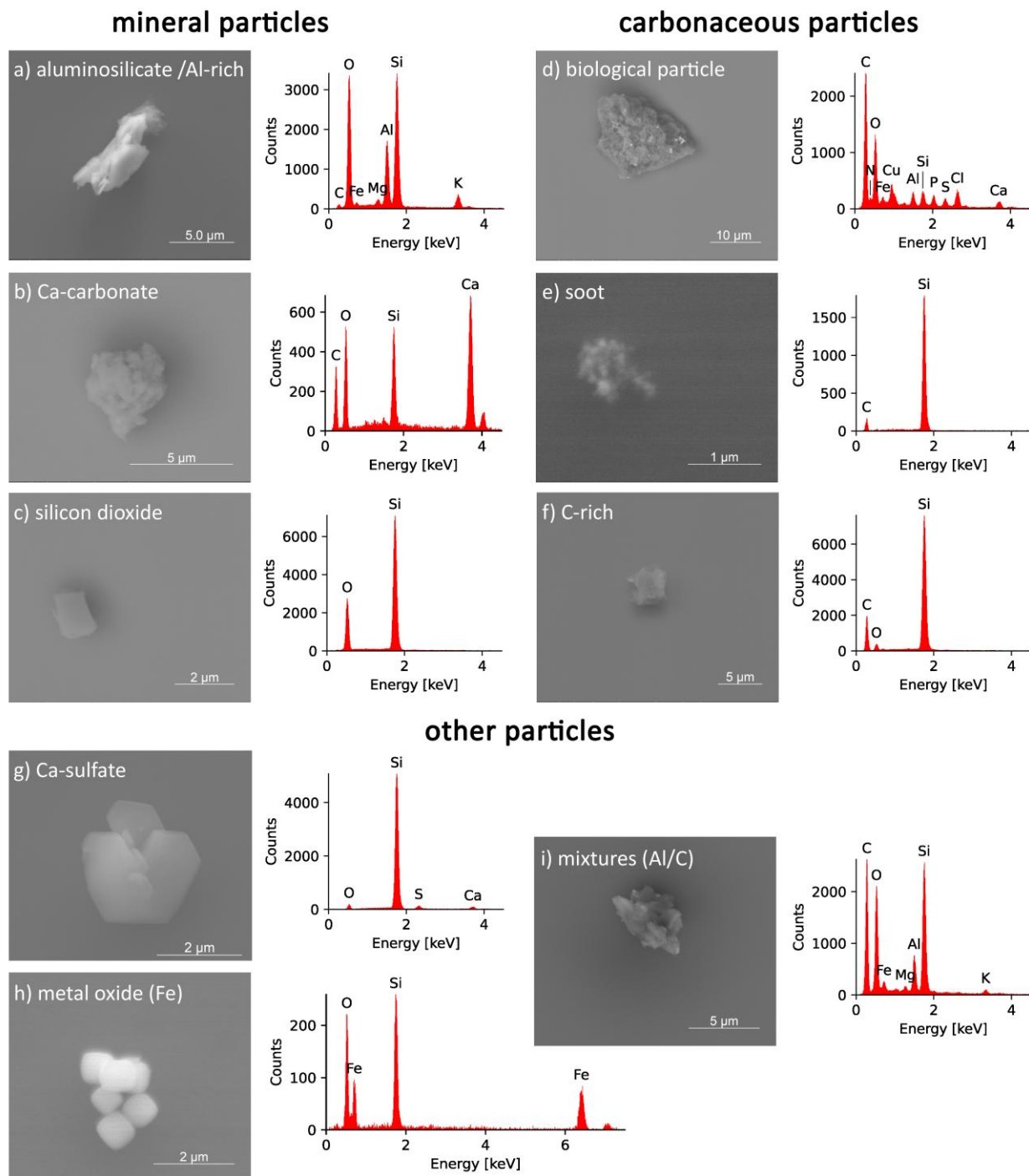

**Figure 10: Overview of representative EDX spectra and corresponding SEM images for the defined INP classes grouped as mineral particles (aluminosilicate / Al-rich (a), Ca-carbonate (b), silicon dioxide (c)), carbonaceous particles (biological particles (d), soot (e), C-rich (f)) and other particle classes (Ca-sulfate (g), metal oxide (h), mixtures (i)).**

## 3 Case Study: Results from the CLACE/INUIT campaign at the high-altitude research station Jungfraujoch in 2017

### 3.1 Sampling site

The high-altitude research station JFJ is located in the Swiss Alps at 3580 m above sea level between the mountain peaks of Mönch and Jungfrau. A general description of the station can be found in Bukowiecki et al. (2016). The samples were collected during the Cloud and Aerosol Characterization Experiment / Ice Nucleation Research Unit campaign (CLACE/INUIT 2017) between January 21 and February 25 2017. During winter, the station is 60 % of the time in the free troposphere (FT) (Herrmann et al., 2015), which enables characterization of the global background aerosol. A temporary influence of the planetary boundary layer is possible at any time of the year. According to Baltensperger et al. (1998) the station is in clouds (mixed-phase and ice) 40 % of the time. Since the average temperature did not fall below -15 °C during the sampling period, we assume that most INPs with activation temperatures of -20 °C to -30 °C were not activated under the prevailing environmental conditions. Even though some of them might have been activated previously in higher clouds, sampling under cloudy conditions likely does not introduce a large bias due to previously activated INPs. Aerosol sampling for the FRIDGE experiment was conducted downstream of the total inlet (Lacher et al., 2021).

### 3.2 INP concentration & sample selection

Since the INP concentration determined by FRIDGE for -20 °C and -25 °C was sometimes very low during the campaign (Fig. S3 in the supplement), we focus on the INPs activated at -30 °C for the single particle analysis. Their concentration varied between 0.1 and 1 stdL$^{-1}$ for most of the time. Towards the end of the campaign, a Saharan dust event (SDE) was identified, which resulted in an increase in INP concentration at both -25 °C and -30 °C. This was not the case for -20 °C, as Saharan dust particles primarily activate as INPs at temperatures below -20 °C (Murray et al., 2012; Niemand et al., 2012). Based on the experience that most ice nuclei active at warmer temperatures also activate at colder temperatures, it can be assumed that only a few INPs are neglected due to limiting the analysis to INPs activated at -30 °C.

A total of 14 substrates were selected from the larger set of samples obtained from the campaign for analysis using the coupling method presented herein. The particular samples were chosen based on their ice crystal abundance and homogeneous distribution on the substrate during the FRIDGE measurements. These samples are indicated by the corresponding sample number and triangles in Fig. S3. Overall, based on the parameters described in Sect. 2, we were able to clearly identify and characterize the associated INPs for 200 ice crystals, which corresponds to 30 % of the ice crystals positions analyzed (Fig. 12b). For the remaining 70 % we were unable to make a statement. While the multiple particle positions have no effect on the proportion of particle classes (Fig. 11) or size (Fig. 12a), the blank positions can cause a bias (discussed in Sect. 2.5.3). In this campaign, only a square with a side length of 2 cm in the center of the substrate was analyzed by electron microscopy. Since the area analyzed in SEM corresponds to roughly half of the area considered in

FRIDGE, the 200 INPs identified represent about 15% of the total sites activated in FRIDGE. This limitation has no influence on the individual chemical fractions or on the size distribution of the INPs, as the area was chosen arbitrarily and a homogeneous distribution of the particle groups on the wafer can be assumed.

Although the number of identified INPs appears comparatively low for a campaign period of five weeks, these INPs were identified with a high degree of reliability (Sect. 2.5.3). The small number of particles identified bears the risk that individual, time-limited variations occurring randomly during the sampling periods may influence the resulting total composition to a certain degree. It should therefore be noted that the results presented below may not comprehensively reflect the main composition of the INPs over the entire campaign period. Nevertheless, it can be shown that the method provides valid results for the main groups of INPs (see confidence intervals for Fig. 11 in the supplement (Tab. S1)).

## 3.3 chemical composition of the INPs

The 200 particles identified as INPs in the vicinity of the calculated coordinate were grouped into particle classes according to the classification scheme shown in Sect. 2.6. One particle with attached gold traces was classified as an artifact and therefore excluded from further discussions. The chemical composition of the remaining 199 INPs activated at -30 °C is shown in Fig. 11. Due to the limited number of identified INPs per sample (Fig. S4), mapping daily fluctuations is not possible for this campaign. Figure 11a) provides the chemical composition for all INPs sampled over the entire campaign period, within the restrictions mentioned in Sect. 3.2. Figures 11b) and 11c) illustrate the efficiency of the method in representing major INP-relevant trends. Despite the small number of INPs, the SDE can be clearly recognized by a different chemical composition of the INPs from this period (confidence intervals are given in the supplement (Tab. S1)).

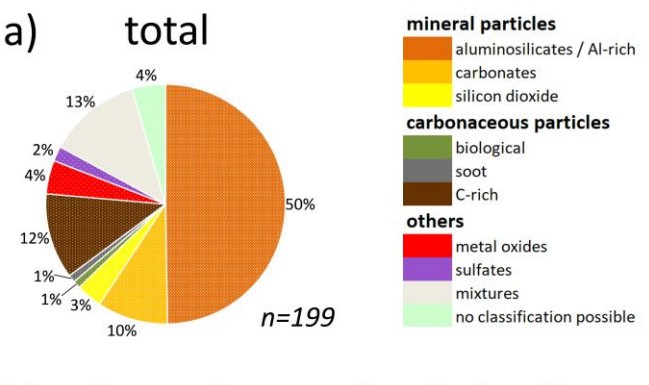

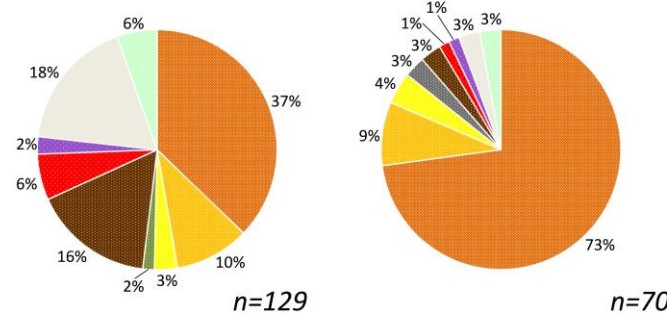

**Figure 11: Chemical INP composition with the number of analyzed particles (n) from CLACE/INUIT 2017 at the high-altitude research station Jungfraujoch (activated at T = -30 °C and RH = 99 / 101 %; RH = 95 / 97 % was chosen for one sample due to cluster formation at higher RH). a) total composition over the whole campaign period; b) composition prior to the Saharan dust event and c) during the Saharan dust event. For confidence intervals see supplement Tab. S1.**

*Mineral particles*

The analyzed INPs sampled at the high-altitude research station JFJ in January and February 2017 were found to be predominantly composed of mineral components (63 % in total). Their proportion is 50 % prior to the SDE, rising to 86 % during the SDE. Among these, the aluminosilicate / Al-rich group increases from 37 % to 73 %, whereas the carbonates and the silicon dioxide have almost the same proportion. Throughout the entire campaign period, aluminosilicates / Al-rich particles were most prevalent, compromising 50 % of the total identified particles. These particles were present in all individual samples, except for W10, which had only one identified INP in total. Carbonates were detected in 9 individual samples (total contribution of 10 %) with Ca as the main counter ion. *Silicon dioxide* particles contributed 3 % of all identified INPs, with mostly irregular shapes indicating a geogenic origin in three samples.

These findings are in good agreement with results reported from other INP/IR measurements at the same site. Eriksen Hammer et al. (2018) and Lacher et al. (2021) observed the presence of mineral particles in comparable quantities during the

same research campaign, despite analyzing IR activated between -10 °C and -18 °C. In previous campaigns Worringen et al. (2015) also identified terrigenous material as a significant contributor to ice nucleation at JFJ and Kamphus et al. (2010) observed a significant enrichment in minerals in IR compared to the total aerosol.

*Carbonaceous particles*

The carbon-dominated particles represented 14 % of the total INP composition during the CLACE/INUIT 2017 campaign. Only some single particles (1 % in each case) could be clearly assigned to a biological origin or soot. The remaining 12 % were classified as C-rich, and both biological material and soot, as well as any other carbonaceous particles existing in the atmosphere, may be included in this group. The fraction of carbonaceous particles decreased from 18 % before the SDE to only 6 % during the SDE.

Carbonaceous material was also identified as minor component in INPs/IRs at the high-altitude research station JFJ during winter 2013 by Worringen et al. (2015) as well as by Eriksen Hammer et al. (2018) and Lacher et al. (2021) for January and February 2017.

*Other particle classes*

During the CLACE / INUIT 2017 campaign metal oxides, which were primarily iron oxide, were found with a proportion of 4 % in total. A few particles within this group contained iron together with Ni and Cr as alloying elements that could be characteristic for steel. An anthropogenic origin or a local source from the station for these particles is assumed, but not confirmed. Apart from some single metal oxides with spherical shapes, most of them showed irregular shapes which may hint at a geogenic origin. Metallic particles and metal oxides have also been identified as a minor ice-forming compound by other studies conducted at the high-altitude research station JFJ (Eriksen Hammer et al., 2018; Kamphus et al., 2010; Lacher et al., 2021; Worringen et al., 2015). Ebert et al. (2011) also found metal oxides in their IRs and classified them primarily as iron oxide, which is consistent with our results.

Sulfates were rare (2 %) with Ca as the main counterion. Their morphology indicates a predominantly mineral origin.

The most abundant particle type of our mixture group features in addition to an Al peak, which is characteristic for the aluminosilicate / Al-rich group, also a distinct carbon peak (C/Al-ratio > 0.2). Most of these particles had a stronger Al-Peak with respect to the carbon peak (C/Al-ratio < 1). This Al/C-mixture may be an indication of soil dust, which contains carbonaceous material in addition to aluminosilicate / Al-rich minerals. In contrast to the aluminosilicate / Al-rich group these Al/C-mixed particles were found primarily in samples without influence of the SDE (18 % vs. 3 % during SDE), potentially pointing to a different origin of the air masses and thus a different type of mineral material (e.g., soil dust) which was transported to the station. Apart from this, the described composition can also be generated by mixing or coating with carbonaceous materials during particle aging in the atmosphere, in contrast to the freshly emitted Saharan dust. Such a mixed group at the high-altitude research station JFJ was also characterized by previous studies (Ebert et al., 2011; Worringen et

al., 2015). Lacher et al. (2021) reported also that many of their mineral dust particles from the CLACE/INUIT 2017 campaign showed signals of biological material, which may be equivalent to the mixed INPs in our study.

For 4 % of all particles which can be clearly identified as INPs based on their position, no chemical classification could be performed as outlined in Sect. 2.6.

It is generally difficult to make direct comparisons between the results of different INP/IR measurement techniques, as the results can vary significantly depending on the sampling configuration, ice nucleation activation conditions, and the classification schemes used for each instrumentation. Despite these constraints and with only a limited number of identified INPs we were able to demonstrate that our method provides reliable and valid results for the main particle groups relevant to ice nucleation by comparing our main results with other INP/IR measurements performed at the high-altitude research station JFJ.

## 3.4 Chemically-resolved INP size distribution

In addition to providing information on the concentration and elemental composition of INPs, our coupling method offers the significant advantage of providing an INP size distribution that can be coupled to the chemistry of individual INPs. The size of each identified INP is determined by calculating the projected area diameter ($d_{pa}$). Therefore, the particle is regarded as an ellipse and the dimensions of its major and minor axis are determined in order to subsequently calculate the diameter of an equivalent circle, which is referred to as $d_{pa}$.

Figure 12a) shows the chemically-resolved size distribution of all identified INPs from the CLACE/INUIT 2017 campaign at the high-altitude research station JFJ (confidence intervals for each size range are given in the supplement Tab. S2). The graph displays the absolute INP numbers within a specific size range (blue line), with the maximum found between 1 µm and 2 µm. All particles with $d_{pa} > 6$ µm were summarized due to statistical purposes. In comparison to the total aerosol size distribution from the whole campaign period (Weber, 2019), the maximum of the INP size distribution is significantly shifted towards larger diameters. We hypothesize that, in addition to the primary suitability of larger particles as ice nuclei, the absence of the small volatile aerosol components (nitrates, sulfates, and volatile organics) may play a role here (see Sect. 2.6). In an enrichment and depletion study, Eriksen Hammer et al. (2018) determined the depletion of the complex secondary aerosol in IRs compared to the total aerosol. This leads to the conclusion that the absence of these compounds does not significantly influence the results.

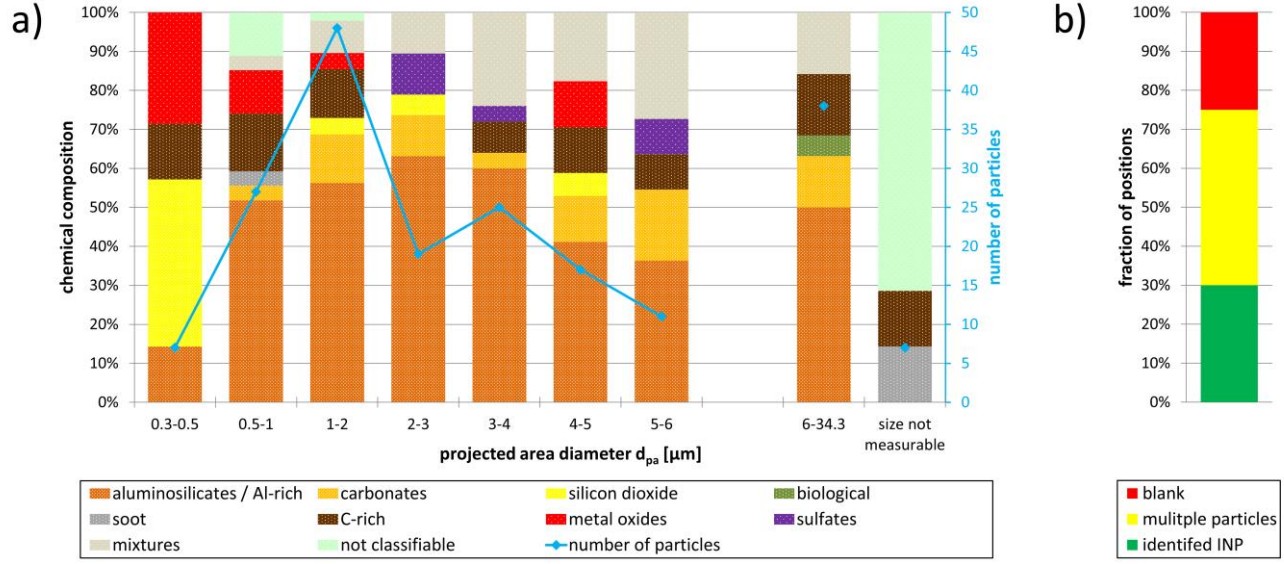

**Figure 12: a) Chemically resolved size distribution for all identified INPs (artifact excluded; n = 199) from CLACE/INUIT 2017 at the high-altitude research station Jungfraujoch (activated at T = -30 °C and RH = 99 / 101 %; RH = 95 / 97% was chosen for one sample due to cluster formation at higher RH). For confidence intervals see supplement Tab. S2.; b) The proportion of identified INPs (green), multiple particle positions (yellow) and blanks (red) compared to the total number of analyzed positions. The data shown in a) correspond to the 30% of identified INPs.**

As the mineral particles (aluminosilicates / Al-rich, carbonates and silicon dioxide) are in general the most prevalent group within the analyzed samples, they are also the most prevalent group in all size ranges, with proportions ranging from 50 % to 80 %. The proportions of the individual components vary for the different size ranges. However, the aluminosilicates / Al-rich particles represent the most abundant group among all size bins from $d_{pa} > 0.5$ µm.

Only 3 % of the analyzed INPs had a quantifiable diameter smaller than 0.5 µm. The smallest INP whose size could be determined with confidence was 300 nm, although we can generally also see smaller particles in the SEM. This agrees to well-established findings in the literature substantiating that most particles that act as effective ice nuclei are above a size of 500 nm (DeMott et al., 2010). Besides the mineral particles, which primarily consist of silicon dioxide, the smallest size bin shows the highest proportion of metal oxides.

Significantly more particles (14 %) were found in the size range between 0.5 µm and 1 µm. In addition to the mineral fraction, carbonaceous particles as well as metal oxides and mixtures can be assigned to this size bin. However, due to their submicron size, some particles could not be chemically classified. The size range of 1-2 µm contained the largest number of INPs (24 %), with a similar chemical composition to the previous size bin.

The number of particles decreases for INPs with $d_{pa} > 2\,\mu m$, which is consistent with a lower occurrence of these particles due to a reduced residence time in the free troposphere. The proportion of mixtures increases notably for INPs larger $2\,\mu m$. Additionally, C-rich particles were found in the range up to $6\,\mu m$, as well as all sulfates.

    Since the abundance of INPs with $d_{pa} > 6\,\mu m$ is low in the individual size ranges, they are summed up. The largest INP had a $d_{pa}$ of $34.3\,\mu m$. Apart from mineral particles, C-rich particles, and mixtures, the two biological particles were also found in

this size range. Due to losses by sedimentation, long-range transport of large particles is very unlikely. However, a local influence by air that is advected from the planetary boundary layer to the station cannot be excluded.

    It was not possible to determine the size of some particles, as their small size caused them to provide an insufficient image. This was the case for some carbonaceous particles, as well as for those particles for which a chemical classification was also not possible.

    Lacher et al. (2021) and Worringen et al. (2015) provide size distributions for INPs/IRs measured with different techniques at the high-altitude research station JFJ up to a size of $3\,\mu m$ and $5\,\mu m$, respectively. In both studies, the highest concentration was found for IRs smaller than $0.5\,\mu m$, but the broad maximum (diameters between $1.3\,\mu m$ and $5\,\mu m$) from Lacher et al. (2021) agrees reasonably well to our findings. The same is the case for particles collected with the Ice Selective Inlet by

Worringen et al. (2015), which also showed a secondary maximum at $1 - 1.5\,\mu m$. The shift towards larger particle diameters in our results in comparison to the maxima from Lacher et al. (2021) and Worringen et al. (2015) may be caused by the differences in sampling and ice activation. INPs in FRIDGE are activated through deposition nucleation / condensation freezing under defined conditions, while the IRs collected from ambient air are activated under natural and even more complex conditions, including the potentially more important immersion freezing mode (Ansmann et al., 2009; Murray et

al., 2012).

    The comparison of such INP size distributions with chemical information from different methods is difficult, since in addition to the influencing factors discussed in Sect. 3.3, a possible size selection or limitation of the sampling process, and different techniques of particle sizing may also play a role. Nevertheless, the results for our main groups are in reasonable agreement with the results from Worringen et al. (2015), although we were able to assign INPs to only 30% of the analyzed

positions (Fig. 12b). In our results, both the metal oxides and the few soot particles were observed at very small diameters, which is comparable to carbonaceous particles/soot and metal oxides predominantly detected in the submicron range by Worringen et al. (2015). Terrigenous particles, including silicates and Ca-rich particles, were primarily found in the larger size ranges, while our mineral components were distributed over all size ranges, with silicates domination for particles from $d_{pa} > 0.5\,\mu m$. In contrast to Worringen et al. (2015), our C-rich particles were present over the entire size range. The reason

for this is possibly that our classification scheme assigned the larger potentially biological particles as C-rich.

## 4 Summary and conclusions

A method for analyzing the concentration and individual physico-chemical properties of ambient INPs, which has been used in several campaigns (He et al., 2023; Schrod et al., 2020b), is discussed here from a methodological perspective. The method benefits from the coupling of two instruments already used for the analysis of INPs and IRs: the static diffusion chamber FRIDGE and the SEM. As the individual methods are already known, the focus here was on a description of the coupling and the associated advantages and uncertainties, as well as the resulting potential of the method.

Ambient atmospheric aerosol samples are collected on silicon wafer substrates using a simple electrostatic precipitator setup. Deposition nucleation and condensation freezing mode INPs are activated in the static diffusion chamber FRIDGE at various combinations of temperature and humidity and the resulting ice crystal growth is photographed. To link the FRIDGE measurements to the SEM analysis, it is important not to allow the ice crystals to grow too large, as this may cause problems determining the coordinates for the ice crystal origins. The ice crystal center points are located based on size and brightness thresholds using an image analysis software based on the pictures taken during the FRIDGE measurement. It has been shown, that our position identification algorithm works reasonably well, with a negligible number of incorrect positions due to condensation or coalesced ice crystals. Each ice crystal origin is assigned a coordinate based on a coordinate system previously defined by the engraved crosses on the wafer surface. These defined calibration points are used to transfer the identified ice crystal center point coordinates to the internal electron microscope coordinate system. Uncertainties for the center point identification (asymmetrical ice crystal growth and limited resolution of the FRIDGE images), and the calibration (different image quality and resolution of FRIDGE and SEM) were discussed, with the conclusion that it is necessary to consider not only the exact coordinate but also the surrounding area.

A model simulation was carried out to get insights in the effects of position uncertainty, INP fraction and the total number of particles on the wafer on the chance to identify the INPs in SEM. It was shown, that the position uncertainty and the total number of particles on the wafer have a significant influence on the identification of individual particles, while the fraction of INPs plays a minor roll. A search radius of 50 µm around the calculated coordinate was derived from model simulations and experimental values. This limitation may lead to the exclusion of potential INPs in the case of multiple particle positions or particles which are further away from the calculated coordinate due to e.g., particle shift or miscalculated ice crystal origins. At the same time, it increases the accuracy of the results, because we are analyzing only those particles which can be unambiguously associated with the origin of a real grown ice crystal. At each position where only one particle is found in the defined radius around the coordinate, comprehensive single particle analysis by SEM and EDX provides the elemental composition of the associated INP, as well as information on its size and morphology.

The number of ice crystals which can be assigned to an INP from the field data is highly variable and depends on the wafer loading. Nevertheless, it is not so easy to estimate suitable collection parameters in advance, as it is important to balance the number of INPs (which are typically quite rare in the atmosphere) and the total substrate loading. A higher substrate loading increases the probability of multiple particle positions. Although this reduces the absolute number of INPs identified, it has

no influence on the resulting distribution of the particle classes. In case of a blank position due to incorrect position identification or a not optimally selected search radius, the results are also not influenced. The situation may be different in the case of a particle shift or non-detectible volatile components and films as the reason for a blank as this may cause a bias in the resulting chemical composition and size distribution of the INPs.

Although the method has some drawbacks and uncertainties, it enables high accuracy in the identification and in this way physico-chemical characterization of individual INPs. This is, from our point of view, its significant strength compared to other INP/IR methods, which may have difficulties distinguishing between true INPs/IRs, additional collected particles, and sampling artifacts. In addition, this method can also determine ice activity for particles with a size of several micrometers, making it a useful complement to methods with size restrictions due to inlet systems or other factors. The detailed information on physico-chemical particle properties that can be obtained from SEM can be a valuable addition to pure INP counting methods for gaining information on the relevance of particle properties to ice nucleation efficiencies and could help to bridge the knowledge gap towards INP aerosol-type-specific parametrizations that could be used in modeling studies (Burrows et al., 2022).

The presented case study with samples from the CLACE/INUIT 2017 campaign at the high-altitude research station JFJ demonstrates that the method yields valuable results for the main INP classes, despite comparably low counting statistics. However, detailed statements about the minor INP classes are not possible for this study. Mineral components (aluminosilicates / Al-rich particles, carbonates and silicon dioxide), were the most prevalent as INPs in the predominantly free tropospheric air masses at the JFJ. They were distributed over the entire size range, except for silicon dioxide, which was mainly found in the size range below 500 nm. These particles originated mainly from non-local background dust sources - and in particular from a SDE, which can also be identified by a different chemical INP composition. Carbonaceous INPs of various sizes were found, including a minute amount of small soot particles as well as large biological particles. In addition, a small amount of metal oxides, mostly iron oxide, were also identified, primarily with $d_{pa} < 0.5$ µm. Sulfates were rare. Mixed particles, predominantly aluminosilicate / Al-rich particles with increased carbon content, were more common at larger diameters with a higher proportion in air masses prior to the SDE.

Future studies may relate the composition and sizes of INPs to the different activation conditions. This may help to identify which particle types and features are atmospherically most relevant as a function of temperature. The comprehensive single particle information provides also the opportunity to study for example the potential enrichment of high-effective ice nucleating particles (e.g., K-rich feldspar) by comparing the activated particles to the total wafer loading. However, due to the relatively low counting statistics, further improvements are necessary to obtain a more reliable insight into the relevance of the particle properties for the INP activation. Following the uncertainty analysis presented here, we identified the FRIDGE camera resolution as one important area of improvement. A higher camera resolution to document the ice crystal

growth in FRIDGE would improve the accuracy of the ice crystal center point identification and make it easier to find the calibration point in the SEM. This could significantly reduce a substantial part of the uncertainties.

As experimental knowledge about concentration and composition of INPs, and their contribution to upper tropospheric ice nucleation processes in cirrus cloud formation is severely lacking (Kanji et al., 2017), the method will be adapted for future aircraft campaigns.

## Data availability

The complete data set is available for the community and can be accessed by request to Martin Ebert ([mebert@geo.tu-darmstadt.de](mailto:mebert@geo.tu-darmstadt.de)) of the Technical University Darmstadt.

## Author contributions

The concept of the study was designed by JC, ME and HB. The samples at the high-altitude research station Jungfraujoch were collected by DW, who also performed the FRIDGE measurements and interpreted the INP concentration data with HB, JC and JS. The scanning electron microscopy analysis was performed by LS, the interpretation was done with support from ME. KK wrote the program to perform the ice crystal center determination and coordinate conversion and conducted the model simulation. The general method evaluation was done by LS, KK, HB, JS, ME. The paper concept was drafted and written by LS, with contributions from JS, DW, HB, JC, ME and KK.

## Competing interests

One Co-author is member of the editorial board of the journal Atmospheric Measurement Techniques.

## Acknowledgements

This work was funded by the Deutsche Forschungsgemeinschaft (DFG, German Research Foundation) – TRR 301 – Project-ID 428312742 and by the Ice nucleating research unit INUIT (FOR 1525 (project ID 170852269)).

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
