# Peer review of "Analyzing the chemical composition, morphology and size of icenucleating particles by coupling a scanning electron microscope to an offline diffusion chamber"

_EGUsphere, 2024_

## Referee Comment (RC1)

**Review of "Analyzing the chemical composition, morphology and size of ice nucleating particles by coupling a scanning electron microscope to an offline diffusion chamber"**

**General comments**

In this study, the authors present an offline method for characterizing ice nucleating particles (INPs) by coupling an offline diffusion chamber, the FRankfurt Ice nucleation Deposition freezinG Experiment (FRIDGE), with a scanning electron microscope (SEM). The approach is potentially useful to provide comprehensive information of INPs, including their chemical composition, morphology, and size, and to improve the parameterization in the corresponding models. However, the advantages of combining the two instruments are not well demonstrated from the case study at JFJ. The FRIDGE-SEM coupling technique is not new and no convincingly new findings are shown in this study compared to the former studies at the JFJ. The methodology needs more details as intended to, in particular the coordinate system that allows for recovery of the particles, which is critical to determine whether particles are original or processed INPs. Statistical significance is also a big concern. It is unlikely to use the data present in this study to evaluate the INP-type-specific parameterizations in the model. In addition, the manuscript is not very well structured. The potential contribution of this study is within the scope of AMT. However, the current manuscript is behind publication quality due to the reasons as mentioned above. Therefore, I recommend that substantial revision needs to be done before considering publication.

**Major Comments**

The novelty of this study is supposed to be the technical details and direct measurement of INPs. However, it needs more details to convince the readers that the individual particles can be fully recovered after measurement cycle of the FRIDGE. Additional controlled experiments may need to be done and shown to prove that hypothesis. Otherwise, the SEM measurement would be on the IRs rather than original INPs.

It is good that comparisons with previous studies at JFJ are discussed. However, the authors may want to add more discussion on the differences and emphasize the new findings.

The results were based on 200 individual particles from 5-week measurement, and only half of the total analyzable area of each wafer was analyzed by SEM. The author should evaluate the statistical significance, discuss more about the representativity, uncertainty and limitations.

The structure of this manuscript needs revision. The introduction, method and results sections have several overlaps, which need to be improved. E.g., in the introduction section, L 126 to 130 belongs to method part; Section 2.6 Chemical classification should be merged into Section 3.4 INP chemistry or go to supporting information; Section 3.1 Sampling site needs to go to method and/or introduction; Section 3.3 Method evaluation needs to go to method section.

**Minor Comments:**

L33: Please remove "e.g." from the citation and do it through the entire manuscript.

L41-42: What is the temperature range you mentioned? Please add the values and explain why the main focus has been on such range.

L44: Quantify the "small fraction". How much%?

L46 & 51: Change "history within the atmosphere" to "atmospheric processing"

L60: Mineral dust is a good INP, rather than an important factor in ice nucleation.

L76-93: Please consider remove or shorten this part. Lab, field and model studies are the main approaches for atmospheric science. This is very basic and general, not specific for ice nucleation studies.

L103-104: The sentence is unclear. Please revise.

L105-106: It is unclear why the IR and INP are in the brackets. Please add more descriptions.

L111: What are the problems? Please elaborate.

L116-122: You might want to emphasize more on the advantages of EM over online measurement technique. For example, the morphology and quantitative results that SPMS generally does not provide without reference instrument. Note that SPMS can provide information of mixing state, which is not only obtained by the EM.

L131: I am not convinced that the particles characterized by SEM are INPs rather than IRs.

L133-148 Figure 1: It is unclear regarding the second and the third pictures, please consider adding B1, B2 and B3 for second and third pictures regarding the FRIDGE, C1, C2 and C3 for SEM. Please add the missing legends for the bars and pie chat. Please add missing units for the x-axes on the EDX spectra and the time series of INP conc., respectively. Please increase the resolution of the pictures, especially the scales and labels which are too vague for current version.

L174: Change "at" to "in".

L205: Will measurement cycle change e.g., morphology or properties of particles? If no, you need to give the proof.

L204: Change "electron microscopy" to "EM". The full name only needs to be mentioned when shown for the first time in the manuscript (from introduction section).

L223: What is ImageJ? Please explain.

L231: Consider changing "time-consuming" to "labour-intensive" or other more appropriate word. Please revise and give the approximation of the duration of such analysis.

L247: Give full name for "FEI"

L254: Remove the full name of SEM.

L267: How much is the uncertainty? Please give the value.

L294: How much smaller? Please give the value.

L302: "appearance", do you mean morphology?

L305: Consider changing "excellent" to "efficient"

L374 Figure3: The scales and labels are too vague. Please increase the resolutions.

L437-438: Please separate the samples into cloudy and clear sky cases and discuss accordingly.

L440-444 Figure 6: Cloudy and clear sky cases should be separated.

L634-636: Consider archiving data in a reliable public data repository.

---

## Author Comment (AC1)

**Author response to Reviewer 1**

*First of all, we would like to thank the reviewer for reading and commenting on our manuscript. The comments and remarks helped us a lot to focus more on the methodological aspects and to improve the structure of the manuscript.*

*In the following, the reviewer comments are written in bold and our answers in italics. Text passages from the revised manuscript are in quotation marks, modified or newly added passages are marked in green.*

**General comments**

**In this study, the authors present an offline method for characterizing ice nucleating particles (INPs) by coupling an offline diffusion chamber, the FRankfurt Ice nucleation Deposition freezinG Experiment (FRIDGE), with a scanning electron microscope (SEM). The approach is potentially useful to provide comprehensive information of INPs, including their chemical composition, morphology, and size, and to improve the parameterization in the corresponding models. However, the advantages of combining the two instruments are not well demonstrated from the case study at JFJ. The FRIDGE-SEM coupling technique is not new and no convincingly new findings are shown in this study compared to the former studies at the JFJ. The methodology needs more details as intended to, in particular the coordinate system that allows for recovery of the particles, which is critical to determine whether particles are original or processed INPs. Statistical significance is also a big concern. It is unlikely to use the data present in this study to evaluate the INP-type-specific parameterizations in the model. In addition, the manuscript is not very well structured. The potential contribution of this study is within the scope of AMT. However, the current manuscript is behind publication quality due to the reasons as mentioned above. Therefore, I recommend that substantial revision needs to be done before considering publication.**

*We have revised the entire manuscript (see our responses to the major and minor comments below as well as our responses to the other two reviewers for details). Structural changes have been made, to focus more on the method. The entire method section has been revised to make it less manual-like, as it was suggested by the reviewers. The case study discussion has been shortened, in favor of a more concise and focused methodological manuscript better fitting the scope of AMT.*

*The coupling of FRIDGE and the SEM is now described in more detail and in a more structured way in the methods section, where advantages, drawbacks and uncertainties are discussed and a general overview of the potential of the method is provided.*

*The case study in this manuscript is intended to illustrate that the method presented provides reliable results for the main INP groups, despite relatively low identification rates, and not primarily to present new atmospherically relevant results. The discussion of the results has been adapted to emphasize comparability with other studies. Despite the small number of analyzed INPs, we have decided to show a comparison of air masses before and during a Saharan dust event, as the potential of the unambiguous particle assignment to the corresponding ice crystals becomes obvious here.*

*We do not claim that the presented results of our case study are sufficient to be used for model evaluation, we just want to express that this type of results could be used for a closure study as suggested to Burrows et al. (2022).*

**Major Comments**

**The novelty of this study is supposed to be the technical details and direct measurement of INPs. However, it needs more details to convince the readers that the individual particles can be fully recovered after measurement cycle of the FRIDGE. Additional controlled experiments may need to be done and shown to prove that hypothesis. Otherwise, the SEM measurement would be on the IRs rather than original INPs.**

*Based on the definition by Cziczo et al. (2017) the particles activated in FRIDGE are INPs because they were activated under defined conditions after the collection of the total aerosol. In the electron microscope, we are able to locate and analyze these INPs at the positions of crystal origin.*
*Of course, in this context one could also speak of ice residuals (IR), since the particles have already been processed in FRIDGE. IRs are defined as particles remaining after the collection of atmospheric ice crystals and subsequent evaporation of the ice phase. In this case, there is a risk that additional particles deposited on the surface of the original ice crystal are subsequently identified as ice forming particles. This problem does not exist in our method. This is why we refer to the analyzed particles as INPs.*

"*Based on the definition by Cziczo et al. (2017) we refer to the identified particles as INP, as they were activated under defined conditions after the collection of the total aerosol and not sampled as ice crystals. Therefore, we are able to investigate truly activated particles in contrast to methods analyzing IRs, which face challenges in order to distinguish between IRs and scavenged particles. However, some of the INPs analyzed with SEM may have undergone changes (see Sect. 2.6) due to the measurement procedure in FRIDGE, but we assume that these changes are of minor importance for the main INP classes that we can analyze with this method.*"

**It is good that comparisons with previous studies at JFJ are discussed. However, the authors may want to add more discussion on the differences and emphasize the new findings.**

*In our opinion, a case study in a methodological paper should exemplarily showcase the capabilities of a measurement system by presenting plausible results that, in the best case, agree to previously published literature. As shown in the revised manuscript the results indeed are in good agreement to those from Eriksen Hammer et al. (2018) and Lacher et al. (2021) that were obtained from the same campaign, as well as to those of Worringen et al. (2015), Kamphus et al. (2010) and Ebert et al. (2011) measured at JFJ but a few years prior. For a paper that focuses on atmospheric measurements and their implications, we agree with the reviewer that it is important to discuss differences and, above all, to focus on new findings, however, we think that such an in-depth discussion would be beyond the scope of an AMT paper. In fact, to strengthen the methodological focus of the paper, we have streamlined the case study section.*

**The results were based on 200 individual particles from 5-week measurement, and only half of the total analyzable area of each wafer was analyzed by SEM. The author should evaluate the statistical significance, discuss more about the representativity, uncertainty and limitations.**

*Of course, 200 INPs for a period of more than 5 weeks is comparatively low. Therefore, we do not claim that our results are representative of an average composition of INPs over the entire campaign period. This is now also noted in the manuscript.*

"*Although the number of identified INPs appears comparatively low for a campaign period of five weeks, these INPs were identified with a high degree of reliability (Sect. 2.5.2). The small number of particles identified bears the risk that individual, time-limited variations occurring randomly during the sampling periods may influence the resulting total composition to a certain degree. It should*

*therefore be noted that the results presented below may not comprehensively reflect the main composition of the INPs over the entire campaign period. Nevertheless, it can be shown that the method provides valid results for the main groups of INPs (see confidence intervals for Fig. 9 in the supplement (Tab. S2))."*

However, by comparing air masses before and during a Sahara dust event (see Fig. 9 in the revised manuscript), we were able to show that it is possible to identify important trends in INP-relevant groups. This illustrates the great strength of our method, which has low identification rates, but identifies the INPs with high accuracy and thus still delivers credible results. Confidence intervals for the chemical distribution are given in the revised supplement (Tab. S2).

**The structure of this manuscript needs revision. The introduction, method and results sections have several overlaps, which need to be improved. E.g., in the introduction section, L 126 to 130 belongs to method part; Section 2.6 Chemical classification should be merged into Section 3.4 INP chemistry or go to supporting information; Section 3.1 Sampling site needs to go to method and/or introduction; Section 3.3 Method evaluation needs to go to method section.**

*We have revised the structure of the paper and created a distinct separation between the general methods section and the results of the case study. With regard to the points mentioned, we have proceeded as follows:*

*L126 to L130 from the introduction section was moved to the method part as an introduction.*

*Section 2.6: We have decided to leave this section (renamed as "individual particle analysis") in the methods section and have adapted it accordingly. It no longer describes the specific particle classes found at the JFJ but defines a general classification scheme and discusses strengths and limitations of the individual particle analysis.*

*Section 3.1: For a paper focusing on the results, we would agree with the reviewer. But in our opinion, in this case, the sampling site description belongs to the case study, because the method part describes and discusses the coupling procedure and the sampling site is part of the case study.*

*Section 3.3: As the identification rates are highly dependent on the total wafer loading, which is influenced by the aerosol concentration and activated fraction during the sampling time, the values for the JFJ campaign are not universally valid. We have therefore not included these values in the methods section. However, we understand the reviewer's point, which is why we have decided to include the identification rates with the corresponding restriction in the method section (Section 2.5.2 "INP identification").*

*"The number of INPs that can be unambiguously attributed to an ice crystal origin is significantly influenced by the total wafer loading, which is determined by the sampling parameters (e.g., flow rate, sampling time, deposition efficiency) in combination with the aerosol concentration present. However, even if the aerosol concentration is known, it is difficult to specify a suitable collection volume in advance, as the ratio of potential INPs to the total aerosol also plays a role. This ratio is variable and usually unknown prior to measurement. As a result, the amount of atmospheric aerosol and the proportion of INPs deposited on a wafer are highly variable. This variability is also seen in the identification rates, which is why it would be misleading to give an average identification rate for the method presented. However, a specific identification rate for the case study conducted at the high-altitude research station Jungfraujoch (Sect. 3) can be given here as a guideline. The average INP identification rate was calculated to be 30% (ranging from 13% to 50%). Furthermore, the study identified the presence of multiple particles at 45% of the locations (ranging from 7% to 81%), while the remaining 25% (ranging from 2% to 66%) were found to be blank positions."*

*The comparison between the different ice crystal counting methods (formerly Fig. 5) was also moved to the method section (Section 2.4 "identification of ice crystal positions") and replaces the graphical representation of coordinate determination (formerly Fig. 2), which has been moved to the supplement.*

**Minor Comments:**

**L33: Please remove "e.g." from the citation and do it through the entire manuscript.**

*We have removed the "e.g.," in many cases. In some cases, however, we have left it to indicate that the reference cited is just an example.*

**L41-42: What is the temperature range you mentioned? Please add the values and explain why the main focus has been on such range.**

*The temperature range for heterogeneous nucleation in mixed phase clouds is 0°C > T > -38°C. At temperatures colder than -38°C spontaneously homogeneous freezing might dominate the formation of ice. In the case of homogeneous freezing no INPs are needed to exceed the energy barrier. That is why the main research focus for INPs has been on the temperature range between 0°C and -38°C.*

*However, these sentences have been deleted in favor of a more focused introduction with less of a review character.*

**L44: Quantify the "small fraction". How much%?**

*The sentence has been adapted and now reads:*

*"Only a small fraction of the total aerosol can act as INPs and their concentrations can show variations of several orders of magnitude in space and time (DeMott et al., 2010; Kanji et al., 2017)."* *Values for the proportion of INPs in the total aerosol are given in the two references.*

*However, later in the Introduction (L96-97) we give values:*

*"… extremely low number of INPs within a sampled air volume compared to the much higher number of non-INP particles (ratio ~ $1/10^4 - 1/10^6$),…"*

**L46 & 51: Change "history within the atmosphere" to "atmospheric processing"**

*Changed as requested. The entire section has also been shortened. The new sentence now reads as follows:*

*"In addition to the prevailing environmental conditions i.e., temperature and humidity, the potential for an INP to become activated is dependent upon individual particle properties (surface imperfections (Kiselev et al., 2016), chemical composition and specific chemical properties, crystal structure, coating (Kanji et al., 2008), etc.) as well as its atmospheric processing including potential agglomeration or pre-activation (Marcolli, 2017)."*

**L60: Mineral dust is a good INP, rather than an important factor in ice nucleation.**

*Changed as requested.*

*"Mineral dust, which is emitted from arid and semi-arid regions and is globally distributed in the atmosphere (Perry et al., 1997; Ansmann et al., 2003; Schepanski et al., 2018), is a good INP at temperatures below -15°C (Hoose & Möhler, 2012)."*

**L76-93: Please consider remove or shorten this part. Lab, field and model studies are the main approaches for atmospheric science. This is very basic and general, not specific for ice nucleation studies.**

*We have removed the entire section. The references for laboratory and field experiments have been moved to the following section.*

*"Although there is a variety of methods to determine the INP concentration in the laboratory (Hoose & Möhler, 2012; Hiranuma et al., 2015; DeMott et al., 2018; Hiranuma et al., 2019) and in the field (Wex et al., 2019; Schrod et al., 2020b; Brasseur et al., 2022; Lacher et al., 2024), only a few of them are simultaneously able to report on the chemical characteristics of individual ice-nucleating particles."*

**L103-104: The sentence is unclear. Please revise.**

*Revised sentence: "In another approach, the particles are activated under defined conditions in an online reaction chamber (e.g., Rogers, 1988) after the collection of the total aerosol."*

**L105-106: It is unclear why the IR and INP are in the brackets. Please add more descriptions.**

*The entire section on the collection of INPs/IRs has been rewritten and shortened. As a result, the entire definition of INPs and IRs (including this sentence) has been removed.*

**L111: What are the problems? Please elaborate.**

*Online systems may have problems to resolve low concentrations of INPs as they have typically a lower sampling flow, resulting in low INP counts in the range of background counts. Offline methods, on the other hand, can enrich the number of INPs on the sample substrates due to higher sampling flows and longer collection times.*

**L116-122: You might want to emphasize more on the advantages of EM over online measurement technique. For example, the morphology and quantitative results that SPMS generally does not provide without reference instrument. Note that SPMS can provide information of mixing state, which is not only obtained by the EM.**

*We removed the mention of the mixing state in the section about EM.*

**L131: I am not convinced that the particles characterized by SEM are INPs rather than IRs.**

*The particles originally collected and activated in FRIDGE are INPs, since they were collected as total aerosol and not as ice crystals. The processing in FRDIGE (repeated activation/evaporation) can undoubtedly lead to changes in particle properties in some cases, which is why one could also speak of IRs in EM. However, according to our understanding, these changes are more likely to affect the soluble/volatile components, which are not known to contribute significantly to ice nucleation in the considered temperature range. Since our method does not allow us to detect the small volatile components well, we assume that the majority of particles which can be analyzed with our method do not undergo serious changes during the activation/evaporation processes. Therefore, we stick to the definition of Cziczo et al. 2017, according to which we collected and activated INPs. In addition, the coupling allows for investigating truly activated particles in contrast to methods analyzing IRs, which face challenges in order to distinguish between IRs and scavenged particles.*

**L133-148 Figure 1: It is unclear regarding the second and the third pictures, please consider adding B1, B2 and B3 for second and third pictures regarding the FRIDGE, C1, C2 and C3 for SEM. Please add the missing legends for the bars and pie chat. Please add missing units for the x-axes on the**

**EDX spectra and the time series of INP conc., respectively. Please increase the resolution of the pictures, especially the scales and labels which are too vague for current version.**

*The desired labels (B1, B2 and B3 for the FRIDGE as well as C1, C2 and C3 for SEM) and the missing labels for the pie chart and the bars have been inserted. The labels for the pie chart and the bars have been kept general (TA, TB, TC and Class A, Class B, …) to make it clear that this is only a sketch of the method and that no real results are shown. The missing units have also been added and the scales and the font size of the labels have been increased.*

**L174: Change "at" to "in".**

*Changed as requested.*

**L205: Will measurement cycle change e.g., morphology or properties of particles? If no, you need to give the proof.**

*Please, see comment on INP discussion above. Of course, the morphology, especially for soluble particles can change during the activation/evaporation process. Since most of the particles we identified as INPs consist of non-soluble components, we believe that this does not significantly affect the results.*

**L204: Change "electron microscopy" to "EM". The full name only needs to be mentioned when shown for the first time in the manuscript (from introduction section).**

*Changed as requested.*

**L223: What is ImageJ? Please explain.**

*ImageJ is a free image processing software. We added this information to the text.*

*"The ice crystal positions are identified by image analysis using the internal particle analyzer of the free image processing software ImageJ (Schneider et al., 2012), with a minimum size of 30 pixels proven to be useful."*

**L231: Consider changing "time-consuming" to "labour-intensive" or other more appropriate word. Please revise and give the approximation of the duration of such analysis.**

*The corresponding section has been rewritten for the purpose of a less manual-like writing style, as it was requested by the reviewers. The wording is no longer used in the new section.*

**L247: Give full name for "FEI"**

*FEI is the official company name. It stands for "Field Electron and Ion Company".*

*"A Quanta 200 FEG Environmental Scanning Electron Microscope (ESEM) by FEI (Field Electron and Ion Company; Eindhoven, Netherlands) coupled to an energy dispersive X-ray detector (EDX) (EDAX, AMETEK, Tilburg, Netherlands) was used for analysis."*

**L254: Remove the full name of SEM.**

*Changed as requested.*

**L267: How much is the uncertainty? Please give the value.**

*This uncertainty is based on the calculation of the ice crystal origin and the coordinate calibration. We added a comment on the uncertainty for the ice crystal calculation in Section 2.4.*

*"Nevertheless, a potentially imperfect radial symmetry of the ice crystal growth, coupled with the restricted resolution of the FRIDGE images (20 x 20 µm), may result in an uncertainty in the calculation of the ice crystal origin. As the size of an ice crystal increases, the probability and extent of such a non-symmetrical growth also increases. The quantification of this uncertainty proved to be difficult, as it depends on the symmetry deviation present. To reduce this uncertainty based on an imperfect radial symmetry, the ice crystal position calculation should be performed on the basis of FRIDGE images, that show the ice crystals in a state close to activation."*

*The coordinate calibration uncertainty is mentioned in Section 2.5.1.*

*"Due to the limited resolution of the FRIDGE images of about 20 x 20 µm, the calibration has of course an uncertainty in the same order of magnitude."*

*Regarding the accuracy of the coordinate, the following is stated in the INP identification section (Section 2.5.2):*

*"Each ice crystal position, based on a real grown ice crystal, is inspected by SEM to identify the presence of particles. Given the uncertainties associated with the ice crystal identification process (Sect. 2.4) and the coordinate calibration (Sect. 2.5.1), it is crucial to consider not only the exact calculated coordinate but also the surrounding area. This area must take into account the aforementioned uncertainties and, at the same time, limit the probability that several particles will be observed in the scanned area. In this context, a radius of 50 µm has proven to be useful. While the previously discussed uncertainties may suggest a lager radius to be beneficial, in fact, the high substrate loading often proves to be the limiting factor."*

**L294: How much smaller? Please give the value.**

*The section has been rewritten:*

*"The number of INPs that can be unambiguously attributed to an ice crystal origin is significantly influenced by the total wafer loading, which is determined by the sampling parameters (e.g., flow rate, sampling time, deposition efficiency) in combination with the aerosol concentration present. However, even if the aerosol concentration is known, it is difficult to specify a suitable collection volume in advance, as the ratio of potential INPs to the total aerosol also plays a role. This ratio is variable and usually unknown prior to measurement. As a result, the amount of atmospheric aerosol and the proportion of INPs deposited on a wafer are highly variable. This variability is also seen in the identification rates, which is why it would be misleading to give an average identification rate for the method presented. However, a specific identification rate for the case study conducted at the high-altitude research station Jungfraujoch (Sect. 3) can be given here as a guideline. The average INP identification rate was calculated to be 30% (ranging from 13% to 50%). Furthermore, the study identified the presence of multiple particles at 45% of the locations (ranging from 7% to 81%), while the remaining 25% (ranging from 2% to 66%) were found to be blank positions."*

**L302: "appearance", do you mean morphology?**

*Yes, we mean morphology. We changed the wording.*

*"Therefore, the morphology of the particles may have undergone alterations."*

**L305: Consider changing "excellent" to "efficient"**

*Changed as requested.*

**L374 Figure3: The scales and labels are too vague. Please increase the resolutions.**

*Changed as requested. The y-axis of the EDX-spectra are not important, as they show only the counts, which are dependent on the EDX sampling time and the current of the electron beam.*

**L437-438: Please separate the samples into cloudy and clear sky cases and discuss accordingly.**

*As requested by the reviewers, the case study section was shortened to emphasize the focus on the methods section. To emphasize the potential of the method, we decided to show a separation for airmasses influenced by an SDE and prior to that event. Although the suggestion to separate the samples into cloudy and clear sky cases is certainly interesting, we fell like that this would go beyond the scope of an AMT paper.*

**L440-444 Figure 6: Cloudy and clear sky cases should be separated.**

*As previously mentioned, we have decided to distinguish the SDE air masses from the air masses prior to that event, as this illustrates the potential of the method. Despite low particle numbers, the high accuracy in identifying the INPs allows significant differences between different air masses to be recognized.*

**L634-636: Consider archiving data in a reliable public data repository.**

*The complete data set is available for the community and can be accessed by request to the corresponding authors.*

---

## Author Comment (AC2)

**Author response to Reviewer 2**

*Firstly, we would like to thank the reviewer for reading our manuscript and the comprehensive review, which helped us to describe the method in a more focused way.*

*In the following, the reviewer comments are written in bold and our answers in italics. Text passages from the revised manuscript are in quotation marks, modified or newly added passages are marked in green.*

**The manuscript reviews how ice nucleating particles can be located on a wafer and their composition and size analysed using an electron microscope. The use of the method is exemplified on a set of wafers collected at Jungfraujoch in 2017. The coupling of the ice nucleation chamber FRIDGE with EM analysis is very suitable for the task of gaining information on the abundance of a specific category of particles active as INP at a certain temperature. However, the methodology and detail of coupling FRIDGE with EM has been discussed in previous papers and it is not made clear what novel information is provided in the current manuscript. Concerning the methodology, it seems not to go beyond what is already published in Schrod et al., 2016 and He et al., 2023.**

*Schrod et al. (2016) essentially evaluates FRIDGE as a method for determining INP concentrations based on the deposition nucleation/condensation freezing modes, without consideration of the coupling process. He et al. (2023) provides a brief overview of the coupling, with the primary focus being on the presentation of results.*

*This technical paper presents an overview of the FRIDGE measurement, outlining the critical aspects that must be taken into account to ensure the reliability of the results. It also provides a comprehensive description of the techniques employed for ice crystal detection and electron microscopic analysis. By providing a detailed description of the method, it is possible to demonstrate both the strengths and limitations of the method and to establish a guideline for the interpretation of the corresponding results. The method has been in development for several years in our working groups, and the paper by He et al., (2023) is based on our method, although the coupling of the devices (crystal detection, etc.) does not correspond exactly to our method. From the paper it is not clear, which exact method is used to identify the ice crystals and to find the corresponding positions of the ice crystal origin in the SEM.*

**The JFJ case study is a valuable dataset by itself, but the attempted validation of the FRIDGE-EM coupling by comparing to different techniques that investigated the INP composition at different activation temperatures on JFJ is not convincing. As the authors note themselves at best only a rough comparison can be made.**

*We agree that it is often challenging to make direct comparisons between INP measurements obtained from different devices, as the different collection processes, specific activation conditions and different classification schemes can have a significant impact on the resulting data. Nevertheless, we firmly believe that the comparison for the main INP classes, which is made in the revised manuscript, is meaningful enough to demonstrate that our method yields reliable results.*

**Listed in the comments below are several inaccuracies and inappropriate references.**

**The line of explanations should also be structured clearer. On several occasions, statements are made that are not comprehensible and are explained only by information given later in the text. Leading with the necessary information and explanations before a conclusion or result, would make it easier to follow.**

*We understand that in some places it may have been a little difficult to follow our line of thought. The whole manuscript has been restructured. All methodological elements are now part of the method description and discussion. Only the INP-specific results (concentration, chemistry, size) for the campaign are shown in the case study section.*

**It is irritating that the author state on couple occasions that a detailed analysis is not possible or feasible, but the analysis is then done anyway in the following.**

*Maybe at some points, reading the comparison of the case study results to other studies was a bit imprecise and thus lead the reviewer to believe that a deeper analysis is generally not possible. The idea behind our structuring of the results section (chemistry and size of the INPs at the JFJ) was to first present the results, followed by a*

*comparison with other data from the same campaign and then a comparison with previous studies at the same location. Since the comparisons were carried out study by study, we also saw the need to point out differences / non-comparable points for the purpose of providing a complete picture.*

*The structure of the result discussion (where most of these statements were made) was changed. The results are now always compared directly after the individual particle class description. This allows us to highlight comparable results from different studies. The statements are now more general, and therefore a comparison is feasible.*

*Nevertheless, the above-mentioned limitation of comparability is mentioned in a general statement.*

*Section 3.3 (INP chemistry):" It is generally difficult to make direct comparisons between the results of different INP/IR measurement techniques, as the results can vary significantly depending on the sampling configuration, ice nucleation activation conditions, and the classification schemes used for each instrumentation."*

*Section 3.4 (chemically-resolved INP size distribution): "The comparison of such INP size distributions with chemical information from different methods is difficult, since in addition to the influencing factors discussed in Sect. 3.3, a possible size selection or limitation of the sampling process, and different techniques of particle sizing may also play a role."*

**Because the manuscript lacks novelty, rigor and structure, I recommend major revisions before consideration for publication.**

*We disagree on the point, that our manuscript lacks novelty, as this paper describes the coupling between FRIDGE and SEM for the first time in detail (see first comment above).*

*A lot of structural changes have been made as recommended by all reviewers (e.g., method evaluation from the case study moved to the method section).*

**Specific comments:**

**Title: The analysis of particle morphology is not discussed in the manuscript. Consider adjusting the title accordingly.**

*We decided to stick to morphology in our title, because the morphology of particles was used to classify them. (revised manuscript Sect. 2.6). In addition to that, the focus of this paper should be the method description. And therefore, it is possible to analyze the morphology of individual INP with this method, even if we didn't focus on this aspect in our case study.*

*"The analysis by SEM and EDX is an efficient method for characterizing INPs in detail, as it provides information on elemental composition and distribution as well as on morphology and surface properties. The morphological information can be used for source apportionment (e.g., biological particles, soot, spherical particles from high temperature processes). With this detailed information, it is possible, for example, to determine the mixing state of a particle (see Fig. 3 and Fig. 4)."*

**Abstract**

**Line 29: Specify how the results can be used to evaluate parametrizations.**

*Such results may be useful as part of a closure study, which combines size-resolved aerosol composition measurements, particle class dependent INP parametrizations and INP measurements, including size and chemistry information in addition to their concentration (see Burrows et al. 2022). For more details see also comment on L85-89.*

**Introduction**

**The introduction should be shortened and focussed on motivating the presented methodology and be less of a review of the subject area in general.**

*We now focused more on the method motivation. Therefore, we shortened / re-arranged some sections in the introduction (e.g., removed the part on laboratory/field experiments and modelling (L76-93 in the original manuscript)). The motivation of our method by analyzing particles at the position where ice growth has occurred compared to the collection of ice crystals and the associated risk of artifacts, e.g., due to previous riming processes, was clarified. However, we believe it is essential to provide a comprehensive introduction and overview of INPs, as it is the basis for our method.*

*"These methods typically analyze large numbers of INPs/IRs. However, the major challenge of all these methods is that due to the extremely low number of INPs within a sampled air volume compared to the much higher number of non-INP particles (ratio ~ $1/10^4 – 1/10^6$), the separation must be carried out with a very high accuracy. Even with an accuracy of 99.9% for INP separation, this would mean that for every correctly separated INP 1000 non-INP particles would be separated incorrectly, when an INP to total aerosol ratio of $1/10^6$ is assumed. In this way, no conclusions about the chemistry of the INP would be possible at all, since there is no way to distinguish particles that have been falsely separated as INP from real INP afterwards. There is also the risk that additional artifacts can be introduced into the INP fraction during the multi-step process. This problem is partially illustrated in the comparison of the chemical analysis of the INP/IR fraction by three different methods in Worringen et al. (2015)."*

**Instead of repeatedly citing chapter 1 and 8 from the 2017 AMS reviews of Kanji et al. and Cziczo et al., it would be more helpful to cite specific references to the individual topics.**

*We think that it makes sense for general statements to cite the reviews. When there were more specific references that better reflect the statements made, we now cite those studies instead. For example, in line 59-60 we now cite Archuleta et al. (2005) and Yakobi-Hancock et al. (2013) when discussing metallic particles.*

*"Their efficiency to activate as INPs depends on the type of metallic cation as well as on the oxidation state (Archuleta et al., 2005; Yakobi-Hancock et al., 2013)."*

*Archuleta C. M., DeMott, P. J., Kreidenweis, S. M.: Ice nucleation by surrogates for atmospheric mineral dust and mineral dust/sulfate particles at cirrus temperatures, Atmos. Chem. Phys., 5, 2617-2634, doi: 10.5194/acp-5-2617-2005, 2005*

*Yakobi-Hancock, J. D., Ladino, L. A., Abbatt, J. P. D.: Feldspar minerals as efficient deposition ice nuclei, Atmos.Chem. Phys., 13, 11175-11185, doi: 10.5194/acp-13-11175-2013, 2013*

**Line 64: clarify what is meant by "the efficiency of metal oxides to activate as INP depends on the type of metallic particle". Do you mean the type of metal cation?**

*With "type of metallic particle", we meant the type of cation as well as the oxidation state. We adjusted the sentence in accordance to this explanation.*

*"Their efficiency to activate as INPs depends on the type of metallic cation as well as on the oxidation state (Archuleta et al., 2005; Yakobi-Hancock et al., 2013)."*

**Line 83: All the references provided here seem to be for measurements of INP concentrations only. Add references specific for the mentioned identification of particle type and size.**

*We rearranged this part. The general description of laboratory experiments, field experiments and modeling studies was removed due to a more focused introduction, as it was suggested by the reviewers. The references for the experiments were moved to the following section. There it is clearly stated, that the references refer to concentration measurements, so no further references are needed.*

*"Although there is a variety of methods to determine the INP concentration in the laboratory (Hoose & Möhler, 2012; Hiranuma et al., 2015; DeMott et al., 2018; Hiranuma et al., 2019) and in the field (Wex et al., 2019; Schrod et al., 2020b; Brasseur et al., 2022; Lacher et al., 2024), only a few of them are simultaneously able to report on the chemical characteristics of individual nucleating particles."*

**Line 85-89: The logic is not clear in these sentences. Clarify if it is the aerosol composition, particle class, main type, or specific type that should be related to INP to improve simulations.**

*To combine observational data and model simulations Burrows et al. (2022) identified a key need for a closure study that combines size-resolved aerosol composition measurements, particle class dependent INP parametrizations and INP measurements. In order to evaluate the results of such a closure study in the most comprehensive way, the INP measurements ideally contain, in addition to the INP concentration, information on the size and chemistry of the ice-forming particles. However, gaining this desired set of experimental parameters from field measurements is challenging.*

*Such an experiment could be conducted as follows: First you have a type-specific, and best size-dependent parametrization (which don't exist in this detail yet). Then you measure the aerosol size-distribution, and best composition in order to use this data to put into your parametrization or use model data as input variables. Then you check the resulting INP concentration with INP measurements and when available you can also check the INP composition with what individual type-specific parametrizations predict.*

*Nevertheless, this section was removed in the revised version in order to give the introduction less of a review character.*

**Line 101-103: If for IR, scavenged particles cannot be distinguished from INP, there is no information on the INP.**

*Yes, there is a risk that collected ice crystals contain scavenged particles or that they have formed by secondary ice formation. This is, why they are referred to IR and not as INPs.*

*However, this discussion on INPs/ IRs was completely removed from the revised manuscript, due to a more focused introduction.*

**Line 103-104: Clarify how INP can be identified using a CFDC by activation of sampled non-activated aerosol.**

*What was meant here was that, in contrast to ice crystal collection, the entire aerosol is collected and then activated under defined conditions. We revised the sencences.*

*"In another approach, the particles are activated under defined conditions in an online reaction chamber (e.g., Rogers, 1988) after the collection of the total aerosol. To analyze the activated particles, it is necessary to separate the ice crystals from droplets and evaporate the ice by one of the specialized inlet systems or a droplet evaporation zone. In a second step, the separated INPs/IRs are then either analyzed in the air stream or separated and transferred to an offline analysis."*

**Line 106-107: specify what is meant by the "appearances of particles" and explain what observable differences can result from the evaporation process.**

*At this point, the physico-chemical properties of the particles are meant, especially the morphology. Since soluble/volatile components may also evaporate during the ice evaporation process, this can change the properties of the particles. This means that on a morphological level, for example, information on the original particle shape can be lost, while on a chemical level, for example, highly volatile organic coatings can no longer be detected. These sentences were also removed due to a more focused introduction. We now mention the potential changes in Section 2.6, where the individual particle analysis is described.*

*"The INPs have been processed in FRIDGE (multiple activation / evacuation cycles) and they were analyzed in a high vacuum under the electron beam. Therefore, the morphology of the particles may have undergone alterations. This may be especially the case for soluble / volatile components within a sample, which may evaporate during the analysis procedure."*

**Line 111: specify what problems online methods can run into with low INP concentrations.**

*Online systems may have problems to resolve low concentrations of INPs as they have typically a lower sampling flow, resulting in low INP counts in the range of background counts. Offline methods, on the other hand, can enrich the number of INPs on the sample substrates due to higher sampling flows and longer collection times.*

**Line 129-130: Clarify how the property influencing ice nucleation can be isolated using SEM. It would be more precise to state that the chemistry, shape and size of INP can be obtained.**

*It was meant in this sense. We changed it as suggested.*

*"The activated INPs can subsequently be characterized by SEM to gain information on their chemistry, morphology and size (Fig. 1C)."*

**Fig.4: The perspective of the SEM image on top of the EDX spectra is confusing. Why is there a large shadow?**

*The image is a BSE image. The position of the particle in relation to the detector can lead to this shadowing effects.*

**Line 151: Schrod et al., 2017 state that: "From the present SEM analysis we cannot draw conclusions on the chemical composition and nature of INPs, which make only a 10^-3 to 10^-5 fraction of the randomly selected particles on a wafer." This was obviously a different FRIDGE-SEM-coupling, and the novelty of the method presented here should be highlighted. However, the description of chemical analysis of wafers provided in He et al., 2023 seems to already describe the current method.**

*With regard to the Cyprus study (Schrod et al., 2017), it should be noted that the objective was not to specifically investigate the individual INPs. Rather, the ambient aerosol on the wafer was analyzed in a random scan.*

*The method described by He at al. (2023) is based on our method, although the coupling of the devices (crystal detection, etc.) does not correspond exactly to our method. From the paper it is not clear, how they define the coordinates for the individual INPs from the ice crystals grown in FRIDGE (which is important to make sure that you re-find the INP and not just a particle on the wafer surface), as well as on how they define a particle unambiguous. For a paper with focus on the results, this may be sufficient, but a discussion on potential restrictions with respect to the results is missing in this paper. Our manuscript details the method, identifies strengths and weaknesses to give the reader an idea on how to interpret the results from such a coupling.*

**Methodology**

**This section resembles an operation manual, and it doesn't substantially go beyond Schrod et al., 2016 and He et al., 2023.**

*We can understand that the reader gets the impression of an operation manual. To lose this impression, we have restructured this section and changed the wording in some parts. In particular for the FRIDGE method already evaluated by Schrod et al. (2016), we focused more on the points that are important for the coupling procedure.*

*We do strongly disagree with the second point. Schrod et al. 2016 evaluates FRIDGE and He et al. 2023 essentially shows results of this method. Neither paper describes the coupling with its strengths and weaknesses as it is the case in this manuscript. For more details, see our argumentation further above.*

**I'm missing an explanation on how the ice is evaporated between the FRIDGE experiment and the SEM, and an analysis if IR are moved during the process.**

*The ice is evaporated after each measurement cycle in FRIDGE, consequently also at the end of a measurement before the wafers are transferred to the SEM. Of course, a particle drift during the FRIDGE measurement cannot be completely excluded, but measures are taken to limit the effect of a potential particle drift, if they actually appear.*

*"For the coupling procedure it is beneficial to stop the growth of ice before individual ice crystals grow to large sizes or coalescence, because the determination of the ice crystal center (Sect. 2.4), which is assumed to be the position of the INP, is more precise with small crystals. Additionally, this also reduces the spatial extent of potential particle drift during the ice crystal growth. By directly evaporating the ice crystals at the end of a measurement cycle with the objective of avoiding the liquid phase, the risk of possible particle drifts is also reduced."*

*Even if particle drift occurs during ice growth, this does not affect the results for clearly identified INPs. The particles would then no longer be in the center of the ice crystal causing only a blank position.*

*"A blank position may be the consequence of possible particle drift during the processing in FRIDGE (Sect. 2.3), or the result of an erroneous calculation of the ice crystal origin (Sect. 2.4)."*

**Line 267: quantify the coordinate uncertainties and discuss where the uncertainties come from. Based on the pixel size and the criteria of 30 pixels to identify an ice crystal location, the INP could be up to 300um away from the centre if the crystal grows as needle. It could be explained in more detail why 50um is a good value. Is it because at the investigated conditions the ice growth regime is plate like?**

*This uncertainty is based on the calculation of the ice crystal origin and the coordinate calibration. We added a comment on the uncertainty for the ice crystal calculation in Section 2.4. In our configuration (low pressure of near vacuum, temperatures usually -20 to -30°C), the ice crystals usually grow rather radially symmetrical, but not perfectly, hence needle-like crystals are not observed.*

*"It can be assumed that this coordinate represents the position of the corresponding INP, since an approximately radially symmetric ice crystal growth can be observed in the range of the selected activation conditions in FRDGE. Nevertheless, a potentially imperfect radial symmetry of the ice crystal growth, coupled with the restricted resolution of the FRIDGE images (20 x 20 µm), may result in an uncertainty in the calculation of the ice crystal origin. As the size of an ice crystal increases, the probability and extent of such a non-symmetrical growth also increases. The quantification of this uncertainty proved to be difficult, as it depends on the symmetry deviation present. To reduce this uncertainty based on an imperfect radial symmetry, the ice crystal position calculation should be performed on the basis of FRIDGE images, that show the ice crystals in a state close to activation."*

*The coordinate calibration uncertainty is mentioned in Section 2.5.1.*

*"Due to the limited resolution of the FRIDGE images of about 20 x 20 µm, the calibration has of course an uncertainty in the same order of magnitude."*

*Regarding the accuracy of the coordinate, the following is stated in the INP identification section (Section 2.5.2):*

*"Each ice crystal position, based on a real grown ice crystal, is inspected by SEM to identify the presence of particles. Given the uncertainties associated with the ice crystal identification process (Sect. 2.4) and the coordinate calibration (Sect. 2.5.1), it is crucial to consider not only the exact calculated coordinate but also the surrounding area. This area must take into account the aforementioned uncertainties and, at the same time, limit the probability that several particles will be observed in the scanned area. In this context, a radius of 50 µm has proven to be useful. While the previously discussed uncertainties may suggest a lager radius to be beneficial, in fact, the high substrate loading often proves to be the limiting factor."*

*This is also illustrated in a new Figure (Fig. 3 in the revised manuscript).*

**Line 276: please define refractory particles in this context.**

*Refractory particles are particles that are stable under the electron beam (defined in Ebert et al 2024).*

*Ebert, M., Weigel, R., Weinbruch, S., Schneider, L., Kandler, K., Lauterbach, S., Köllner, F., Plöger,F., Günther, G., Vogel, B., Borrmann, S.: Characterization of refractory aerosol particles collected in the tropical upper troposphere-lower stratosphere (UTLS) within the Asian tropopause aerosol layer (ATAL), Atmos. Chem. Phys., 24, 4771-4788, doi: 10.5194/acp-24-4771-2024, 2024*

*In the original version of the manuscript, we used the term refractory for the particles that can be analyzed with our method. According to the definition by Ebert et al. (2024), this term is not entirely correct for our particles, as we can also analyze particles that are partly unstable under the electron beam. In the new version of the manuscript, the term refractory is no longer used.*

**Line 277: it is unclear what is meant by "with respect to the analysed particles" here.**

*We agree that the phrasing in this line is somewhat unclear. We wanted to say that the particles that can be analyzed with this method are not affected by the loss of volatile particles. Due to the restructuring this part was removed in this section. It is now part of the single particle analysis (Sect. 2.6).*

*"At this point, it should also be noted that our findings revealed an absence of small volatile compounds on the wafers in the EM, which are typically present in larger numbers in the total aerosol. Presumably, there is a loss of these components during sampling collection or processing. However, as these volatile particles are not*

*known to be efficient INPs in the considered temperature range (Murray & Liu, 2022), it can be assumed that their absence does not significantly affect the results."*

**Line 279: Explain why adjusting the scanning radius optimizes the analysis. It could be quantified on a lightly loaded wafer what the distance of INP from the coordinate usually is to exclude particles outside the range.**

*Our chosen radius of 50 μm represents a compromise. On the one hand, as already discussed above, possible uncertainties in the coordinate calculation and calibration in the SEM must be taken into account. On the other hand, the radius must not be too large, as otherwise the chance of finding several particles in the selected radius increases. The number of particles in the corresponding scanning radius depends strongly on the total wafer load and often represents the limit for the selected radius. If the wafers are heavily loaded, hardly any particles can be unambiguously identified. For lightly loaded wafers, expanding the scanning radius can provide an opportunity to assign more particles to ice crystals and thus increase the yield.*

**Line 285: Explain how it can be known if a feature is relevant for ice formation.**

*This should just be a list of particle features that can be analyzed with the SEM method which may be important for its ice nucleation ability. We do not say that we know explicitly what exactly make a particular particle form ice from our analysis. We changed the wording to physico-chemical properties instead of mixing state and distinct morphological patterns on the particle surface.*

*"The comprehensive single particle analysis (Sect. 2.6) enables the identification of physico-chemical properties that may be pertinent to ice formation."*

**Line 299-302: Describe the stage where ice is evaporated before the SEM analysis.**

*The ice is evaporated after each measurement cycle in FRIDGE. So, all the ice is gone before the wafer is taken to the SEM.*

*"By directly evaporating the ice crystals at the end of a measurement cycle with the objective of avoiding the liquid phase, the risk of possible particle drifts is also reduced."*

**2.6 Chemical classification: Fig. S1 could be shown here and referred to, to guide the reader and help to follow the descriptions.**

*The Figure was added to the main paper.*

**Line 307: Again, how can be identified if a certain surface property is relevant for ice nucleation?**

*In the current setup we cannot investigate this in detail, that's true. But in general, it's possible to analyze the surface properties of the identified INPs. Even if we cannot define the exact location where ice growth started, we can analyze the INP surface for structures that can promote ice formation.*

**Line 315: Explain, based on what information the classification scheme is modified. Does this make the scheme subjective?**

*The classification scheme is essentially based on chemistry and in some cases also on morphology. The mentioned modification does not refer to the criteria of the individual particle classes, but rather to the grouping of the particle classes based on different abundances at different locations. We added and modified the sentences to clarify this.*

*"In the following section we define a classification scheme, which is mainly based on elemental composition (Fig. 5) and in some cases on the morphology of particles (Fig. 7). ... Based on the research question or the occurrence of specific particle classes at the sampling site, the classification scheme may be adapted."*

**Fig. 3: x-, y-axis scale are too small to read. Also, axis labels should be added.**

*The scales are increased and the missing labels are added. The y-axis of the EDX-spectra are not as meaningful, as they show only the counts, which are dependent on the EDX sampling time and the current of the electron beam.*

**Case Study**

**Clarify if sampling was conducted downstream of an inlet or in the open.**

*We added a sentence to clarify this.*

*"FRIDGE sampling was conducted downstream of the GAW total inlet (Lacher et al., 2018)."*

**Line 391: Define cINP as INP concentration.**

*We removed the cINP abbreviation due to rare occurrence in the script.*

**Fig.4: increase the contrast of the figure. Corresponding sample numbers are not shown. In the caption, do you mean adapted from Weber (2019) instead of modified according to Weber (2019)? Specify that the 5-day average is a running average.**

*Fig. 4 (now Fig.8) has been adjusted: Corresponding sample numbers were added.*
*In the caption we changed "modified according to Weber (2019)" to "adapted from Weber (2019)" and it is now specified in the caption that is a 5-day running average.*

**Line 402: Provide a reference for Saharan dust being active below -20°C.**

*Niemand, M., Möhler, O., Vogel, B., Vogel, H., Hoose, C., Connolly, P., Klein, H., Bingemer, H., DeMott, P., Skrotzki, J., Leisner, T.: A Particle-Surface-Area-Based Parametrization of Immersion Freezing on Desert Dust Particles, J. Atmos. Sci, 69 (10), 3077-3092, doi: 10.1175/JAS-D-11_0249.1, 2012*

*Murray, B. J., O'Sullivan, D., Atkinson, J. D., Webb, M. E.: Ice nucleation by particles immersed in supercooled cloud droplets, Chem. Soc. Rev., 41, 6519-6554, doi: 10.1039/c2cs35200a, 2012*

*The references were added to the manuscript.*

**Line 402: From Fig. 4 it is not clear when the 14 samples were taken. 31 cINP on 11 days are marked with triangles.**

*We added the corresponding sample numbers to Fig.4 (now Fig. 8). Now it should be clear, on which days the 14 samples were collected. For each sample, the INP concentration for -20°C, -25°C and -30°C was marked with a triangle, that's why there are more than 14 triangles.*

**Fig. 5: What's the point of this figure? As shown in Fig. 2 and explained in Sec. 2.4. the edge region ice crystals were excluded for SEM. There should therefore be clearly more ice crystals detected by FRIDGE than SEM positions.**

*This Figure and the related analysis are moved to the section 2.4 in the method part, as suggested by one reviewer. We have therefore moved Figure 2 of the original script to the supplement.*

*The point of this figure is to show that two separate ice crystal identification algorithms (FRIDGE and the one discussed in section 2.4) come to similar results, which serves to validate the efficacy of our software-based approach to identify the ice crystals from the from the FRIDGE images. Note, that the edges are also excluded in FRIDGE, the counting areas are the same for both methods.*

**In the caption, mention that the 1:1 line is shown in red.**

*Changed as requested.*

**Line 420: Clarify why the algorithm is inconsistent with excluding the temperature sensor area.**

*Based on the standard wafer positioning in FRIDGE, the counting algorithm has a given area in which the ice crystals are detected from the FRIDGE images. Normally, the area around the temperature sensor is excluded from the analysis. However, in some cases it may happen, that the wafer is not correctly positioned is FRIDGE, which can lead to the temperature sensor being in the counting area of the algorithm. The counting algorithm is not able to identify this by itself, and counts all differences in brightness around the sensor as ice crystals.*

*Further improvements in the calculation of individual ice crystal positions can be achieved by specifically avoiding the sources of error previously mentioned.*

*This part was removed in the revised manuscript as it only represents a special case and should be avoided according to the description in FRIDGE.*

**Line 437: Explain why a meteorological interpretation is not feasible for the current manuscript. Chapter 6 in the thesis of Weber 2019 contains a meteorological interpretation of the JFJ results and Fig.6.18 therein shows a comparison during and outside SDE.**

*Here we wanted to say that such a detailed interpretation goes beyond the scope of this manuscript, submitted for AMT. Moreover, the number on identified particles is low. This part was rewritten.*

*Nevertheless, we decided to show the subdivision into SDE and non-SDE, as it shows that the method allows statements to be made about INP-relevant trends due to its high accuracy of identification despite low INP numbers.*

**Fig.6: Fig. S2 implies that the composition of the INP population active at -30°C can substantially vary from day to day. E.g., looking at W2, W7, W11 and W34 where a similar number of particles were analysed, the abundance of components is never similar. Please analyse and discuss the implications on sampling statistics and what the total chemical composition in Fig.6 represents.**

*Of course, the chemical composition can vary from day to day. Regarding the main particle classes, from the previously mentioned samples, only W7 looks significantly different, as it has very few mineral particles. Overall, the composition is rather similar.*

*The requested evaluation was added to the manuscript as follows.*

*Section 3.2: „Although the number of identified INPs appears comparatively low for a campaign period of five weeks, these INPs were identified with a high degree of reliability (Sect. 2.5.2). The small number of particles identified bears the risk that individual, time-limited variations occurring randomly during the sampling periods may influence the resulting total composition toa certain degree. It should therefore be noted that the results presented below do not comprehensively reflect the main composition of the INPs over the entire campaign period. Nevertheless, it can be shown that the method provides valid results for the main groups of INPs (see confidence intervals for Fig. 9 in the supplement (Tab. S2)).“*

*Section 3.3: "Due to the limited number of identified INPs per sample (Fig. S3), mapping daily fluctuations is not possible for this campaign. Figure 9a) provides the chemical composition for all INPs sampled over the entire campaign period, within the restrictions mentioned in Sect. 3.2."*

**In the caption, what artifact is excluded?**

*A particle with traces from gold is excluded, as gold may originate from our gold wires inside the EAC. This now clarified in the text.*

*"One particle with attached gold traces was classified as an artifact and therefore excluded from further discussions."*

**Line 487-488: It is mentioned that a comparison of abundance is not possible. Clarify the purpose of doing a comparison if the abundance, which is the main result, cannot be compared.**

*We stated that the particle classes they found for their IRs are similar to our INP particle classes, which represents an initial comparison. However, since the activation conditions have a substantial influence on the activation of an INP, it is generally difficult to compare chemical compositions of INPs that have been activated under different conditions. Therefore, we just mentioned, that is not possible to compare the distinct abundances for different particle classes, because of different activation conditions.*

*However, the discussion part has been restructured. We are now focusing more on supporting our data with results from other studies rather than on discussing differences and mentioning points which we cannot compare.*

**Line 505: Clarify how the agreement can be considered good if only a rough comparison can be made. The comparison suffers from the previous mentioned differences in activation temperature and overall technique, making a direct comparison questionable.**

*We agree and refer to our previous response.*

**Line 537: Specify the role of volatility.**

*Typically, the size distribution of an atmospheric aerosol is shifted towards smaller diameters, compared to our INP size distribution. We assume, this is the case for two reasons: On the one hand, particles with a diameter from 0.5 µm are known to be more ice active in the considered temperature regime, on the other hand, we have noticed an absence of small volatile compounds on the wafer in the SEM.*

*The particle loss is mentioned in Section 2.6:*

*"At this point, it should also be noted that our findings revealed an absence of small volatile compounds on the wafers in the EM, which are typically present in larger numbers in the total aerosol. Presumably, there is a loss of these components during sampling collection or processing. However, as these volatile particles are not known to be efficient INPs in the considered temperature range (Murray & Liu, 2022), it can be assumed that their absence does not significantly affect the results."*

**Fig. 7: Clarify if analysed particles activated at all of the listed RH's or at least one.**

*In FRIDGE all samples were activated at a set of 3 temperatures (-20°C, -25°C and -30°C) and a minimum of 4 relative humidities (95% / 97% / 99% / 101%). Since measurements at higher humidities typically show a larger number of ice crystals, we have chosen the measurements cycles at RH=99% and RH=101% for the coupling procedure. RH = 95 / 97% was chosen for one sample due to cluster formation at higher RH. We added the information to the corresponding figure captions.*

**Line 549: This is a misunderstanding. DeMott et al. 2010 found that the concentration of INP correlates to the concentration of particles >500nm. Not that they are >500nm.**

*This is basically correct, but from the results of DeMott et al. (2010) it is very likely that INPs are often larger than 500 nm, surely not all INPs are larger than 500 nm. The larger the particle the higher is the probability of active sites on the surface.*

*"This agrees to well-established findings in the literature substantiating that most particles that act as effective ice nuclei are above a size of 500 nm (DeMott et al., 2010)."*

**Line 569: Quantify the statistical uncertainty."**

*We have quantified the confidence intervals for the chemical composition given in Fig. 9 (revised version). The values are shown in the supplement (Tab. S2). In addition, all particles with a $d_{pa}$ larger that 6 µm have been summed up, due to their low abundance.*

**Line 573: Looking at Fig.7 in Lacher et al., 2021 the second maximum in the OPC data appears between 0.5.1um. Clarify how the IR OPC data is compared to the current results.**

*We mean the broad maximum from 2-5 µm (OPS) and between 2-3 µm for the Sky-OPC. This indicates an enrichment of particles with larger diameters, supporting our theory of an enrichment of larger particles in the ice active fraction.*

**Line 574: Is a comparison just difficult or not possible?**

*We now agree that a meaningful comparison is probably not possible due to the different sizes of the identified particles. In this paragraph (comparison of the size-resolved chemical composition), we have decided to refrain from a comparison with Lacher et al. (2021) and limit the comparison to Worringen et al. (2015).*

**Line 586: Explain how the size of ice crystals is linked to the size of INP/IR.**

*The description of Ice-CVI by Mertes et al. 2007 states that only ice crystals between 3 and 20 µm are extracted, as the probability of scavenging is low for them. If an unactivated INP already has a size close to or above 20*

*μm, the resulting ice crystal will probably be larger than 20 μm and will therefore not be extracted. So consequently, there is a size limitation for big particles.*

*This sentence was removed for the revised manuscript, because it does not fit into the new text.*

**Line 589: Better references for the importance of immersion freezing are Ansmann et al. 2009 or Westbrook and Illingworth, 2011.**

*Changed to Ansmann et al. (2009). Additionally, Murry et al. 2012 was also added as a reference.*

*Murray, B. J., O'Sullivan, D., Atkinson, J. D., Webb, M. E.: Ice nucleation by particles immersed in supercooled cloud droplets, Chem. Soc. Rev., 41, 6519-6554, doi: 10.1039/c2cs35200a, 2012*

**Line 591: It needs to be pointed out clearly what part of the presented method is novel.**

*We now state in the conclusions as well as in the introduction, that the method was already used in campaigns, and that the discussion from a methodological perspective in new. For more details, we refer to the previous comments and our answers to the other reviewers.*

*Section 1: "The FRIDGE-SEM-coupling technique has been used for several campaigns in recent years, providing valuable results (Schrod et al., 2017; Schrod et al., 2020b; Weber, 2019; He et al., 2023). Details of the FRIDGE method were described by Schrod et al. (2016)."*

*Section 4: "A method for analyzing the concentration and individual physico-chemical properties of ambient INPs, which has been used in several campaigns (Schrod et al., 2020b; He et al., 2023), is discussed here from a methodological perspective. The method benefits from the coupling of two instruments already used for the analysis of INPs and IRs: the static diffusion chamber FRIDGE and the SEM. As the individual methods are already known, the focus here was on a description of the coupling and the associated advantages and uncertainties, as well as the resulting potential of the method."*

**Line 594: The analysis of morphology and surface properties has not been demonstrated in this work.**

*The morphology is now explicitly mentioned in chapter 2.6 ("individual particle analysis") and an example for a mixed particle is also provided. The possibility to identify surface properties is given by the method, but not discussed in the case study. But since the focus of this paper is to highlight the possibilities of the method, we find it important to point out that this is generally possible.*

**Line 606: Specify what improvements are necessary.**

*Improvements are generally necessary in the wafer cleaning process, in improving the uncertainties during the coordinate calculation for the ice crystal origin, and in increasing the identification rates.*

*The knowledge about the background counts for clean wafers can be improved, for example, by checking every cleaned wafer and not just random samples from a set of cleaned wafers.*
*The uncertainties in crystal detection could be reduced by the following. The shortest possible measuring time in FRIDGE enables a more precise determination of the crystal origin and minimizes the risk of particle drift. Another improvement could be a camera with a higher resolution in the FRIDGE setup. This could increase the resolution of the FRIDGE images, which would lead to a more precise recovery of the calibration point on the SEM.*
*A systematic, automated analysis of the INPs could possibly also reduce the working time and thus open up the possibility of a larger number of samples.*

**Line 625-627: It has not been demonstrated in this work that meaningful structural information can be obtained, and it is unclear how information on the relevance of a property for ice nucleation can be gained with this method.**

*We have revised the sentence.*

*"The detailed information on physico-chemical particle properties that can be obtained from SEM can be a valuable addition to pure INP counting methods for gaining information on the relevance of particle properties*

*to ice nucleation efficiencies and could help to bridge the knowledge gap towards INP aerosol-type-specific parametrizations that could be used in modeling studies (Burrows et al., 2022)."*

**Line 631-632: Clarify what element of the method need adaptation. It can be assumed that only the EAC would be flown, and the method of analysis remains the same. Also, is the EAC not already usable in an aircraft setting?**

*A prototype was used in 2011 in an aircraft campaign, but the samplers had to be rebuilt in terms of material and adjustable parameters due to the different environmental conditions in the UT/LS. For example, it is assumed that the HV has to be reduced to avoid flash overs in regions with lower pressure.*

*But also, the FRIDGE measurement procedure has to be adapted. FRIDGE has to be operated at colder temperatures to get closer to the conditions in the UT/LS region for example. Therefor the wafer cleaning has to be improved, as the background counts typically rise sharply for measurements at -35°C.*

**Technical corrections:**

**Line 48: I can't find information about hydrogen-bridging functional groups in Kanji et al., 2008. Double check the reference.**

*This is a mistake on our part. The reference Kanji et al. (2008) was incorrectly placed as a reference for hydrogen-bridging. It has been moved accordingly as a reference for coating.*

*"In addition to the prevailing environmental conditions i.e., temperature and humidity, the potential for an INP to become activated is dependent upon individual particle properties (surface imperfections (Kiselev et al., 2016), chemical composition and specific chemical properties, crystal structure, coating (Kanji et al., 2008), etc.) as well as its atmospheric processing including potential agglomeration or pre-activation (Marcolli, 2017)."*

**Line 51: Replace Hoose & Möhler, 2012 with a more specific reference about the influence of particle size on ice nucleation.**

*The reference has been changed to: Welti et al. (2009)*

*Welti, A., Lüönd, F., Stetzer, O., Lohmann, U.: Influence of particle size on the ice nucleating ability of mineral dusts, Atmos. Chem. Phys., 9, 6705-6715, doi: 10.5194/acp-9-6705-2009, 2009*

**Line 52: Marcolli 2017 would be a more specific reference for pre-activation than Abdelmonem et al., 2020.**

*Changed as requested.*

**Line 174: I can only find information on the EAC in the Supplement of DeMott et al., 2018 and it is not clear how the EAC was modified compared to the description in Klein et al., 2010.**

*The EAC is described in detail in Klein et al., 2010 and in Schrod et al., 2016. Our intention here was to state that different versions have been used but not that they are described in detail in the given literature. We rearranged the section:*

*"Aerosol is precipitated onto the substrates using an Electrostatic Aerosol Collector (EAC) (Klein et al. 2010). Several EACs have been deployed for the use in the laboratory (DeMott et al., 2018), in field campaigns (DeMott et al., 2024), for measurements with unmanned aerial vehicles (Schrod et al., 2017), and for long-term observations at research stations (Schrod et al., 2020b). The most recent version, which was also used in the case study (Sect. 3), PEAC7, is a programmable EAC (Schrod et al., 2016) designed for semi-automated operation for one week of daily sampling."*

**Line 175: Lacher et al., 2024 report FRIDGE measurements using filter samples, the EAC seems not to have been used.**

*Yes, that is a mistake. The EAC was used but the corresponding measurements were not included in the fonal study. We removed the reference.*

**Line 270: "two effects" instead of "to effects"**

*This section has been rewritten. The wording no longer occurs in the new version.*

**Line 496: superfluous )**

*The complete case study discussion was restructured. The sentence was deleted.*

*References:*

Abdelmonem, A., Ratnayake, S., Toner, J. D., Lützenkirchen, J.: Cloud history can change water-ice-surface interactions of oxide mineral aerosols: a case study on silica, Atmos. Chem. Phys., 20, 1075-1087, doi:10.5194/acp-20-1075-2020, 2020

Ansmann, A.; Tesche, M.; Seifert, P.; Althausen, D.; Engelmann, R.; Fruntke, J.; Wandinger, U.; Mattis, I.; Müller, D. Evolution of the ice phase in tropical altocumulus: SAMUM lidar observations over Cape Verde, J. Geophys. Res., 2009, 114, D17208, doi:10.1029/2008JD011659

Cziczo, D. J., Ladino, L., Boose, Y., Kanji, Z. A., Kupiszewski, P., Lance, S., Mertes, S., Wex, H.: Measurements of Ice Nucleating Particles and Ice Residuals, Meteor. Mon., 58, 1-13, doi: 10.1175/AMSMONOGRAPHS-D-16-0008.1, 2017

DeMott, P. J., Prenni, A. J., Liu, X., Kreidenweis, S. M., Petters, M. D., Twohy, C. H., Richardson, M. S., Eidhammer, T., Rogers, D. C.: Predicting global atmospheric ice nuclei distributions and their impacts on climate, P. Natl. A. Sci, 107, no. 25, 11217-11222, doi: 10.1073/pnas.0910818107, 2010

DeMott, P. J., et al.: The Fifth International Workshop on Ice Nucleation phase 2 (FIN-02): laboratory intercomparison of ice nucleation measurements, Atmos. Meas. Tech., 11, 6231-6257, doi: 10.5194/amt-11-6231-2018, 2018

He, C., Yin, Y., Huang, Y., Kuang, X., Cui, Y., Chen, K., Jiang, H., Kiselev, A., Möhler, O., Schrod, J.: The Vertical Distribution of Ice-Nucleating Particles over the North China Plain: A Case of Cold Front Passage, Remote Sens., 15, 4989, doi: 10.3390/rs15204989, 2023

Hoose, C. and Möhler, O.: Heterogeneous ice nucleation on atmospheric aerosols: a review of results from laboratory experiments, Atmos. Chem. Phys., 12, 9817-9854, doi: 10.5194/acp-12-9817-2012, 2012

Kanji, Z. A., Florea, O., Abbatt, J. P. D.: Ice formation via deposition nucleation on mineral dust and organics: dependence of onset relative humidity on total particulate surface area, Environ. Res. Lett., 3, 025004, doi: 10.1088/1748-9326/3/2/025004, 2008

Kanji, Z. A., Ladino, L. A., Wex, H., Boose, Y., Burkert-Kohn, M., Cziczo, D. J., Krämer, M.: Overview of Ice Nucleating Particles, Meteor. Mon., Vol. 58, doi: 10.1175/AMSMONOGRAPHS-D-16-0006.1, 2017

Klein, H., Haunold, W., Bundke, U., Nillius, B., Wetter, T., Schallenberg, S., Bingemer, H.: A new method for sampling of atmospheric ice nuclei with subsequent analysis in a static diffusion chamber, Atmos. Res., 96, 218-224, doi: 10.1016/j.atmores.2009.08.002, 2010

Lacher, L., Clemen H.-C., Shen, X., Mertes, S., Gysel-Beer, M., Moallemi, A., Steinbacher, M., Henne, S., Saathoff, H., Möhler, O., Höhler, K., Schiebel, T., Weber, D., Schrod, J., Schneider, J., Kanji, Z. A.: Sources and nature of ice nucleating particles in the free troposphere at Jungfraujoch in winter 2017, Atmos. Chem. Phys., 21, 16925-16953, doi: 10.5194/acp-21-16925-2021, 2021

Lacher et al.: The Puy de Dôme ICe Nucleation Intercomparison Campaign (PICNIC): comparison between online and offline methods in ambient air, Atmos. Chem. Phys., 24, 2651-2678, doi: 10.5194/ACP-24-2651-2024, 2024

Marcolli, C.: Pre-activation of aerosol particles by ice preserved in pores, Atmos. Chem. Phys., 17, 1595–1622, https://doi.org/10.5194/acp-17-1595-2017, 2017.

Schrod, J., Danielczok, A., Weber, D., Ebert, M., Thomson E. S., Bingemer H. G.: Re-evaluating the Frankfurt isothermal static diffusion chamber for ice nucleation, Atmos. Meas. Tech., 9, 1313-1324, doi: 10.5194/amt-9-1313-2016, 2016

Schrod, J., Weber, D., Drücke, J., Keleshis, C., Pikridas, M., Ebert, M., Cvetkovic, B., Nickovic, S., Marinou, E., Baars, H., Ansmann, A., Vrekoussis, M., Mihalopoulos, N., Sciare, J., Curtius, J., Bingemer, H. G.: Ice nucleating particles over the Eastern Mediterranean measured by unmanned aircraft systems, Atmos. Chem. Phys., 17, 4817-4835, doi: 10.5194/acp-17-4817-2017, 2017

Weber, D.: Eisnukleation von Aerosolen: Laborexperimente und Messungen im Feld, Ph.D. thesis, Goethe Universität Frankfurt, Germany, 2019

Westbrook, C. D., and Illingworth, A. J.; Evidence that ice forms primarily in supercooled liquid clouds at temperatures > −27°C, Geophys. Res. Lett., 2011, 38, L14808, doi:10.1029/2011GL048021.

---

## Author Comment (AC3)

**Author response to Reviewer 3**

*First of all, we would like to thank the reviewer for reading and evaluating our manuscript. The comments helped us to improve the manuscript and strengthen the desired focus on the methodological part.*

*In the following, the reviewer comments are written in bold and our answers in italics. Text passages from the revised manuscript are in quotation marks, modified or newly added passages are marked in green.*

**In this manuscript, the authors present an offline method that combines an ice nucleation counter, the FRankfurt Ice nucleation Deposition freezinG Experiment (FRIDGE), with Scanning Electron Microscopy (SEM) to analyze the chemical composition, size, and morphology of Ice Nucleating Particles (INPs) collected from ambient air. The authors begin by providing an overview of the methodology, followed by a case study demonstrating its application to ambient aerosols collected during the 2017 CLACE/INUIT campaign at the Jungfraujoch station.**
**The methodological section appears unfinished, as it fails to demonstrate all the potential features the authors claim the method can analyze (e.g., morphology, pores). Additionally, the lack of standards in this section raises concerns about evaluating the performance of the technique and estimating statistical error, particularly for size measurements and the coupling of particle size with ice nucleation efficiency.**

*The methodological part of the manuscript has been improved. Uncertainties (e.g., for the identification of ice crystal origins (Sect. 2.4 in the revised manuscript) and the coordinate calibration for SEM (Sect. 2.5.1 in the revised manuscript)) are now defined and discussed more clearly. Please note that the pure FRIDGE method including statistics has already been evaluated by Schrod et al. (2016).*

*Sect. 2.4:* "*It can be assumed that this coordinate represents the position of the corresponding INP, since an approximately radially symmetric ice crystal growth can be observed in the range of the selected activation conditions in FRDGE. Nevertheless, a potentially imperfect radial symmetry of the ice crystal growth, coupled with the restricted resolution of the FRIDGE images (20 x 20 μm), may result in an uncertainty in the calculation of the ice crystal origin. As the size of an ice crystal increases, the probability and extent of such a non-symmetrical growth also increases. The quantification of this uncertainty proved to be difficult, as it depends on the symmetry deviation present. To reduce this uncertainty based on an imperfect radial symmetry, the ice crystal position calculation should be performed on the basis of FRIDGE images, that show the ice crystals in a state close to activation.*"

*Sect 2.5.1:* "*Due to the limited resolution of the FRIDGE images of about 20 x 20 μm, the calibration has of course an uncertainty in the same order of magnitude.*"

*The method evaluation from the case study was also included in the methodological section. The comparison between the different ice crystal counting methods (formerly Fig. 5) was moved to the method section (Section 2.4 "identification of ice crystal positions") and replaces the graphical representation of coordinate determination (formerly Fig. 2), which has been moved to the supplement. The discussion on the identification rates was included in section 2.5.2 (now called "INP identification").*

*Sect. 2.5.2:* "*The number of INPs that can be unambiguously attributed to an ice crystal origin is significantly influenced by the total wafer loading, which is determined by the sampling parameters (e.g., flow rate, sampling time, deposition efficiency) in combination with the aerosol concentration present. However, even if the aerosol concentration is known, it is difficult to specify a suitable collection volume in advance, as the ratio of potential INPs to the total aerosol also plays a role. This ratio is variable and usually unknown prior to measurement. As a result, the amount of atmospheric aerosol and the proportion of INPs deposited on a wafer are highly variable. This variability is also seen in the identification rates, which is why it would be misleading to give an average identification rate for the method presented. However, a specific identification rate for the case study conducted at the high-altitude research station Jungfraujoch (Sect. 3) can be given here as a guideline. The average INP identification rate was calculated to be 30% (ranging from 13% to 50%). Furthermore, the study identified the presence of multiple particles at 45% of the locations (ranging from 7% to 81%), while the remaining 25% (ranging from 2% to 66%) were found to be blank positions.*"

*Section 2.6 (now called "individual particle analysis") has been generalized, to show the potential of the coupling method. The morphology is now mentioned in Sec. 2.6 as it can be used to classify INPs. We included*

*also a BSE picture showing a particle with different chemical compositions on its surface (Fig. 4 in the revised manuscript), to illustrate that it is also possible to see the element distribution on the particle surface.*

*"The analysis by SEM and EDX is an efficient method for characterizing INPs in detail, as it provides information on elemental composition and distribution as well as on morphology and surface properties. The morphological information can be used for source apportionment (e.g., biological particles, soot, spherical particles from high temperature processes). With this detailed information, it is possible, for example, to determine the mixing state of a particle (see Fig. 3 and Fig. 4)."*

*The size of the particles can be determined with high precision due to the high resolution of the electron microscope, in contrast to the FRIDGE camera, where the resolution is limited.*

*For more details we refer also to our following responses and our comments to Review 1 and 2.*

**The case study lacks sufficient statistical analysis and fails to provide a clear connection to other parameters measured during the campaign (e.g., aerosol size distribution and number concentration).**

*The focus of the manuscript is the methodological description and discussion of the FRIDGE-SEM coupling. Therefore, the case study Section was shortened. Nevertheless, for statistical purposes, we have calculated the confidence intervals (Supplement Fig. S2) for the chemical composition shown in Fig. 9 (revised manuscript). Additionally, we have connected our INPs size measurement to the aerosol size distribution from the CLACE INUIT campaign provided by Weber (2019). For more details see our comment on Fig, 7 below.*

**It appears the authors did not clearly decide whether to (A) describe and evaluate the method in detail, suitable for AMT journal or (B) focus on the CLACE/INUIT campaign with further analysis. As a result, the manuscript presents two incomplete studies that are not well connected.**

*We can see that a reader may get this impression. In fact, the decision between ACP and AMT was under discussion for a long time. In the end, we decided on AMT, but it seem that the manuscript was not sufficiently adapted to this decision.*
*For the revised version, we have placed a clear focus on the methodological discussion and shortened the case study.*

**Specific comments**

**- A clearer explanation is needed as to why deposition and condensation freezing modes are grouped together as the two primary ice nucleation modes. Specifically, the cycles with FRIDGE include measurements taken below water saturation at RH=100%, which would typically prevent condensation freezing. Is this grouping due to uncertainty in the RH measurements, which could allow for RH to exceed 100%?**

*Apart from an uncertainty in the RH measurement, the two nucleation modes are mainly combined, as we measure both below RH=100% water saturation and specifically beyond RH=100% (95% / 97% / 99% / 101%). While INPs are activated solely by deposition freezing at RH distinctly below 100%, condensation freezing also takes place around/above 100%.*
*As more INPs are usually activated at higher RH, the series of measurements around RH=100% are better suited to the coupling method.*

**- The authors argue that volatile compounds are not detected and that these compounds are generally not known to be efficient INPs. Is there any estimation of which type of volatile material is lost, a lower estimation of vapor pressure? How does this affect SEM measurements, notably for carbon? Could there be potential effect of freezing point depression, such as the competition for adsorption on active ice nucleating sites between water and other volatile compounds?**

*We work under near-vacuum conditions in both the FRIDGE and the ESEM. Before starting the measurement, a vacuum is created in the FRIDGE chamber (p<<0.1 mbar). After each measurement cycle, the chamber is evacuated again until the chamber pressure from before the measurement is restored. This is to ensure that all water vapor that has been added to the chamber during the measurement, and thus also the ice crystals, is removed again before starting the next measurement cycle. So typically, the chamber is evacuated 13 times for one sample. In the SEM, the samples are analyzed in a high vacuum at approx. $10^{-6}$ mbar. All compounds, which are stable under the vacuum conditions can be analyzed with SEM/EDX.*

*A second limitation is the stability under the electron beam. All particles which are not stable under the electron bombardment can possibly be seen, but no chemical classification is possible as they evaporate during the spot analysis. In SEM, HVOCs and (ammonium)nitrates are particularly problematic, whereas sulfates, nitrates and especially carbon-rich particles as an important INP compound can be analyzed by SEM/EDX. As a consequence, the limitation for the analysis of volatile particles is not given by SEM.*

*Nevertheless, we observe that the dominant secondary atmospheric particles of the atmosphere are not visible on the wafers in the EM. It is therefore reasonable to suspect that these particles are lost during the sampling collection or processing. In the manuscript we say the following:*

*"At this point, it should also be noted that our findings revealed an absence of small volatile compounds on the wafers in the EM, which are typically present in larger numbers in the total aerosol. Presumably, there is a loss of these components during sampling collection or processing. However, as these volatile particles are not known to be efficient INPs in the considered temperature range (Murray & Liu, 2022), it can be assumed that their absence does not significantly affect the results."*

**- The case study part focuses primarily on chemical composition and size analysis, but the title of the paper also mentions morphology. The conclusion, line 594, states, " This coupling allows for detailed analysis of various INP properties, such as chemical composition, mixing state, size, morphology, and surface properties like cracks or pores." Yet, no information is provided regarding the mixing state, morphology or surface properties.**

*It is true that the case study was primarily focused on the chemistry and size of INPs.*
*Nevertheless, the detailed single particle analysis by SEM and EDX provides, in addition to the chemical composition, information on the surface structure (secondary electrons) and the element distribution (backscattered electrons). Even if it is not possible to define the exact origin of the ice growth on the particle surface, with the current setup, the method can generally be used to investigate the mixing state, morphology and surface of the identified INPs.*
*The potential of the method is described in more detail in the new version of the manuscript in Section 2.6. There we mention, that the morphology can be used to identify certain particle classes (soot, biological particles) and sometimes to determine the origin of particles (e.g., irregular geogenic minerals vs. fly ash). We have also added an example of a mixed particle which, in contrast to the mixed particle in Fig. 3 of the old manuscript, allows clear differentiation of the regions with different chemistry.*

**- Figure 3: The x and y axis are not readable. Only few EDX spectra are provided in figure 3 for the case study analysis. How can we evaluate the reliability of the chemical composition on all particles?**

*The labeling of the x-axis has been adjusted. The y-axis provides information on the number of counts which depend on the parameters set during the EDX analysis. The exact numbers are of no further importance for the evaluation carried out here, the peak intensities are clearly visible.*
*The entire description of the particle classes was isolated from the case study and generalized. This section now highlights the potential and discusses the limitations of this method for analyzing INPs. We provided example EDX spectra in Fig. 7 (revised manuscript) to illustrate the classification of INP types. We think it would be overly excessive and uninformative to present those for all particles.*

**- Figure 4: How is the 5-day average calculated and plotted? Why are there no error bars? What is the background level? How many measurements were performed per wafer, 1 only? Schrod, J. et al. (2016) provided statistical analysis for FRIDGE.**

*The 5-day average is a running average which is calculated for each day from the two previous, the current and the two following daily values.*

*For a detailed discussion and analysis of the INP concentrations at the JFJ, error bars would be essential, we agree with the reviewer. However, such a discussion is not within the scope of this paper. At this point, we show the Figure to give the reader an overview and to illustrate why we focus on the activated INPs at -30°C for the analysis. In this case, we see no need to discuss uncertainties in the INP measurement in depth. This FRIDGE method was evaluated by Schrod et al. (2016).*

*As you have already mentioned, Schrod et al. (2016) provides a statistical analysis of the FRIDGE method. A value of 20% is given for the statistical fluctuation of the determined ice crystal number. A statistical fluctuation*

*of the same order of magnitude can be assumed for the concentrations shown here, which is based on 1 measurement. This is now also described in the manuscript.*

*"The concentration for each sample is calculated on the basis of one measurement. The relative error of the counting uncertainty for individual measurements is 20% (Schrod et al., 2016), so the error of the concentrations given here is also in this range."*

*Typically, even cleaned wafers show low ice formation activity at -30°C. As already mentioned in chapter 2.1, this is typically around 10 counts per wafer. These counts are normally subtracted from the number of ice crystals before the concentration is calculated. The concentrations are therefore usually already background corrected. For the campaign at the JFJ, particular emphasis was paid to the cleanliness of the wafers, as we expected very low INP concentrations in advance. The background counts for -30°C were less than 3 counts per wafer. This results in a maximum background concentration of 0.03 $L^{-1}$ for a collection volume of 100 L. As the collection volumes were generally larger than 100 L, the value decreases accordingly.*

**- Figure 5: What is the red curve, is this the ideal version with 1:1 ratio?**

*Yes, the red line shows the 1:1 ratio. A description was added to the figure caption.*

**- Figure 6: Why is n=199, shouldn't it be 200? Are the same particles active at all RH values (95%, 97%, 99%, 101%)? The percentages should be linked to the previous figure, as you mention the total chemical INP composition, but only 15% of all INPs are considered. Additionally, there are no error bars. Soot accounts for 1% of the total INP, does this fall within the uncertainty range?**

*One artifact is excluded from further analysis, that's why it is only 199. This is now clearly stated in the text.*

*"One particle with attached gold traces was classified as an artifact and therefore excluded from further discussions."*

*In FRIDGE all samples were activated at a set of 3 temperatures (-20°C, -25°C and -30°C) and a minimum of 4 relative humidities (95% / 97% / 99% / 101%). Since measurements at higher humidities typically show a larger number of ice crystals, we have chosen the measurements cycles at RH=99% and RH=101% for the coupling procedure. RH = 95 / 97% was chosen for one sample due to cluster formation at higher RH. We added the information to the corresponding figure captions.*

*In consideration of the representativeness of our findings, we have included the following additional information.*

*"Although the number of identified INPs appears comparatively low for a campaign period of five weeks, these INPs were identified with a high degree of reliability (Sect. 2.5.2). The small number of particles identified bears the risk that individual, time-limited variations occurring randomly during the sampling periods may influence the resulting total composition to a certain degree. It should therefore be noted that the data presented below may not comprehensively reflect the main composition of the INPs over the entire campaign period. Nevertheless, it can be shown that the method provides valid results for the main groups of INPs (see confidence intervals for Fig. 9 in the supplement (Tab. S2))."*

*For the chemical composition (Fig. 9 in the revised manuscript) we have calculated the 95% confidence intervals and provide these in the supplement (Tab. S2). Of course, for the minor particle classes the differences in the abundance are within the uncertainty range, but for the major components it is possible to derive type-specific conclusions. The discussion of the results was shortened and more generalized.*

**- Figure 7 and discussion: No details comparison is made between initial size distribution of particles and size of INP, which do not enable correct estimation how size affect ice nucleation. In lines 556-566, the range referred here may be biased by simply higher initial concentrations. You need to better connect this part to size distribution measurements in lines 571-584, with statistical analysis.**

*Our INP size distribution is now compared to the particle size distribution over the whole campaign period provided by Weber (2019). The comparison with the size distribution of the total aerosol during the same campaign period shows that the maximum of our size distribution is shifted to larger diameters compared to the total aerosol. As we laid more emphasis on the methodological part of the manuscript, we chose to describe the likely enrichment of larger particles an INPs based on the size distributions presented in Weber (2019) in a more qualitative manner.*

*"In comparison to the total aerosol size distribution from the whole campaign period (Weber, 2019), the maximum of the INP size distribution is significantly shifted towards larger diameters. We hypothesize that, in addition to the primary suitability of larger particles as ice nuclei, the absence of the small volatile aerosol components (nitrates, sulfates, and volatile organics) may play a role here (see Sect. 2.6)."*

*The comparison to Lacher et al. (2021) and Worringen et al. (2015) has been rewritten and divided according to a comparison of the INP size and a comparison of the size-resolved chemical composition.*

*"Lacher et al. (2021) and Worringen et al. (2015) provide size distributions for INPs/IRs measured with different techniques at the high-altitude research station JFJ up to a size of 3 μm and 5 μm, respectively. In both studies, the highest concentration was found for IRs smaller than 0.5 μm, but the broad maximum (diameters between 1.3 μm and 5 μm) from Lacher et al. (2021) agrees reasonably well to our findings. The same is the case for particles collected with the Ice Selective Inlet by Worringen et al. (2015), which also showed a secondary maximum at 1 - 1.5 μm. The shift towards larger particle diameters in our results in comparison to the maxima from Lacher et al. (2021) and Worringen et al. (2015) may be caused by the differences in sampling and ice activation. INPs in FRIDGE are activated through deposition nucleation / condensation freezing under defined conditions, while the IRs collected from ambient air are activated under natural and even more complex conditions, including the potentially more important immersion freezing mode (Ansmann et al., 2009; Murray et al., 2012).*

*The comparison of such INP size distributions with chemical information from different methods is difficult, since in addition to the influencing factors discussed in Sect. 3.3, a possible size selection or limitation of the sampling process, and different techniques of particle sizing may also play a role. Nevertheless, the results for our main groups are in reasonable agreement with the results from Worringen et al. (2015). In our results, both the metal oxides and the few soot particles were observed at very small diameters, which is comparable to carbonaceous particles/soot and metal oxides predominantly detected in the submicron range by Worringen et al. (2015). Terrigenous particles, including silicates and Ca-rich particles, were primarily found in the larger size ranges, while our mineral components were distributed over all size ranges, with silicates domination for particles from $d_{pa} > 0.5$ μm. In contrast to Worringen et al. (2015), our C-rich particles were present over the entire size range. The reason for this is possibly that our classification scheme assigned the larger potentially biological particles as C-rich."*

**Technical corrections**

**- Line 350 "natural mineral durst" correct to dust**

*The typo has been corrected.*

**-Line 401: "This was not the case for -20°C, because Saharan primarily activate as temperatures below -20°C." reference?**

*Niemand, M., Möhler, O., Vogel, B., Vogel, H., Hoose, C., Connolly, P., Klein, H., Bingemer, H., DeMott, P., Skrotzki, J., Leisner, T.: A Particle-Surface-Area-Based Parametrization of Immersion Freezing on Desert Dust Particles, J. Atmos. Sci, 69 (10), 3077-3092, doi: 10.1175/JAS-D-11_0249.1, 2012*

*Murray, B. J., O'Sullivan, D., Atkinson, J. D., Webb, M. E.: Ice nucleation by particles immersed in supercooled cloud droplets, Chem. Soc. Rev., 41, 6519-6554, doi: 10.1039/c2cs35200a, 2012*

*The references were added to the manuscript.*

---

## Referee Report (RR1)

First of all, I would like to thank the authors for their responses to my comments and the revisions made to the manuscript. The new version places greater emphasis on the methodology, which is satisfying regarding the purpose of this paper. This new version also includes better discussion on statistical analysis. However, in my opinion, the manuscript still requires improvements for potential publication. Specifically, it appears that (1) authors did not pay enough attention to details, (2) authors can further develop Sections 2.4, 2.5 and 2.6 that are the ones actually focusing on the novel coupling method and (3) authors must imperatively improve the discussion on statistical and error analysis of the technique started in Section 2.5.2 to provide readers a real understanding of the limitations of this method and potential improvements.

(1) Attention to detail, which was a major concern in my initial review, remains insufficient. For the figures, many of them require better formatting, clarity, and descriptions (e.g., many figures contain spectra that are misaligned, inconsistent in font size, or even cut off). Additionally, several figures include elements, text, or numerical values that are not properly explained in the captions. For the main text, several terms are utilized but not carefully defined. For the references, there are inconsistencies between the style of the listed publications. These issues must be addressed to improve the overall quality and readability of the manuscript.

(2) The coupling method, which should be the main focus of the manuscript as it represents the novelty in this work, requires further development. The authors state that they have placed greater emphasis on the methodological discussion while shortening the case study, making the manuscript more suitable for AMT. While these changes are noticeable, Sections 2.4, 2.5, and 2.6, which should form the core of the manuscript, must be expanded with a more detailed discussion and further explanation of the method. Additional figures would be beneficial in guiding the reader through the methodology.

(3) The statistical analysis and evaluation of uncertainties remain limited and require further discussion. Relying on ambient measurements (the case study) for statistical analysis is, in my opinion, an inadequate approach for rigorously assessing the method. Analysing aerosols generated under controlled laboratory conditions would have been preferable. In Section 2.5.2, the authors state that, for the case study, only an average of 30% (ranging from 13% to 50%) of identified INPs with FRIDGE were analysed using SEM. Among the remaining 70%, 45% were excluded due to excess aerosol loading on the wafer, while 25% were blank positions. For the 45% related to aerosol loading, high particle concentration increases the likelihood of multiple particles within the 50 μm radius used for SEM analysis, making many positions non-analysable. However, a more in-depth discussion of this issue is needed, including an evaluation of different aerosol concentrations or sampling durations to identify possible improvements. Unfortunately, such an assessment is not feasible within the current case study framework, significantly limiting the statistical evaluation of the technique. Similarly, the issue of blank positions (25%) is not discussed at all, which is very concerning. This lack of discussion limits the transparency of the method's performance. Furthermore, the FRIDGE-SEM analysis cycle is not repeated, meaning no statistical validation through multiple similar measurements is provided.

**Specific comments**

-L.221: "information on their chemistry". It is very vague, please refer to elemental composition, as it is only what EDX provides.

-Fig.1: I strongly recommend not mixing the style between subfigures, as A1, B1, B2 and C1 are schematics while and B3, C2 and C3 are experimental results. If results are added to a subfigure, I expect them to be explained in the text and legend. In B3, what does TA, TB and TC mean? What is the x-axis? In C2, what is class A, class B, etc.? Explain why in figure C1 why you used a straight line for BSE and SE while you used wavy line for EDX.

-L.163: "ice formation can regularly be observed at temperatures at or below -30°C." I highly doubt that ice formation above -30°C is never observed. It would be very useful to include blank results in the supplementary information to support this statement.

-L.188:" it is important to keep the three laser-engraved crosses on the wafer surface visible during the FRIDGE measurement." Which ones? Please refer to figure in SI.

-L.215: "with a minimum size of 30 pixels proven to be useful" elaborate, explain why it is useful. If you refer to Schrod et al. 2016, cite the publication.

-L.218: "as the center of the detected bright area" what do you mean? Explain better what is bright.

-L.225: "To reduce this uncertainty based on an imperfect radial symmetry, the ice crystal position calculation should be performed on the basis of FRIDGE images, that show the ice crystals in a state close to activation." I don't understand this part. Do you mean that you need to take images before ice crystal formation? Or what do you refer to as "close to activation"?

-L.228: "calculated for SEM" what do you mean by calculated? The sum of all ice crystals?

-Fig.2: What does y, $R^2$, and p represent? These parameters must be explicitly explained, even if they seem obvious.

-L.245: "solid-state detector (SSD), providing the distribution of elements on the particle by backscattered electrons (BSE) giving information on homogeneous or heterogeneous distribution of elements and on inclusions." why don't you provide examples of such analysis?

-L.248:"and origin" what do you mean by origin?

-L.253: "As the internal SEM coordinate system is centered around the origin in the middle of the stage aligning the axes to the directions of mechanical movements, it is necessary to perform a coordinate transformation to link the SEM coordinates to the coordinates defined by the crosses in the previous step." I don't understand this sentence, please be more explicit and pedagogical.

-L.254: "Based on a calibration image, which indicates the marked center points from the previous ice crystal identification step." Did you consider to add a Figure to illustrate the calibration process from FRIDGE to SEM, I find it difficult to follow properly all the steps. At least in the SI?

-L.266: "In this context, a radius of 50 µm has proven to be useful." I think this can be further discussed. You can develop on the changes seen with different radius and show the statistics. As it is now, the choice of such radius sounds very vague.

-L.279: "possible particle drift during the processing in FRIDGE." Could you elaborate on this? Additionally, could particle drift also occur when transitioning from FRIDGE to SEM-EDX analysis? Is there different possibility of drift based on particle size?

-L.287, "even if the aerosol concentration is known, it is difficult to specify a suitable collection volume in advance, as the ratio of potential INPs to the total aerosol also plays a role." In L.270–285, you mentioned that SEM analysis is only feasible if no other particle is present within the 50 µm radius. This shows that the feasibility of the analysis is not a matter of how many INPs are on the filter but rather how spaced the particles are. Thus, I think information on mass loading or recommended sampling time can be provided by the authors. This information can also be accompanied with statistical analysis of occurrence of multiple particles within 50 µm radius. Furthermore, in Schrod et al. 2016, some guidelines about sampling for FRIDGE measurement is provided. As an AMT paper, I am expecting some guidelines for applying the method proposed here.

-L.291: "identification rate" you introduce a new term; a clear definition is needed.

-L.293: "INP identification rate was calculated to be 30% (ranging from 13% to 50%)." Is this based on all measurements from case study? This is crucial information; it needs further discussion.

-L.293:" identified the presence of multiple particles at 45 % of the locations (ranging from 7 % to 81 %)" why such a high variation? Is there any correlation with sampling time or aerosol concentration? You need to discussed these values.

-L.294: "While the remaining 25% (ranging from 2% to 66%) were found to be blank positions." This part also needs further clarification and discussion. How were these values determined? Does this come from particles drifting? If yes, why such a high variation?

-L.298:" In most cases, the small number of clearly identified INPs still allows general statements to be made, e.g., about the most frequently occurring characteristics of INPs" you need to discuss that further. Why would I believe you? What are most cases? Can you provide a lower limit?

-Fig. 3: The spectra require a y-axis and should be replotted, as the Si spectra appear to be cut off. The x-axis has inconsistent font sizes, making the figure unsuitable for publication in its current form. Also, I first thought the image resolution 20 x 20 µm was the size of the grid on the FRIDGE image, please add some scale to avoid confusion.

-L.312: "determine the mixing state of a particle" In cases of mixing state, how are INPs classified? Is the composition of the main particle that is assumed to be the INP?

-L.312: "surface properties" what kind of properties?

-L.327: "chemical characterization" why chemical or not elemental composition?

-L.345: "Carbonates can contain, in addition to carbon and oxygen" why on Figure 6 the ratio of signal for carbon and oxygen (C:O) is not 1:3 for carbonate ($CO_3$)?

-L.378:" Sulfates are mainly characterized by the presence of sulfur and oxygen." Similar question here, why is there not a ratio S:O of 1:4 in Figure 6?

-Fig.6: Adjust the axes, align the spectra, why does the spectra don't start from zero?

-Fig.7: The images must be numbered and explicitly referenced in the text.

-L.409: "across the three activation temperatures," and RH.

-L.412:" so the error of the concentrations given here is also in this range." Why not adding error bars on the Figure?

-L.417:" Their concentration varied between 0.1 and 1 $stdL^{-1}$ for most of the time" what is the collection volume if I compare it to the background value of FRIDGE $0.1L^{-1}$ for 100L volume sampled.

-Fig.8: The blue color is difficult to distinguish, especially when a red triangle is placed over it. Consider improving contrast for better readability. Also, in my opinion, this figure is not directly relevant to the main focus of this paper: the coupling method. This figure can easily be removed from the manuscript and be replaced by more attention to Sections 2.4, 2.5 and 2.6.

-L.429:" Overall, based on the parameters described in Sect. 2.5.2, we were able to clearly identify and characterize the associated INPs for 200 ice crystals." You mentioned in the same section that the identification rate is 30%, so you were able to characterize 200 from 600 ice crystals, no?

-L.437:" INP chemistry" be more specific.

-L.454:" Mineral components" why not proving all spectra in the SI?

-L.469:" Carbonaceous particles" why not proving all spectra in the SI?

-L.479:" Other particle classes" why not proving all spectra in the SI?

-Fig.9: I recommend labeling "during the Saharan dust event" with the letter (c) for clarity.

-L.512: Explain the calculation of the projected area diameter.

-Fig.10: This figure needs uncertainties. Please compare the amount of analysed INP with SEM-EDX compared to total INP number detected with FRIDGE.

**Technical corrections**

-L.204: "humidity settings" change to relative humidity.

-L.221: "FRDGE" please change to FRIDGE

-L.234: "These Positions" no capital letter.

-L. 242: "Environmental Scanning Electron Microscopy (ESEM)" no capital letters to keep consistent with other abbreviations in the manuscript.

-L.572: "which ich assigned to" is to ich

-L.406: "GAW" what does it stand for?

-L.582: "It has been shown, that this position calculation works reasonably well". Who showed that, you?

-L. 594: "ice-active particles" why not INP?

---

## Referee Report (RR2)

Dear authors,

Thank you for this new version of the manuscript and for the replies to my questions. In my opinion, this revised version is easier to follow, particularly due to the numerous explanations added to your methodology. I also appreciate the inclusion of the Monte Carlo simulation, which provides a statistical understanding of the search radius, and the use of bootstrapping to evaluate the relevance of particles' compositions during the campaign.

One overall aspect I still find difficult to understand is how the center of the search radius is chosen. From L212: "the software identifies the image with the highest number of ice crystals for each measurement cycle and tags the ice crystal positions in the images as the center of an area exceeding a size threshold of 30 pixels with a brightness threshold of 30 of 256 on a scale stretching from the darkest to the brightest recorded signal (Schrod et al., 2016)" I understand that the center is determined based on the 30 adjacent pixels. However, in the schematic for your simulation in Fig. 4, the center (red cross) is clearly positioned in an area that is not surrounded by 30 pixels above the brightness threshold of 30. Then, in Fig. 5, it seems that the center of the search radius corresponds to the center of the ice crystal. I find these descriptions contradictory and would appreciate further clarification, especially that the center of search radius is a key step for your method.

**Specific comments**

-L155: "However, even after thorough cleaning a small amount of ice formation can regularly be observed at temperatures at or below -30°C, constituting the background concentration and defining the limit of detection, which is in the order of 0.1 L-1 of atmospheric air for a collection volume of 100 L." It appears that ice formation on silicon wafer only appears at -30°C, so is the background concentration is only for particles that nucleate ice at this temperature or below?

-L234: "can be caused" here you employ a modal verb which suggests that condensation is only a possible explanation for the higher particle counts observed with FRIDGE compared to SEM. Is there any evidence or analysis that could strengthen this claim? In other words, is it not possible to determine with greater certainty whether condensation is indeed the cause?

-L259: "mechanical movement" can you add an explanation?

-Fig. 3b: There are 3 pictures (top left, bottom left and bottom right) with each calibration marks and 1 picture (top right) of the entire calibration system incorporation the 3 marks? Please add explanation.

-Fig.4: Did you think about integrating the pixel size of this schematic? Looking at the radius of the inner circle (25 µm) and the area around, I don't see how there could be 30 adjacent pixels with brightness above the threshold in that area. Perhaps you can increase the size of ice crystal?

-L297: "The direction of shift is randomly chosen; the distance is randomly sampled from a mirrored normal distribution with a standard deviation of some typical uncertainty assumptions." Could you further explain what you mean by typical uncertainty assumption?

-L304: "(total number of 20000 / 50000 / 100000 particles on the wafer with different INP fractions of 0.0005 / 0.001 / 0.002)" Please discuss this further and add references.

-L305:" In this case the standard deviation of the position uncertainty of 25 µm was assumed." Why did you choose this one?

-Fig 5: I am having difficulty understanding how the probability of INP correctly identified (thin black line) is equal to 1 for the smallest search radii, but at the same time the fraction of INP missed is increasing as search radii are decreasing.

-Fig.4: Why don't you integrate the pixel size of this schematic? Looking at the radius of the inner circle (25 um) and the area around, I don't believe there are 30 adjacent pixels with brightness above the threshold in that area.

-L395: "From the statistical calculation in Tab. S1 with a 95% confidence level, a limit of approximately 10 particles per group can be derived to make a reliable quantitative statement, for groups with less particles the uncertainties become large" do you mean that since the 95% confidence level for less than 10 particles spans reaches 0 as lower limit, it is not possible to make a reliable quantitative statement?

**Technical corrections**

-Fig.3: "polar coordinatesystem" change to "polar coordinate system"

-Fig 3: in the cation is mentioned "d10" but not in the figure.

-Fig 3: please add in the caption that white circle is an ice crystal.

-Fig. 4: "die" replace by "due"

-Fig. 4: "For this plot" there are several plots here, which one are you referring to?

-L.280: "orange" it is yellow, no?

-L361: "Based on the modeling, we would consider a value of around 100,000 particles on the wafer to be a good starting point, as the proportion of incorrectly identified particles increases significantly with higher particle numbers." Consider changing "good starting point" as this can be understand that 100,000 particles minimum are suitable, while higher number of particles increases uncertainty.

---

## Author Response (AR2)

**Author's response to Reviewer 1**

*First of all, we would like to thank the reviewer for reading our revised manuscript and commenting on some technical improvements.*

*In the following, the reviewer comments are written in bold and our answers in italics. Text passages from the revised manuscript are in quotation marks, modified or newly added passages are marked in green.*

**Suggested technical corrections**

**L46: Please remove the first parentheses and use "e.g.," instead**

*Changed as requested.*

*"In addition to the prevailing environmental conditions, i.e., temperature and humidity, the potential for an INP to be activated depends on individual particle properties, e.g., surface imperfections (Kiselev et al., 2016), chemical composition and specific chemical properties, crystal structure, coating (Kanji et al., 2008), etc., as well as its atmospheric processing, including potential agglomeration or pre-activation (Marcolli, 2017)."*

**L78: Change "IR" to "IRs"**

*Changed as requested.*

*"In this case, the particles are heated after separation so that the water evaporates and ice residuals (IRs) remain."*

**L96-100: Please reshape these sentences, which are confusing and might mislead the readers. One can understand that the false INPs are inevitable, however, based on the statement here, the number of false INPs seems to be several orders of magnitude more than real ones. This brings issues regarding data quality and reliability.**

*We have rewritten the sentences and adjusted the perhaps somewhat drastic statement they contain. The new section reads as follows:*

*"These methods typically analyze large numbers of INPs/IRs. However, the major challenge of all these methods is that due to the extremely low number of INPs within a sampled air volume compared to the much higher number of non-INP particles (ratio ~ $1/10^4 – 1/10^6$), the separation must be carried out with a very high accuracy. Conclusions about the chemistry of INPs may not be entirely accurate, since there is no way to distinguish particles that have been falsely separated as INPs from real INPs afterwards. There is also the risk that additional artifacts can be introduced into the INP fraction during the multi-step process. This problem is partially illustrated in the comparison of the chemical analysis of the INPs/IRs fraction by three different methods in Worringen et al. (2015)."*

**L101: Change "INP/IR" to "INPs/IRs"**

*Changed as requested.*

*"This problem is partially illustrated in the comparison of the chemical analysis of the INPs/IRs fraction by three different methods in Worringen et al. (2015)."*

**L110: Add a statement, e.g., the coupling has not yet been described in detail in previous studies**

*Changed as requested.*

*"Details of the FRIDGE method were described by Schrod et al. (2016), but the coupling has not been described in detail in previous studies."*

**L122: Change "INP" to "INPs". Go through the manuscript and revise when necessary.**

*Changed as requested (see the following sentence). We also went through the whole manuscript and changed it if necessary.*

*"Based on the definition by Cziczo et al. (2017) we refer to the identified particles as INPs, as they were activated under defined conditions after the collection of the total aerosol and not sampled as ice crystals."*

**L125: I would still argue the particles measured by SEM are IRs rather than INPs. As pointed out in the following sentence, these particles have undergone changes in FRIDGE. Such change includes ice activation and at the end of each measurement cycle, the ice crystal will evaporate before the measurement by SEM, therefore the remaining core should be IRs.**

*We can understand the reviewer's point of view. The nomenclature for INPs and IRs always depends on the perspective and is a matter of definition. For the reasons given in the text, we have decided to stick with the chosen nomenclature, but have mentioned in the text that a different perspective is also possible.*

*"However, some of the INPs analyzed with SEM may have undergone changes due to the measurement procedure in FRIDGE, and thus be announced as IRs. But we assume that these changes are of minor importance for the main INP classes that we can analyze with this method (see Sect. 2.6), which is why we have decided to continue referring to them as INPs."*

**L222: Change "x" to multiplication symbol. Go through the manuscript and revise all.**

*Changed as requested.*

**Author's response to Reviewer 3**

*First of all, we would like to thank the reviewer for reading our revised manuscript. The feedback and detailed comments helped us a lot to expand the technical chapters of the paper and to describe the method in a more focused way. Through the comments, some formalities could be improved.*

*In the following, the reviewer comments are written in bold and our answers in italics. Text passages from the revised manuscript are in quotation marks, modified or newly added passages are marked in green.*

**First of all, I would like to thank the authors for their responses to my comments and the revisions made to the manuscript. The new version places greater emphasis on the methodology, which is satisfying regarding the purpose of this paper. This new version also includes better discussion on statistical analysis. However, in my opinion, the manuscript still requires improvements for potential publication. Specifically, it appears that (1) authors did not pay enough attention to details, (2) authors can further develop Sections 2.4, 2.5 and 2.6 that are the ones actually focusing on the novel coupling method and (3) authors must imperatively improve the discussion on statistical and error analysis of the technique started in Section 2.5.2 to provide readers a real understanding of the limitations of this method and potential improvements.**

**(1) Attention to detail, which was a major concern in my initial review, remains insufficient. For the figures, many of them require better formatting, clarity, and descriptions (e.g., many figures contain spectra that are misaligned, inconsistent in font size, or even cut off). Additionally, several figures include elements, text, or numerical values that are not properly explained in the captions. For the main text, several terms are utilized but not carefully defined. For the references, there are inconsistencies between the style of the listed publications. These issues must be addressed to improve the overall quality and readability of the manuscript.**

*We revised many of the figures in terms of clarity (Fig. 1, Fig. 6, Fig. 10, Fig. 11, Fig 12 – numbers referring to the new manuscript) and formatting (Fig. 6, Fig.7, Fig. 9, Fig. 10, Fig. S3 – numbers referring to the new manuscript). We went through all figure captions, and revised them if appropriate. We tried to explain all the used elements and numerical values in the captions. We hope, that they are now clearer and better to understand.*
*We went through the whole manuscript, trying to identify terms which are not properly explained. If this was the case, we added a definition.*
*Details for individual changes can be found in the specific comments below and in the manuscript with track changes.*

*All the references were revised with a citation software and the format given by the Copernicus website. In order to maintain the clarity of the document, these changes have not been marked in the track changes.*

**(2) The coupling method, which should be the main focus of the manuscript as it represents the novelty in this work, requires further development. The authors state that they have placed greater emphasis on the methodological discussion while shortening the case study, making the manuscript more suitable for AMT. While these changes are noticeable, Sections 2.4, 2.5, and 2.6, which should form the core of the manuscript, must be expanded with a more detailed discussion and further explanation of the method. Additional figures would be beneficial in guiding the reader through the methodology.**

*The method sections have been improved and extended as suggested. The identification process (Sect. 2.4) is now described in more detail regarding the coordinate systems. A figure has been added (Fig. 3a – number referring to the new manuscript) to visualize the INP identification in the coordinate systems.*
*Also, the coordinate calibration to recover the positions in SEM (Sect. 2.5.1) is now explained more understandable, in more detail and illustrated by a new figure (Fig. 3b-d number referring to the new manuscript). A new chapter (Sect. 2.5.2 – number referring to the new manuscript) containing a numerical simulation of the effect of different parameters (total particle number, activated fraction, position uncertainty) on our INP identification process was added. This model study should give the reader a more detailed insight in the sensitivity of the method to different parameters and limitations. Also, it can help to derive recommendations for an optimal application of the method.*
*We also improved the discussion on ambiguous and blank positions as well as on the identification rates (Sect. 2.5.3 – number referring to the new manuscript).*
*For a detailed description of the individual points, please refer to the following specific comments.*

**(3) The statistical analysis and evaluation of uncertainties remain limited and require further discussion. Relying on ambient measurements (the case study) for statistical analysis is, in my opinion, an inadequate approach for rigorously assessing the method. Analysing aerosols generated under controlled laboratory conditions would have been preferable. In Section 2.5.2, the authors state that, for the case study, only an average of 30% (ranging from 13% to 50%) of identified INPs with FRIDGE were analysed using SEM. Among the remaining 70%, 45% were excluded due to excess aerosol loading on the wafer, while 25% were blank positions. For the 45% related to aerosol loading, high particle concentration increases the likelihood of multiple particles within the 50 μm radius used for SEM analysis, making many positions non-analysable. However, a more in-depth discussion of this issue is needed, including an evaluation of different aerosol concentrations or sampling durations to identify possible improvements. Unfortunately, such an assessment is not feasible within the current case study framework, significantly limiting the statistical evaluation of the technique. Similarly, the issue of blank positions (25%) is not discussed at all, which is very concerning. This lack of discussion limits the transparency of the method's performance. Furthermore, the FRIDGE-SEM analysis cycle is not repeated, meaning no statistical validation through multiple similar measurements is provided.**

*For the statistical analysis of the method and the influence of different parameters (total particle numbers on the wafer, activated fraction and position uncertainty) on our identification process, we added a model study (Sect. 2.5.2 – number referring to the new manuscript), as the case study is not perfectly suited for such an evaluation as you already noted. The discussion on multiple particle and blank positions has been extended (Sect. 2.5.3 – number referring to the new manuscript).*

*About the repetition of the coupling experiment, we would like to say the following:*
*The individual steps of the coupling procedure (FRIDGE, identification of ice crystal center points, recovery in SEM) were subjected to replication experiments. Schrod et al. (2016) showed that the average variation of the ice crystal number in FRIDGE is about 20%. The identification of ice crystal origins and coordinate calculation by our counting algorithm was also repeated. However, as this is an automated process, it always provides the same output with the same input, in this case the same coordinates.*
*The position of the coordinates in the SEM depends essentially on the calibration points. In the manuscript we give a value of 20 μm for the inaccuracy due to the limited resolution of the FRIDGE images. Repeated calibration on the basis of the calibration picture resulted in a deviation of ± 10 μm. As this deviation does not exceed the inaccuracy due to image quality, it was not considered further in the manuscript. In general, each new calibration of the wafer results in a slight change in the position of the calibration points; in random tests, the particles were found in a slightly different position to the identified coordinate.*
*To repeat the whole experiment procedure is not that easy, as it is not simply transferring the wafers between the two instruments. The oil from the FRIDGE experiment has to be removed, as it would otherwise affect the EDX analysis in SEM. This process is carried out manually using ethanol and special laboratory wipes, but is, however, a source of contamination. While possible lint on the wafer can be clearly identified as artifacts in the SEM, they can cause a bias in the following FRIDGE activation cycle. Another problem may be caused by the SEM/EDX analysis. Although SEM/EDX is generally regarded as a non-destructive method, ice nucleation properties react quite sensitively to the surface properties, which is why these can be altered by the introduction of electrons.*
*As a consequence, from our point of view, there is less potential for errors and it is more representative to repeat the individual steps than the whole multiple step procedure.*

*With regard to potential laboratory experiments we would like to state the following:*
*We can understand the reviewer's point of view, that laboratory experiments may be a good chance to evaluate a method. For the coupling, our aim was to develop and evaluate the method as closely as possible to field conditions. As a typical problem with ice nucleation measurements is that the devices work well under laboratory conditions but have problems in the field under real conditions. In many cases, field and laboratory measurements are also not directly comparable. The FRIDGE chamber itself has been subjected to many evaluation experiments over the years.*

**Specific comments**

-**L.221: "information on their chemistry". It is very vague, please refer to elemental composition, as it is only what EDX provides.**

*Changed as requested.*

*"The activated INPs can subsequently be characterized by SEM to gain information on their elemental composition, morphology and size (Fig. 1C)."*

-**Fig.1: I strongly recommend not mixing the style between subfigures, as A1, B1, B2 and C1 are schematics while and B3, C2 and C3 are experimental results. If results are added to a subfigure, I expect them to be explained in the text and legend. In B3, what does TA, TB and TC mean? What is the x-axis? In C2, what is class A, class B, etc.? Explain why in figure C1 why you used a straight line for BSE and SE while you used wavy line for EDX.**

*A1, B1 and C1 are schematic drawings of the three instrumental steps of the procedure (sampling, FRIDGE and SEM). In order to give the reader a meaningful overview of our method, B2 and C2 show exemplary results of a real analysis. B3 and C3 should be used to give the reader an idea how the overall result of the method looks like. These graphs are NOT based on real data and are therefore also considered to be schematic, which is why the legend is also schematic. TA, TB and TC are placeholders for different activation temperatures, similarly Class A, Class B etc. in C3 are placeholders for different INP classes. The date is shown on the x-axis in B3.*
*In the SEM, the SE and BSE detectors map the electron contrast by secondary/backscattered electrons (straight lines), while the EDX detector registers electromagnetic radiation (wavy line).*
*However, as a result of this discussion, Fig. 1 has been simplified. Only the schematic drawings of the aerosol collection, ice activation in FRIDGE and the analysis in SEM are now included. The presentation of the results has been verbalized for the sake of clarity.*

-**L.188:" it is important to keep the three laser-engraved crosses on the wafer surface visible during the FRIDGE measurement." Which ones? Please refer to figure in SI.**

*It was already mentioned that there are three laser-engraved crosses on the wafer surface in Sect. 2.1., where the substrate is described. For clarity we have added a reference to Fig. 1 and the FRIDGE picture in the supplement (Fig S1).*

*„For coupling the INP activation experiment to the single particle analysis by SEM, it is important to keep the three laser-engraved crosses on the wafer surface visible during the FRIDGE measurement (see Fig.1 and Fig. S1 from the supplement). "*

-**L.215: "with a minimum size of 30 pixels proven to be useful" elaborate, explain why it is useful. If you refer to Schrod et al. 2016, cite the publication.**

*In fact, our ice crystal identification algorithm uses the basic parameters of the FRIDGE INP counting software, which are given in Schrod et al. (2016). As we modified this section more extensively, the thresholds are given now a bit later in the text (see next comment). There, we added also the reference to Schrod et al. (2016).*

-**L.218: "as the center of the detected bright area" what do you mean? Explain better what is bright.**

*Bright area refers to the accumulation of pixels that exceed a previously defined brightness threshold and are therefore recognized by the software as ice crystals. We have adapted the wording accordingly.*

*"The software identifies the image with the highest number of ice crystals for each measurement cycle and tags the ice crystal positions in the images as the center of an area exceeding a size threshold of 30 pixels with a brightness threshold of 30 of 256 on a scale stretching from the darkest to the brightest recorded signal (Schrod et al. 2016). If two adjacent ice crystals are separated by at least one pixel that falls below the brightness threshold, ImageJ detects two separate areas and determines a center point for each of these areas. If this is not the case, a center point is determined for the entire area."*

**-L.225: "To reduce this uncertainty based on an imperfect radial symmetry, the ice crystal position calculation should be performed on the basis of FRIDGE images, that show the ice crystals in a state close to activation." I don't understand this part. Do you mean that you need to take images before ice crystal formation? Or what do you refer to as "close to activation"?**

*In our case, "close to activation" means that ice growth has already begun, but the crystals have not yet grown large. We have adapted the wording accordingly.*

*"To reduce this uncertainty based on an imperfect radial symmetry, the ice crystal position identification should be performed on the basis of FRIDGE images showing the ice crystals in a state shortly after the initial activation."*

**-L.228: "calculated for SEM" what do you mean by calculated? The sum of all ice crystals?**

*The wording was misleading in this case. It refers to the positions that the software has "calculated", i.e., identified. We have changed the wording slightly to make it clearer.*

*"Figure 2 shows a comparison between the number of ice crystals counted by FRIDGE (parameters from Schrod et al. (2016)) and the number of ice crystal positions identified for the SEM analysis by the image analysis procedure described above."*

**-Fig.2: What does y, R2, and p represent? These parameters must be explicitly explained, even if they seem obvious.**

*We added explanations to the capture.*

*"Figure 2: Comparison of ice crystal numbers counted by FRIDGE with the number of identified positions for SEM analysis by the image analysis software. The linear regression is represented by the black line, the 1:1-line is shown in red. The regression equation relates the number of positions identified for SEM (y) to the ice crystal numbers counted by FRIDGE (x). It is given along with its coefficient of determination ($R^2$) and its p-value."*

**-L.245: "solid-state detector (SSD), providing the distribution of elements on the particle by backscattered electrons (BSE) giving information on homogeneous or heterogeneous distribution of elements and on inclusions." why don't you provide examples of such analysis?**

*We provide examples for heterogeneous element distributions later in Fig. 6 and Fig 7(numbers referring to the new manuscript). At this point we only want to describe shortly the basics of SEM and some relevant capabilities of the technique.*

**-L.248:"and origin" what do you mean by origin?**

*Here origin was meant in terms of source. We changed the wording accordingly.*

*"The EDX provides an elemental composition of an individual particle, which can be used to attribute the analyzed particles to different classes of compositions and sources."*

**-L.253: "As the internal SEM coordinate system is centered around the origin in the middle of the stage aligning the axes to the directions of mechanical movements, it is necessary to perform a coordinate transformation to link the SEM coordinates to the coordinates defined by the crosses in the previous step." I don't understand this sentence, please be more explicit and pedagogical.**

*The whole section was reformulated. We tried to be more precise in the explanation using a new figure illustrating the different coordinate systems (for more details on the figure, see the next comment).*

*"To find the identified position for each ice crystal / INP by SEM, its normalized coordinates $[X_n/Y_n]$ (Fig. 3a) from the ice crystal identification step (Sect. 2.4) must be converted to instrument specific coordinates $[X_{ESEM}/Y_{ESEM}]$. This is necessary because the internal SEM coordinate system is a Cartesian coordinate system centered on the middle of the stage, with the x and y axes aligned to the directions of mechanical movements (Fig. 3c). Based on a calibration image (Fig. 3b), showing the defined calibration points from the previous step (Sect. 2.4), these calibration points are obtained in the SEM coordinate system. Based on the internal SEM coordinates for the calibration points, all INP coordinates $[X_n/Y_n]$ can be converted to internal SEM coordinates $[X_{ESEM}/Y_{ESEM}]$. It is highly important to locate these calibration points with the highest possible precision, since the position of each ice crystal in the subsequent analysis is calculated based on this calibration. Manual calibration provides*

*the highest precision, as the systems use different physical imaging processes yielding largely different image resolutions and contrast mechanisms. The high magnification of the electron microscope (Fig. 3d) in conjunction with the surface sensitivity of electron emission, in contrast to the limited resolution of the FRIDGE calibration image (Fig. 3b) produced by visible light reflection, impair any precise automated calibration. Due to the limited resolution of the FRIDGE images of about $20 \times 20$ μm, the calibration leads to a positioning uncertainty of the same scale."*

**-L.254: "Based on a calibration image, which indicates the marked center points from the previous ice crystal identification step." Did you consider to add a Figure to illustrate the calibration process from FRIDGE to SEM, I find it difficult to follow properly all the steps. At least in the SI?**

*We added a new figure (Fig 3 – number referring to the new manuscript) showing the coordinate system for the ice crystal identification step (Fig 3a), a calibration image (Fig. 3b), the Wafer on the SEM stage with the cartesian SEM coordinate system (Fig. 3c) and how a calibration crosse looks like in the SEM (Fig. 3d). We hope, that this figure helps the reader to understand the coordinate calibration step.*

**-L.266: "In this context, a radius of 50 μm has proven to be useful." I think this can be further discussed. You can develop on the changes seen with different radius and show the statistics. As it is now, the choice of such radius sounds very vague.**

*The dependence of the search radius on various parameters such as positioning accuracy, number of particles on the wafer and INP fraction was investigated in a model study (Sect. 2.5.2 - number referring to the new manuscript). The results show a clear variability of the optimal radius, especially with variation of the positioning accuracy and the total substrate loading.*
*Besides the discussion in the new modeling section, the sentence in Sect. 2.5.3 (new manuscript) was also revised.*

*"Based on the simulations and empirical values, a radius of 50 μm around the identified coordinate was chosen as a standard search radius for the following INP identification."*

**-L.279: "possible particle drift during the processing in FRIDGE." Could you elaborate on this? Additionally, could particle drift also occur when transitioning from FRIDGE to SEM-EDX analysis? Is there different possibility of drift based on particle size?**

*(We decided to change the notation from particle drift to particle shift to avoid confusion with the drift problem sometimes encountered in SEM, which has other physical reasons.)*

*Particle shift in FRIDGE means a possible displacement of the particle due to ice growth or sublimation of the ice after a measurement. Such shifts are generally possible and cannot be excluded, but their extent and influence on our method can be reduced by various arrangements, e.g., by limiting ice crystal growth and avoiding the liquid phase in FRIDGE (mentioned in Sect. 2.3). Furthermore, if the identification of the coordinates for the ice crystal centers is based on the last measurement series from FRIDGE, multiple displacement due to repeated ice growth is excluded. Whether a potential shift is size dependent has not yet been investigated, and might be difficult due to the unknown original position. We have included a section on the influence of potential particle shift on the results in the discussion of identification rates (Sect. 2.5.3 – number referring to the new manuscript).*

*"In the case of particle shift being the reason for the blank position, there is a potential for a bias in particle frequencies or size distribution, if the potential for a particle to shift would correlate with particle class or size."*

*Of course, a particle shift can also occur during the transfer of samples from the FRIDGE to the SEM and cannot be completely ruled out. We do everything possible to avoid this (avoiding tilting, shaking, etc.).*

**-L.287, "even if the aerosol concentration is known, it is difficult to specify a suitable collection volume in advance, as the ratio of potential INPs to the total aerosol also plays a role." In L.270–285, you mentioned that SEM analysis is only feasible if no other particle is present within the 50 μm radius. This shows that the feasibility of the analysis is not a matter of how many INPs are on the filter but rather how spaced the particles are. Thus, I think information on mass loading or recommended sampling time can be provided by the authors. This information can also be accompanied with statistical analysis of occurrence of multiple particles within 50 μm radius. Furthermore, in Schrod et al. 2016, some guidelines about sampling for FRIDGE measurement is provided. As an AMT paper, I am expecting some guidelines for applying the method proposed here.**

*While it is true that the total amount of particles on the wafer is crucial for the unambiguous assignment of individual INPs in the SEM (this can be also seen from the model), the absolute number of INPs also plays a role, e.g., for counting statistics at the lower end or ice-coverage at the upper. As the absolute number of INPs on the wafer depends on the proportion of INPs in the total aerosol, which is typically unknown prior to sampling, a compromise must be found between an amount of aerosol that allows for the unambiguous identification of as many particles as possible and a sufficient number of INPs to be deposited on the wafer. The whole section on discussing sampling parameters has been revised (including guidelines for applying the method) and now reads as follows.*

*"The total number of INPs that can be unambiguously attributed to an ice crystal origin is significantly influenced by the total wafer loading (INPs and non-INPs), which is determined by the sampling parameters (e.g., flow rate, sampling time, deposition efficiency) in combination with the aerosol concentration and aerosol properties. Even if the aerosol concentration is known, it is difficult to specify a suitable collection volume in advance due to the priori unknown fraction of INPs. Besides the total number of particles on the wafer, which is decisive for the chance to identify one particle in the defined radius and influences significantly on the ratio of falsely identified INPs (see modelling data in the supplement – Fig. S2), the ratio of potential INPs is also important to ensure that enough INPs are deposited on the wafer for suitable counting statistics. Based on the modeling, we would consider a value of around 100,000 particles on the wafer to be a good starting point, as the proportion of incorrectly identified particles increases significantly with higher particle numbers. In the case of lower particle numbers, the number of INPs on the wafer may be too low depending on the INP fraction, in the case of higher loadings, the fraction of ambiguous positions increases.*
*To obtain the best results, we recommend to determine the particle concentration in the atmosphere in parallel to the collection and then perform a quick analysis of the wafers in FRIDGE to calculate the proportion of INPs in the total aerosol. Based on this proportion the sampling parameter can be adjusted to balance a sufficient number of INPs on the wafer but prevent an overload."*

**-L.291: "identification rate" you introduce a new term; a clear definition is needed.**

*We added a definition of the identification rate.*

*"The identification rate is defined as the proportion of ice crystal coordinates examined by SEM to which an unambiguous INP could be assigned."*

**-L.293: "INP identification rate was calculated to be 30% (ranging from 13% to 50%)." Is this based on all measurements from case study? This is crucial information; it needs further discussion.**

*Yes, this value is based on the 14 analyzed samples from the JFJ campaign. We added a few words to clarify this.*

*"The average INP identification rate for the JFJ samples (based on 14 analyzed samples) was calculated to be 30%.). The large variation from 13 % to 50 % for the individual samples is an expected consequence of the different wafer loadings of the individual samples, as well as of the partly low counting statistics."*

**-L.293:" identified the presence of multiple particles at 45 % of the locations (ranging from 7 % to 81 %)" why such a high variation? Is there any correlation with sampling time or aerosol concentration? You need to discussed these values.**

*Generally, it can be expected that the number of positions with multiple particles depends on the total wafer loading, as previously mentioned in the text and seen from the model: If there are few particles on the wafer, there will be few multiple particle positions. A greater number of particles on the wafer will increase the number of multiple particle positions.*
*However, a correlation with sampling time/volume, or particle concentration / number of particles on the wafer is not a straight forward task. Since we have a loss of volatile components, the airborne particle concentration may not be directly relevant for the number of particles deposited on our wafer. The spatial distribution of the particles on the wafer also plays a role. Normally, the particles should be very uniformly distributed over the entire surface of the substrate. However, some areas of the wafer surface may have a heterogeneous distribution of particles. In such areas, the chance of getting a multiple particle position is higher than on the rest of the substrate.*
*We have checked possible correlations for the samples from the JFJ campaign and did not find a good correlation for the total particle concentration for particles > 0.1 μm and > 0.5 μm.*

*"In addition, the study identified the presence of multiple particles at 45 % of the locations, ranging from 7 % to 81 % for the individual samples. Although a correlation of multiple particle positions with the total number of particles on the substrate surface is suspected, such a correlation is not obtainable directly, as the particle distribution on the wafer and the possible loss of volatile components (see Sect. 2.6) have an impact."*

**-L.294: "While the remaining 25% (ranging from 2% to 66%) were found to be blank positions." This part also needs further clarification and discussion. How were these values determined? Does this come from particles drifting? If yes, why such a high variation?**

*The presence of a blank position can be attributed to several factors. Firstly, particle shift, defined as the displacement of particles during the process of ice growth, can result in a blank position if the particle ceases to be at the center of the ice crystal. Second, erroneous identified coordinates can also lead to a blank position if the crystals did not grow symmetrically and the origin was not identified correctly. Third, the limitation of the search radius to 50 µm can also be a contributing factor. In general, the optimal search radius in SEM increases with increasing uncertainties. In contrast, this radius decreases with increasing wafer load. As previously mentioned, the value of 50 µm represents a compromise between the consideration of the uncertainties and the uniqueness of the position. Therefore, it is possible that a position is classified as a blank, but would have a clearly assignable INP at, say, 80 µm. In such cases, extending the radius to 100 µm might prove beneficial, if the sample loading is sufficiently low.*
*We have added the missing third point to the description of a blank position.*

*"(3) In the absence of a particle within the 50 µm radius, these blank positions are disregarded. A blank position may be the consequence of possible particle shift during the processing in FRIDGE (discussed in Sect. 2.3), or the result of an erroneous calculation of the ice crystal origin (discussed in Sect. 2.4). As it can be seen from the model simulations, an incorrectly selected search radius can also lead to a higher number of blank positions. In cases where the substrate loading is low, it may be beneficial to increase the search radius in the case of a blank position in order to increase the number of identified INPs."*

*The influence of the different factors for a blank position on the results is discussed in the section on identification rates.*

*"The remaining 25 % (ranging in the extremes from 2 % to 66 %) were found to be blank positions, for the reasons discussed above. In case of a blank position due to an uncertain ice crystal position in connection with the search radius restriction, the overall result remains representative. No bias is expected in the relative contribution of individual particle classes. In the case of particle shift being the reason for a blank position, there is a potential for a bias in particle frequencies or size distribution, if the potential for a particle to shift would correlate with particle class or size. Similarly, as volatile components and thin films cannot be detected in the electron microscope reliably (Sect. 2.6), a blank position might be detected here, although they could have triggered ice formation in FRIDGE."*

**-L.298:" In most cases, the small number of clearly identified INPs still allows general statements to be made, e.g., about the most frequently occurring characteristics of INPs" you need to discuss that further. Why would I believe you? What are most cases? Can you provide a lower limit?**

*The uncertainty of our classification can be derived from the counting statistics in the supplement (Tab. S1 – number referring to the new manuscript). Judging from the statistics with 95% confidence interval, quantitative statements about the relative frequency of a particle class are reliably possible starting from approx. 10 particles per group. For groups with less particles, the statistical uncertainties become larger.*
*In our specific case, the counting statistics with 95% confidence interval (Table S1) show that for the SDE with only 70 analyzed particles in total, a quantitative statement is only possible for the main component (Al-rich / aluminosilicates), whereas for the non-SDE period with 129 analyzed particles, a statement about 4 groups (aluminosilicates / Al-rich, carbonates, C-rich and mixtures) is possible due to the higher number of identified particles.*

*"Based on the uncertainties and assumptions discussed, many positions and potential INPs are excluded from further analysis. As a result, the number of identified INPs per sample is limited and typically low in contrast to the grown ice crystals in FRIDGE. These INPs, however, are accurately identified as we know that ice formation has taken place on the substrate at their position. This allows for conclusions on the analyzed INPs within the limits of their counting statistics and depending on the significance level, even for a small number of identified INPs. From the statistical calculation in Tab. S1 with a 95% confidence level, a limit of approximately 10 particles per group can be derived to make a reliable quantitative statement, for groups with less particles the uncertainties*

*become large. Further considerations of the limits of certain statements must depend on the particular statement and should employ common statistical approaches for counting statistics and compositional data."*

**-Fig. 3: The spectra require a y-axis and should be replotted, as the Si spectra appear to be cut off. The x-axis has inconsistent font sizes, making the figure unsuitable for publication in its current form. Also, I first thought the image resolution 20 x 20 μm was the size of the grid on the FRIDGE image, please add some scale to avoid confusion.**

*The figure has been improved as follows:*
*A caption has been added to the FRIDGE image stating that it is a FRIDGE image with ice crystals. In addition, the ice crystals have been numbered so that they can be directly linked to the corresponding identified INPs. A legend for the electron microscopy result has also been added for better understanding, as well as a scale.*
*The axis labeling of the spectra has been made consistent, a y-axis has been added. The figure is intended to give the reader an overview of the method, so the images and spectra have a more symbolic character.*
*Our samples are collected on Si wafers, so the Si peak is ubiquitous in our spectra, and it is extremely high. Therefore, the peak is cut off, which is common practice to show relevant data.*

**-L.312: "determine the mixing state of a particle" In cases of mixing state, how are INPs classified? Is the composition of the main particle that is assumed to be the INP?**

*In the case of an INP consisting of a main particle showing small inclusions or accretions (Fig. 7 – number referring to the new manuscript), the classification is determined by the main particle. However, if the INP obviously consists of several components lying next to each other (Fig. 6 INP4 – number referring to the new manuscript), the particle is considered as a mixture.*
*We added a sentence to clarify, how we proceed with mixed particles.*

*"In case of a main particle with small inclusions (as shown in Fig. 7 and Fig. 10d, for example) the composition of the main particle determines the classification. If a particle clearly consists of several individual components (see Fig. 6 INP4, for example), it will be classified as a mixture."*

**-L.312: "surface properties" what kind of properties?**

*In general, we can identify surface properties such as cracks, smooth or rough structures and edges in SEM. Even if we cannot say exactly at which point the ice nucleation starts. We have decided to mention them here anyway as a general possibility of SEM, even if they are not the focus of this study.*

**-L327: "chemical characterization" why chemical or not elemental composition?**

*In general: When referring to the results of the EDX analysis, we have chosen to use the term "elemental composition" instead of "chemical composition", as this is the result obtained from the analysis for each individual INP. This is done to illustrate that, from this point on, we assume that the detected elements form a chemical compound (and not for example a random mixture). Therefore, when we refer to the different classes of INPs, we prefer stick to 'chemical composition'.*

**-L345: "Carbonates can contain, in addition to carbon and oxygen" why on Figure 6 the ratio of signal for carbon and oxygen (C:O) is not 1:3 for carbonate (CO3)?**
**-L378:" Sulfates are mainly characterized by the presence of sulfur and oxygen." Similar question here, why is there not a ratio S:O of 1:4 in Figure 6?**

*In our spectra, the peak area, even after background subtraction, is not directly proportional to the element concentration, as the peaks are – besides the element contents -impacted by different excitation rates, X-ray absorption and secondary fluorescence in the particle matrix. This is particularly the case for elements with low X-ray energies < 1 keV. The problem eases at higher energies. For example, quantifying the element ratios for the Ca-sulfate from Fig.10 (new manuscript) leads to a 1:1 ratio (atom%) of Ca and S.*

**-Fig.6: Adjust the axes, align the spectra, why does the spectra don't start from zero?**

*Changed as requested, added the zero.*

**-Fig.7: The images must be numbered and explicitly referenced in the text.**

*The images are now labeled and referenced in the figure caption. We have also added a reference to distinct pictures in the text in certain places.*

*"Figure 10: Overview of representative EDX spectra and corresponding SEM images for the defined INP classes grouped as mineral components (aluminosilicate / Al-rich (a), Ca-carbonate (b), silicon dioxide (c)), carbonaceous particles (biological particles (d), soot (e), C-rich (f)) and other particle classes (Ca-sulfate (g), metal oxide (h), mixtures (i))."*

**-L.409: "across the three activation temperatures," and RH.**

*The relative humidity is RH = 99 % for each of the three activation temperatures shown. It is mentioned in the figure caption (Fig. S3 - number referring to the new manuscript).*

**-L.412:" so the error of the concentrations given here is also in this range." Why not adding error bars on the Figure?**

*With this figure, we would like to give the reader a brief overview of the FRIDGE data from the campaign. As the graph already contains a lot of data points and information, we believe it would be too confusing if error bars were also shown for each individual data point. We have therefore decided to only mention the error as a value in the text. For clarification, we added the error also to the figure caption. The figure has been moved to the supplement (see following comment on Fig. 8).*

*"Figure S3: INP concentrations (deposition nucleation / condensation mode freezing) calculated from the FRIDGE measurements at RH = 99 %. The concentration for each sample is calculated on the basis of one measurement. For days with more than one sample, an average value was calculated. Days with analyzed samples are indicated by triangles and the corresponding sample numbers. The 5-day running average concentration is shown by the dotted lines (the figure is adapted from Weber (2019)). The error of the concentrations given here is 20% approx., based on the relative error of the counting uncertainty for individual measurements from Schrod et al. (2016)."*

**-L.417:" Their concentration varied between 0.1 and 1 stdL-1 for most of the time" what is the collection volume if I compare it to the background value of FRIDGE 0.1L.1 for 100L volume sampled.**

*For the campaign, the collection volume of the individual samples ranged from 115 to 306 liters.*
*Since particularly low INP concentrations were expected in the free troposphere at JFJ in advance, special attention was paid to the cleanliness of the sample substrates. The blank value for the campaign was therefore particularly low (less than 3 ice crystals at -30°C).*
*When calculating the INP concentration, the blank values are directly subtracted from the number of grown ice crystals. The concentration shown in Fig. 10 is already blank-corrected.*

**-Fig.8: The blue color is difficult to distinguish, especially when a red triangle is placed over it. Consider improving contrast for better readability. Also, in my opinion, this figure is not directly relevant to the main focus of this paper: the coupling method. This figure can easily be removed from the manuscript and be replaced by more attention to Sections 2.4, 2.5 and 2.6.**

*We changed the colors a bit and increased the size of the datapoints. However, in order to focus even more on the coupling, we have decided to move the figure to the supplement.*

**-L.429:" Overall, based on the parameters described in Sect. 2.5.2, we were able to clearly identify and characterize the associated INPs for 200 ice crystals." You mentioned in the same section that the identification rate is 30%, so you were able to characterize 200 from 600 ice crystals, no?**

*The identification rate indicates that a unique INP could be assigned to 30% of the analyzed positions. When analyzing the JFJ samples, only a fraction of the area, a defined section around the center of the wafer (square of 2x2 cm) was inspected.*

*"Overall, based on the parameters described in Sect. 2, we were able to clearly identify and characterize the associated INPs for 200 ice crystals, which corresponds to 30 % of the ice crystals positions analyzed (Fig. 12b). For the remaining 70 % we were unable to make a statement. While the multiple particle positions have no effect on the proportion of particle classes (Fig. 11) or size (Fig. 12a), the blank positions can cause a bias (discussed in Sect. 2.5.3). In this campaign, only a square with a side length of 2 cm in the center of the substrate was analyzed*

*by electron microscopy. Since the area analyzed in SEM corresponds to roughly half of the area considered in FRIDGE, the 200 INPs identified represent about 15% of the total sites activated in FRIDGE. This limitation has no influence on the individual chemical fractions or on the size distribution of the INPs, as the area was chosen arbitrarily and a homogeneous distribution of the particle groups on the wafer can be assumed."*

.

**-L.437:" INP chemistry" be more specific.**

*Changed to "chemical composition of the INPs".*

*The same applies here as described above: When referring to the results of the EDX analysis, we have chosen to use the term "elemental composition" instead of "chemical composition", as this is the result obtained from the analysis for each individual INP. This is done to illustrate that, from this point on, we assume that the detected elements form a chemical compound (and not for example a random mixture). Therefore, when we refer to the different classes of INPs, we prefer stick to 'chemical composition'.*

**-L.454:" Mineral components" why not proving all spectra in the SI?**
**-L.469:" Carbonaceous particles" why not proving all spectra in the SI?**
**-L.479:" Other particle classes" why not proving all spectra in the SI?**

*All spectra can only be read and edited with the internal Oxford Aztec EDX software. To publish a spectrum, it must be exported, converted and edited individually. In our opinion, the effort involved is too great to publish a complete data set in the supplement.*
*If the data is required for a specific purpose, it can be requested. This allows us to customize the data to specific interests.*

**-Fig.9: I recommend labeling "during the Saharan dust event" with the letter (c) for clarity.**

*Changed as requested.*

**-L.512: Explain the calculation of the projected area diameter.**

*We added the explanation how the projected area diameter is calculated.*

*"Therefore, the particle is regarded as an ellipse and the dimensions of its major and minor axis are determined in order to subsequently calculate the diameter of an equivalent circle, which is referred to as $d_{pa}$."*

**-Fig.10: This figure needs uncertainties. Please compare the amount of analysed INP with SEM-EDX compared to total INP number detected with FRIDGE.**

*We included the uncertainties in the supplement (Tab. S2 – number referring to the new manuscript). It is now stated at the beginning of the case study, how much INPs were analyzed and they are related to the total number of INPs grown in FRIDGE (Sect. 3.2). For a better understanding, we have added a second graph (Fig. 12 b – number referring to the new manuscript) which shows the proportion of the ice crystals examined that correspond to the graph in Fig. 12a.*

*„Overall, based on the parameters described in Sect. 2, we were able to clearly identify and characterize the associated INPs for 200 ice crystals, which corresponds to 30 % of the ice crystals positions analyzed (Fig. 12b). For the remaining 70 % we were unable to make a statement. While the multiple particle positions have no effect on the proportion of particle classes (Fig. 11) or size (Fig. 12a), the blank positions can cause a bias (discussed in Sect. 2.5.3). In this campaign, only a square with a side length of 2 cm in the center of the substrate was analyzed by electron microscopy. Since the area analyzed in SEM corresponds to roughly half of the area considered in FRIDGE, the 200 INPs identified represent about 15% of the total sites activated in FRIDGE. This limitation has no influence on the individual chemical fractions or on the size distribution of the INPs, as the area was chosen arbitrarily and a homogeneous distribution of the particle groups on the wafer can be assumed. "*

**Technical corrections**

**-L.204: "humidity settings" change to relative humidity.**

*Changed as requested.*

*"As a routine, the wafers are measured in 12 cycles combining three temperatures (T = -20 °C, -25 °C, -30 °C) and four relative humidities (RH) (RH = 95 %, 97 %, 99 %, 101 %)."*

**-L.221: "FRDGE" please change to FRIDGE**

*Changed as requested.*

*"It can be assumed that this coordinate represents the position of the corresponding INP, since an approximately radially symmetric ice crystal growth can be observed in the range of the selected activation conditions in FRIDGE."*

**-L.234: "These Positions" no capital letter.**

*Changed as requested.*

*These positions are excluded from further analysis.*

**-L. 242: "Environmental Scanning Electron Microscopy (ESEM)" no capital letters to keep consistent with other abbreviations in the manuscript.**

*Changed as requested.*

*"A Quanta 200 FEG environmental scanning electron microscope (ESEM) by FEI (Field Electron and Ion Company; Eindhoven, Netherlands) coupled to an energy dispersive X-ray detector (EDX) (EDAX, AMETEK, Tilburg, Netherlands) was used for analysis."*

**-L.572: "which ich assigned to" is to ich**

*Changed as requested.*

*"The metal oxides are characterized by the presence of oxygen and a corresponding metal (except Al, which is assigned to the previous aluminosilicate / Al-rich particles group)."*

**-L.406: "GAW" what does it stand for?**

*GAW stands for "global atmosphere watch" which is a program of WMO which focusses on the monitoring and understanding of atmospheric composition. As the information is not necessary for the reader to understand the results, we removed the GAW in this case.*

*"Aerosol sampling for the FRIDGE experiment was conducted downstream of the total inlet (Lacher et al., 2021)."*

**-L.582: "It has been shown, that this position calculation works reasonably well". Who showed that, you?**

*This statement refers to Fig. 2, so yes, we have shown that. However, we adapted the wording.*

*"It has been shown, that our position identification algorithm works reasonably well, with a negligible number of incorrect positions due to condensation or coalesced ice crystals."*

**-L. 594: "ice-active particles" why not INP?**

*Changed as requested.*

*"Although the method has some drawbacks and uncertainties, it enables high accuracy in the identification and in this way physico-chemical characterization of individual INPs."*

---

## Author Response (AR3)

*In the following, the reviewer comments are written in bold and our answers in italics. Text passages from the revised manuscript are in quotation marks, modified or newly added passages are marked in green.*

**Author's response to Reviewer 3**

*First of all, we would like to thank the reviewer for reading our revised manuscript and providing feedback on ambiguities, which helped us to improve the new sections and figures of the revised manuscript.*

**Dear authors,**
**Thank you for this new version of the manuscript and for the replies to my questions. In my opinion, this revised version is easier to follow, particularly due to the numerous explanations added to your methodology. I also appreciate the inclusion of the Monte Carlo simulation, which provides a statistical understanding of the search radius, and the use of bootstrapping to evaluate the relevance of particles' compositions during the campaign.**
**One overall aspect I still find difficult to understand is how the center of the search radius is chosen. From L212: "the software identifies the image with the highest number of ice crystals for each measurement cycle and tags the ice crystal positions in the images as the center of an area exceeding a size threshold of 30 pixels with a brightness threshold of 30 of 256 on a scale stretching from the darkest to the brightest recorded signal (Schrod et al., 2016)" I understand that the center is determined based on the 30 adjacent pixels. However, in the schematic for your simulation in Fig. 4, the center (red cross) is clearly positioned in an area that is not surrounded by 30 pixels above the brightness threshold of 30. Then, in Fig. 5, it seems that the center of the search radius corresponds to the center of the ice crystal. I find these descriptions contradictory and would appreciate further clarification, especially that the center of search radius is a key step for your method.**

*The software of FRIDGE determines the position of the INP as the center of the ice crystal it detects, as you cited from the manuscript. That is symbolized as the blue circle marker in the schematic in Fig. 4, which is in the center of the grey shaded area. The center of the search radius (red cross) is the location reported later to the SEM analysis. It is not necessarily the true position of the INP, as it includes the various errors as described in the text.*
*To clarify that, we have modified the caption of Fig. 4.*

*„Figure 4: Conceptual model for the simulation of the INP-recovery from FRIDGE in SEM. The blue circle is the true location of the INP, the yellow circles are non-INPs. The red cross is the search origin position in SEM. It deviates from the true INP position due to the errors by INP shift, unprecise ice crystal center determination and the re-positioning uncertainty due to calibration mismatch and mechanical backlash. The circles show search radii of 25, 50 and 75 μm from the red search origin position. The grid shows the approximate pixel size of the optical camera of FRIDGE."*

*We have added the relevant information on the search radius origin to the caption of Fig. 5, too.*

*Figure 5: Model simulation results. Cumulative fraction of unrecovered particles $frac_{miss}$ (pink line), fraction of ambiguous locations $frac_{ambig}$ (blue), and fraction of correctly identified INPs $prob_{INP}$ (thin black) as function of search radius – originating from the reported INP position including errors - for different INP fraction $frac_{INP}$ and total particle numbers n. For this figure, a standard deviation of the positioning uncertainty of 25 μm is used. The total number of INPs $n_{INP}$ is shown for each plot. The search radius where 20% (blue), 10% (red), 5% (orange) and 1% (green) fraction / probability is reached is marked by vertical lines for $frac_{miss}$ (broken line) and for $frac_{ambig}$ (solid line). The lowest possible error probability range is indicated by a shaded area for below 20% (blue), below 10% (red), below 5% (orange) and below 1% (green). If no shaded area is visible, the error probability is greater than 20%.*

*Figure 6 shows the search radius of 50 μm around the identified ice crystal center point (blue point in Fig.4), without the error calculations.*

**Specific comments**

**-L155: "However, even after thorough cleaning a small amount of ice formation can regularly be observed at temperatures at or below -30°C, constituting the background concentration and defining the limit of detection, which is in the order of 0.1 L-1 of atmospheric air for a collection volume of 100 L." It appears that ice formation on silicon wafer only appears at -30°C, so is the background concentration is only for particles that nucleate ice at this temperature or below?**

*In general, ice nucleation can occur on cleaned substrates at warmer temperatures. Typically, however, the number of ice crystals on the wafer increases as the temperature decreases. In the present study, the lowest operating temperature of FRIDGE was -30°C. Since the highest number of ice crystals is typically observed at this temperature, the results at -30°C are affected the most by background concentration. The limit value was specified in relation to a specific volume. For warmer temperatures, the limit value is significantly lower.*
*Thus, this limit is based on the typical number of counts on cleaned wafers at -30°C, calculated on a sample volume. The limit can vary with the cleaning procedure, the sampling volume etc. and should be seen as a guideline.*

**-L234: "can be caused" here you employ a modal verb which suggests that condensation is only a possible explanation for the higher particle counts observed with FRIDGE compared to SEM. Is there any evidence or analysis that could strengthen this claim? In other words, is it not possible to determine with greater certainty whether condensation is indeed the cause?**

*The choice of words may be somewhat misleading.*
*Condensation is undoubtedly the cause of the higher number of positions identified for SEM. We observed this quite frequently in higher amounts during the initial phase of the development of the method. There may be other reasons, but they are rare (and therefore of minor importance). Condensation, however, cannot always be avoided, and it is sometimes visible though the whole picture series, especially at a relative humidity close to 100%. A compromise must then be made between the visibility of condensation and the size of the ice crystals.*
*We changed the wording as follows.*

*"Higher numbers of ice crystal positions identified for SEM are mainly caused by misclassifying areas with condensation, which may occur while working close to RH = 100 %."*

**-L259: "mechanical movement" can you add an explanation?**

*The sample stage in the SEM can be moved mechanically by actuators in orthogonal directions. By positioning the wafer accordingly on the stage, the x- and y-axis directions of movement correlate with the x- and y-axes of the coordinate system on the wafer.*

*"This is necessary because the internal Cartesian SEM coordinate system is determined by the axes of the mechanical movement of the stage, whose position is encoded by high-precision encoders. The SEM coordinate system is centered in the middle of the stage (Fig. 3c)."*

**-Fig. 3b: There are 3 pictures (top left, bottom left and bottom right) with each calibration marks and 1 picture (top right) of the entire calibration system incorporation the 3 marks? Please add explanation.**

*We have added an explanation to the figure caption and the heading in b). In addition, we have changed figures 3a) and 3c) to clarify the coordinate transformation procedure and make it easier for the reader to follow.*

*„Figure 3: a) pixel coordinate system from the FRIDGE image (blue) and wafer internal normalized Cartesian coordinate system(orange) to locate ice crystal positions (white circle) from the FRDIGE images: the normalized coordinate system (orange) is defined by two normalized vectors (n01, n10 with a length of 100) identifying a position by its x- and y-coordinate; b) calibration image showing the marked calibration points for the coordinate system from the ice crystal identification step and a picture showing the entire calibration system (top right); c) wafer on the SEM stage showing the SEM internal cartesian coordinate system aligned to the directions of the mechanical movement in millimeters (mm); d) SEM picture of a calibration cross on the wafer surface.”*

**-Fig.4: Did you think about integrating the pixel size of this schematic? Looking at the radius of the inner circle (25 µm) and the area around, I don't see how there could be 30 adjacent pixels with brightness above the threshold in that area. Perhaps you can increase the size of ice crystal?**

*Changed as requested.*

**-L297: "The direction of shift is randomly chosen; the distance is randomly sampled from a mirrored normal distribution with a standard deviation of some typical uncertainty assumptions." Could you further explain what you mean by typical uncertainty assumption?**

*As outlined, there are errors leading to the uncertainty of the reported INP position. For instance, the INP may no longer be at the center of the crystal due to a particle shift (see Chapter 2.3), or asymmetric crystal growth could have caused an inaccurate determination of the center point (see Chapter 2.4). The limited resolution of the FRIDGE images compared to the SEM also introduces uncertainty (see Chapters 2.4 and 2.5.1).*
*Charrier (2016) defines the uncertainty of retrieving defined points on a wafer in the SEM with an average standard deviation of 15 µm. Taking a conservative approach, our estimation is based on the higher values from the experiment, as the inaccuracy due to the resolution of the FRIDGE images must also be considered.*
*As we do not know the uncertainties of the particle shift and the potentially asymmetric ice crystal growth, these errors were not taken into account for the simulation.*

*The paragraph reads now:*

*„The direction of shift is randomly chosen, the distance is randomly sampled from a mirrored normal distribution with a standard deviation of some typical uncertainty assumptions for the accuracy of relocating a defined point on the wafer in the electron microscope. The uncertainties associated with the particle shift and the potentially asymmetric ice crystal growth are not incorporated into the simulation, as their values are not yet sufficiently established.”*

**-L304: "(total number of 20000 / 50000 / 100000 particles on the wafer with different INP fractions of 0.0005 / 0.001 / 0.002)" Please discuss this further and add references.**

*The number of particles refers to the typical atmospheric aerosol concentrations and number we had in our samples.*
*We have added two references, dealing with atmospheric INP fractions.*

*"Figure 5 shows the results of a selection of parameters corresponding to typical application conditions (total number of 20000 / 50000 / 100000 particles on the wafer with INP fractions of 0.0005 / 0.001 / 0.002 (DeMott et al., 2017; Ren et al., 2023))."*

**-L305:" In this case the standard deviation of the position uncertainty of 25 μm was assumed."**
**Why did you choose this one?**

*See our answer on comment L297.*

**-Fig 5: I am having difficulty understanding how the probability of INP correctly identified (thin black line) is equal to 1 for the smallest search radii, but at the same time the fraction of INP missed is increasing as search radii are decreasing.**

*With a very small search radius around the calculated coordinate, there is a high probability that the particle found here is the true INP, as the chance that a non-INP is very close to that position is the lower, the smaller the search radius is.*
*But taking the position uncertainty into account, also the chance of missing the INP increases with decreasing search radius, as in this case even small position uncertainties lead to an empty position (i.e., ´miss´).*
*I.e., at a low search radius, due to the position uncertainty many INP are missed, but the ones that are found, have a high probability of having been the actual INP. Increasing the search radius leads to higher recovery rates, but also to higher uncertainties with respect to the particle being the actual INP.*
*To find an optimal radius, a compromise must be found between all parameters.*

**-Fig.4: Why don't you integrate the pixel size of this schematic? Looking at the radius of the inner circle (25 um) and the area around, I don't believe there are 30 adjacent pixels with brightness above the threshold in that area.**

*Scheme modified as requested.*

**-L395: "From the statistical calculation in Tab. S1 with a 95% confidence level, a limit of approximately 10 particles per group can be derived to make a reliable quantitative statement, for groups with less particles the uncertainties become large" do you mean that since the 95% confidence level for less than 10 particles spans reaches 0 as lower limit, it is not possible to make a reliable quantitative statement?**

*Yes, that is what we meant.*
*If the lower and upper limits of the confidence interval are both above 0, it is certain that the corresponding group contributes to the ice formation process and you can give value with a corresponding confidence interval. However, if the lower limit of the confidence interval is 0, we cannot exclude that the group did not participate in the ice formation process within the specified uncertainty.*
*Our statement is only based on the calculations from the case study, and should be seen a guide line. As we stated in the manuscript, further considerations of limits or statements should employ common statistical approaches for counting statistics and compositional data and must be evaluated in the respective context.*
*However, the results with the given confidence intervals are still the best estimated value even if the uncertainties may be very large for small particle numbers. We indicate uncertainties for all values so that the reader can form his or her own opinion.*

**Technical corrections**

**-Fig.3: "polar coordinatesystem" change to "polar coordinate system"**

*For a better understanding, the whole figure 3a) is revised (see Fig. 3). The polar coordinate system is no longer part of the figure.*

**-Fig 3: in the cation is mentioned "d10" but not in the figure**.

*For a better understanding, the whole figure 3a) is revised (see Fig. 3). The polar coordinate system is no longer part of the figure.*

**-Fig 3: please add in the caption that white circle is an ice crystal.**

*Changed as requested.*

*„Figure 3: a) pixel coordinate system from the FRIDGE image (blue) and wafer internal normalized Cartesian coordinate system(orange) to locate ice crystal positions (white circle) from the FRDIGE images: the normalized coordinate system (orange) is defined by two normalized vectors (n01, n10 with a length of 100) identifying a position by its x- and y-coordinate; b) calibration image showing the marked calibration points for the coordinate system from the ice crystal identification step and a picture showing the entire calibration system (top right); c) wafer on the SEM stage showing the SEM internal cartesian coordinate system aligned to the directions of the mechanical movement in millimeters (mm); d) SEM picture of a calibration cross on the wafer surface."*

**-Fig. 4: "die" replace by "due"**

*Changed as requested.*

*„ Figure 4: Conceptual model for the simulation of the INP-recovery from FRIDGE in SEM. The blue circle is the true location of the INP, the yellow circles are non-INPs. The red cross is the search origin position in SEM. It deviates from the true INP position due to the errors by INP shift, unprecise ice crystal center determination and the re-positioning uncertainty due to calibration mismatch and mechanical backlash. The circles show search radii of 25, 50 and 75 μm from the red search origin position. The grid shows the approximate pixel size of the optical camera of FRIDGE."*

**-Fig. 4: "For this plot" there are several plots here, which one are you referring to?**

*We do not refer to a single plot, we refer to the whole figure. We have changed the wording accordingly.*

*Figure 5: Model simulation results. Cumulative fraction of unrecovered particles $frac_{miss}$ (pink line), fraction of ambiguous locations $frac_{ambig}$ (blue), and fraction of correctly identified INPs $prob_{INP}$ (thin black) as function of search radius – originating from the reported INP position including errors - for different INP fraction $frac_{INP}$ and total particle numbers n. For this figure, a standard deviation of the positioning uncertainty of 25 μm is used. The total number of INPs $n_{INP}$ is shown for each plot. The search radius where 20% (blue), 10% (red), 5% (orange) and 1% (green) fraction / probability is reached is marked by vertical lines for $frac_{miss}$ (broken line) and for $frac_{ambig}$ (solid line). The lowest possible error probability range is indicated by a shaded area for below 20% (blue), below 10% (red), below 5% (orange) and below 1% (green). If no shaded area is visible, the error probability is greater than 20%.*

**-L.280: "orange" it is yellow, no?**

*As the color was ambiguous, we now changed it clearly yellow and adapted the plot and text.*

**-L361: "Based on the modeling, we would consider a value of around 100,000 particles on the wafer to be a good starting point, as the proportion of incorrectly identified particles increases significantly with higher particle numbers." Consider changing "good starting point" as this can be understand that 100, 000 particles minimum are suitable, while higher number of particles increases uncertainty.**

*We adjusted the words. The section now reads as follows.*

*"Based on the modeling, we would consider a collection of around 100,000 particles on the wafer as a good number for meaningful measurements under typical free-tropospheric conditions. Of course, it has to be adapted to the actual conditions, e.g., when the fraction of INP is significantly different. Therefore, we recommend to determine the particle concentration in the atmosphere in parallel to the collection and then perform a quick analysis of the wafers in FRIDGE to calculate the proportion of INPs in the total aerosol. Based on this proportion the sampling parameter can be adjusted to balance a sufficient number of INPs on the wafer but prevent an overload."*

**Author's Response to Reviewer 4**

*First of all, we would like to thank the reviewer for reading our manuscript and pointing out some issues which might raise concerns by the reader.*

**This manuscript presented the details of a method for the INP or IR analysis by coupling a SEM and an offline ice nucleation diffusion chamber. The method has been applied in a few peer-reviewed papers without detail information by the authors. This manuscript is attempted to provide a full description on this method. The topic of this study fits the scope of this journal. There are several issues need to be addressed before it can be considered for publication.**

**Major comments:**

**1. There are other groups used a similar SEM/EDX method for INP and IR analysis which the authors didn't mention in the manuscript. Please see the review by Knopf et al. which shows a few studies that have been used the SEM/EDX analysis that coupled with ice nucleation measurements (https://pubs.acs.org/doi/10.1021/acsearthspacechem.7b00120). Although the novelty for this method is not fully described in this manuscript or is still in question, but I do think it may be worth for a publication as it provides the details for the method for their previous publication and future manuscripts.**

*We have added additional references to studies using SEM to analyze INPs.*

*"Even though the method cannot provide high temporal resolution measurements due to longer sampling times, it can provide detailed information on morphology in addition to chemistry and size of individual INPs and IRs (China et al., 2017; Cziczo et al., 2009, 2013; Ebert et al., 2011; Eriksen Hammer et al., 2018; Knopf et al., 2014; McCluskey et al., 2014; Mertes et al., 2007; Prenni et al., 2009; Wang et al., 2012; Worringen et al., 2015)."*

*As you have recognized, only papers showing results using this method have been published so far. But these papers do not describe the method or only describe it very briefly. The novelties of our manuscript have already been described and explained in our answers during the previous review discussion.*

**2. One major concern I have is on the misidentification of INPs. It is mainly due to the low spatial resolution of FRIDGE images during ice nucleation experiments. The successful rate is only 13% to 50%, 30% on average. That limits the number of INPs analyzed. If the authors can identify the INP one by one, then the interpretation on the chemical characterization of INPs remain valid even for a small number of identified INPs. However, that will require improve the spatial resolution of FRIDGE images significantly.**

*As already stated in the conclusions, we agree with the referee that a higher camera resolution in FRIDGE would improve the method a lot (see the following quote). However, we still are confident in the presented results of our methodology, provided that the uncertainties described are sufficiently addressed.*

*"Following the uncertainty analysis presented here, we identified the FRIDGE camera resolution as one important area of improvement. A higher camera resolution to document the ice crystal growth in FRIDGE would improve the accuracy of the ice crystal center point identification and make it easier to find the calibration point in the SEM. This could significantly reduce a substantial part of the uncertainties."*

**3. For the INP classification scheme, if I understood it right, the authors used only the information that whether an element is presented in the INP. The EDX should have provided relative content of each element. Why not use this quantitative information?**

*It is true that relative proportions of the individual elements can be obtained using EDX analysis. However, for our rather rough classification of the particle classes (silicates, carbonates, C-rich, etc.) due to low counting statistics, the classification based on the presence of the elements is sufficient. If the focus lies on subdividing the aluminosilicates into different groups (e.g., K-feldspar, etc.), we would have to do this on the basis of the percentages.*

**4. Details for the ice nucleation experiment are not sufficiently presented. What is the temperature uncertainty across the 45mm wide Si wafer? This is very important for ice nucleation experiments. In L185, -186, it is not clear. How was the ice nucleation conducted? How long did it allow for ice to nucleate after introducing the water vapor?**

*Several details of the ice nucleation experiment FRIDGE are published in previous papers (Schrod et al., 2016; DeMott et al., 2018; DeMott et al., 2025). The uncertainty of the nucleation temperature is estimated to be below 0.5°C for the wafer as a whole (DeMott et al., 2025). This was derived from spot measurements at the surface, as infrared-measurements are too inaccurate. We are confident that this 0.5°C upper bound is valid for the FRIDGE measurements, due to intercomparison of FRIDGE INP data with those from independent methods of other laboratories on test aerosols like ATD, cellulose, Snomax, Saharan dust as well as on ambient aerosol.*
*We added the temperature information and the reference to the manuscript.*

*"A small amount of silicon oil is applied on the bottom of the wafer as well as on the temperature sensor to ensure good thermal contact and a homogeneous temperature distribution. The temperature variance is estimated to be below 0.5°C across the entire wafer (DeMott et al., 2025). "*

*During the initial phase of the coupling procedure, the ice typically grew for 100 seconds before sublimating again (the evaluation of ice as function of time is given by Schrod et al., 2016 in Fig. 2). This time frame for was adopted from the pure FRIDGE experiments, where it was chosen to ensure that as many INPs as possible were activated and have grown significantly.*
*However, as described in the manuscript, large ice crystals introduce some difficulties and inaccuracies for coupling the activation procedure with INP identification experiments in SEM. As stated in the manuscript, the accuracy of INP recovery can be increased if the ice crystals do not grow to large ("are at a stage close to activation"), as particle shifts and possible asymmetries in ice crystal growth are limited. Therefore, the typical growth time of an ice crystal in FRIDGE is currently about 40 seconds, which is enough to activate most of the INPs.*

**5. The contribution of ice nucleation on the blank Si wafer. L346-352, Ice crystal can also form on the blank Si wafer. If you can not find an INP within a certain area, that could mean it formed ice on the blank. Then, increasing the search radius is not helping to increase the number of identified INPs but increase the misidentified INPs. This is also critical to the calculation of INP concentration. In L516, Section 3.2, does the INP concentration calculation consider the blank correction?**

*There are two different aspects, which have to be distinguished:*
*(1) Even on a cleaned wafer, a weak ice crystal growth can be observed in FRIDGE, which becomes stronger with decreasing temperature (for details see Section 2.1 in the manuscript and our answer on comment on L155 from Reviewer 3). These grown ice crystals are called background counts. Reasons for background counts can be residues or particles remaining on the wafer after cleaning, but defects of the wafer surface can also contribute to background counts. Ice crystal growth on surface defects may appear at temperatures primarily below -30°C and contamination particles on the wafer are more likely for the temperature range used in our studies. The number*

*of ice crystals counted in an atmospheric sample in FRIDGE is always corrected by the background counts when calculating the concentration.*
*Certainly, these particles can also be counted as INPs in the real sample, which means that contamination particles can be included in the characterization of the atmospheric aerosol. However, since the wafers are cleaned in advance to keep the background freezing at a low level, the influence of potential contamination particles can be assumed to be minimal.*

*(2)* *During the observation of the individual ice crystal positions in the SEM, it is possible that no particle is found at an identified position (within the selected radius). We refer to such a situation as "blank". There are many potential reasons for this: possible particle shift during the processing in FRIDGE, erroneous position of the ice crystal origin and incorrectly selected search radius (see chapter 2.5.3). The reasons for a blank position in the SEM are mainly due to reasons not related to the FRIDGE part of the method, it is therefore not necessary to correct the concentration obtained from FRIDGE by these "blank" positions.*

**Author's Response to the Editor**

**Dear Authors,**

**I have received two more reviews of your revised manuscript. After going through the reviews and the manuscript, I agree that there have been substantial improvements to the manuscript. Yet some concerns remain and these need to be addressed as indicated in the reviews you have received. I think these can be easily addressed and will improve the manuscript further.**

**Upon reading the reviews and paper, one aspect that stood out to me as raised by a previous round of reviews that I think you need to sufficiently address or justify is evaluating the efficiency of the coupling method (FRIDGE and SEM) with known reference aerosol. i.e. how sure are you that the SEM analysis only detects the particles that nucleated ice based on the co-ordinate system and the distance between the particles. If a lab controlled study of different aerosol types were used, then this can be verified by the particle composition obtained. However, with ambient particles, you can have a variety of compositions nucleating ice because of internal mixing. Could you also address this in your reviews or as a separate response to my comment.**

**I consider this round to be a mix between minor and major revisions. As such a step forward towards a final version. However, I will indicate major revisions so I can get at least one opinion from an external review in addition to my own review after you re-submit a revised version.**

**Best,**
**Zamin A. Kanji**

*We can understand the editor's point of view, that laboratory experiments with test aerosol may be a good chance to evaluate a method.*
*However, the method is used for identification of ambient atmospheric particles with complex mixing state which is more complicated and it is not certain that the method, if verified with pure test aerosol, will work the same way as with ambient aerosol. Samples from an atmospheric mixture of particles with a variety of compositions and probably a high excess of acids and volatile compounds can behave different from pure compounds. Since the individual steps had been verified separately (see list below), our aim for the coupling was to develop and evaluate the method under the more complex situation to show the real uncertainties that apply to the atmospheric situation. Consequently, the whole coupling procedure with the ESEM was tested using ambient particles only.*

*As mentioned above, the individual steps of the FRIDGE-ESEM method were verified separately and independently.*
*-Sampling and analysis in FRIDGE were tested with AgI, Fluorescein aerosol and haematite (Schrod et al. 2016).*
*-The identification of ice crystal origins on the wafer and the coordinate calculation by our counting algorithm was also repeated. However, as this is an automated process, it always provides the same output with the same input, in this case the same coordinates.*
*-To locate the identified ice crystal center coordinates from the wafer internal coordinate system in SEM, the recovery of the calibration points is essential. For this step, we provide a value of 20 μm for the inaccuracy due to the limited resolution of the FRIDGE images. Repeated calibration on the basis of the calibration picture resulted in a deviation of ± 10 μm. As this deviation does not exceed the inaccuracy due to image quality, it was not considered further in the manuscript.*
*-For the uncertainty of recovering defined points in the SEM a mean deviation of 15 μm was determined.*

*Laboratory experiments conducted during the early phase of the method have proven that if you deposit good ice nuclei, such as AgI, onto the wafer surface, you will only find/identify these good INP at the approached positions.*

---

## Author Response (AR4)

*In the following, the reviewer comments are written in bold and our answers in italics. Text passages from the revised manuscript are in quotation marks, modified or newly added passages are marked in green.*

**Author's response to Reviewer 4**

*First of all, we would like to thank the reviewer for reading our revised manuscript and our answers to the review comments.*

**The authors have addressed most of my comments. It would be helpful if the authors can comment on the follow minor issue. In your response to my last comment. You mentioned that "Certainly, these particles can also be counted as INPs in the real sample, which means that contamination particles can be included in the characterization of the atmospheric aerosol. However, since the wafers are cleaned in advance to keep the background freezing at a low level, the influence of potential contamination particles can be assumed to be minimal." Is there any number to show that the background contribution is minimal? For example, with same experimental procedure, what are the ice crystal numbers on the substrates in general after cleaning or before sampling? What are the ratios when compare to the atmospheric samples?**

*In general, all wafers are cleaned before they are used as sample substrates. After the cleaning process, the wafers are subject to random checks in FRIDGE (see Section 2.1). Typically, even cleaned wafers show low ice formation activity. For tropospheric measurements with typical INP concentrations (measured with FRIDGE) ranging from 0.1 to 10 $L^{-1}$ at -30°C (Schrod et al., 2020b) and up to 100 $L^{-1}$ in dust plumes (Schrod et al., 2017), a set of ten wafers is considered as clean if three randomly selected wafers from the set have an ice crystal number of less than ten during the -30°C measurement in FRIDGE. Background counts can subsequently be converted into a volume-dependent limit value (e.g., 0.1 $L^{-1}$ for a sampling volume of 100 L).*
*Basically, background counts are always subtracted from the raw INP counts of each sample before calculating the INP concentration.*
*This background value can be adjusted by setting an individual threshold for defining a set of clean wafers.*

*Since particularly low INP concentrations were expected in the free troposphere at JFJ in advance, particular emphasis was paid to the purity of the sample substrates. The blank value for the campaign was therefore demanded to be particularly lower than ten (less than 3 ice crystals per wafer at -30°C). This results in a maximum background concentration of 0.03 $L^{-1}$ for a collection volume of 100 L. As the collection volumes were generally larger than 100 L, the value decreases accordingly (ranging from 0.026 $L^{-1}$ for the minimum collection volume of 115 L to 0.001 $L^{-1}$ for the maximum collection volume of 306 L). The INP concentrations determined with FRIDGE at -30°C varied between 0.1 and 1 std$L^{-1}$ for the JFJ samples, the background values are therefore 1 to 2 orders of magnitude smaller.*

*As a result, for the campaign at JFJ we assume that an average number of 3 contamination background particles per sample is activated during the FRIDGE measurement at -30°C. Of course, these particles can subsequently contribute to the physico-chemical characterization of the INPs with SEM. But assuming that only 30% of the ice crystals can be assigned a unique INP, the number of contributing background INPs would be reduced from 3 to 0.9 per sample.*

*Martin (2019) found indications, that the group of C-rich particles appears to be primarily affected by this contamination. However, the number of identified particles was low. Results from a recently ongoing experiment show similar results, but the number of samples still needs to be increased in order to obtain a statistically significant statement.*

*Martin (2019): Eiskeim-Charakterisierung von Proben des Stormpeak Laboratory in Colorado, bachelor thesis, TU Darmstadt, 2019*

---

## Author Response (AR5)

**Author's response to the Editor**

*In the following, the editor comments are written in bold and our answers in italics. Text passages from the revised manuscript are in quotation marks, modified or newly added passages are marked in green.*

**Dear Authors,**

**I just have a minor comment. The background you mention in your response to the reviewer comment ranges from 0.026-0.001/L. However in the revised manuscript you wrote that the background was 0.1/L. This doesn't seem consistent. Even if it cannot be said that with statistical confidence, one should acknowledge that in the manuscript that the background is 0.1/L but can be as low as 0.026-0.001, but with less confidence.**

*The background value of 0.1/L is a typical value for the standard cleaning procedure, which is why we mention it in the general procedure description (Section 2.1). For the CLACE/INUIT campaign, we paid particular attention to cleaning the wafers, which resulted in these low background values. We now also mention these individual background concentrations in Sect. 2.1.*

*„However, even after thorough cleaning a small amount of ice formation can regularly be observed at temperatures at or below -30°C, constituting the background concentration and defining the limit of detection. This limit is volume-dependent and typically in the order of $0.1\ L^{-1}$ of atmospheric air for a collection volume of 100 L. However, for the case study, an especially low background value of $0.03\ L^{-1}$ was achieved (based on a collection volume of 100 L), which decreased further to between $0.026\ L^{-1}$ and $0.001\ L^{-1}$ when the actual collection volumes were considered."*